# ChaosNexus: A Foundation Model for ODE-based Chaotic System Forecasting with Hierarchical Multi-scale Awareness

Chang Liu [* 1]  Bohao Zhao [* 1]  Jingtao Ding [1]  Yong Li [1]

## Abstract

Foundation models show great promise in achieving zero-shot or few-shot forecasting for ODE-based chaotic systems via large-scale pretraining. However, existing architectures often fail to capture the multi-scale temporal structures and distinct spectral characteristics of chaotic dynamics. To address this, we introduce Chaos-Nexus, a foundation model for chaotic system forecasting underpinned by the proposed Scale-Former architecture. By processing temporal contexts across hierarchically varying patch sizes, ChaosNexus effectively captures long-range dependencies and preserves high-frequency fluctuations. To address heterogeneity across distinct systems, we integrate Mixture-of-Experts (MoE) layers into each ScaleFormer block and explicitly condition the final forecasts on a learned frequency fingerprint, providing the model with a global spectral view of the system. Extensive evaluations on over 9,000 synthetic systems demonstrate that ChaosNexus achieves superior fidelity in long-term attractor statistics while maintaining competitive point-wise accuracy. Furthermore, in real-world applications, it achieves a remarkable zero-shot mean error below 1°C for 5-day station-based weather forecasting. Code is available at https://github.com/tsinghua-fib-lab/ChaosNexus.

## 1. Introduction

Chaotic systems, characterized by deterministic dynamics yet extreme sensitivity to initial conditions, pervade diverse scientific domains including weather forecasting (Shukla,

*Equal contribution  [1]Department of Electronic Engineering, BNRist, Tsinghua University, Beijing, China. Correspondence to: Jingtao Ding <dingjt15@tsinghua.org.cn>, Yong Li <liyong07@tsinghua.edu.cn>.

*Proceedings of the 43$^{rd}$ International Conference on Machine Learning*, Seoul, South Korea. PMLR 306, 2026. Copyright 2026 by the author(s).

1998; Rind, 1999), fluid dynamics (Yorke & Yorke, 2005; Najm, 2009), and neural processes (Jia et al., 2023; Vignesh et al., 2025). While this sensitivity precludes precise long-term pointwise prediction, chaotic behavior is confined to a strange attractor (Rössler, 1976; Grassberger & Procaccia, 1983) with invariant statistical properties. Effective forecasting models should capture both short-term evolution and the long-term geometry and statistics of the system's attractor.

The intrinsic difficulty of forecasting chaotic systems is compounded by data sparsity in real-world applications. Traditional system-specific models (Srinivasan et al., 2022; Brenner et al., 2022; Hess et al., 2023) require extensive, high-quality observational data to accurately infer the underlying dynamics and attractor geometry of each novel system, creating a significant bottleneck. This motivates a paradigm shift toward pretraining universal models on diverse synthetic dynamical systems (Jiao et al., 2025; Hemmer & Durstewitz, 2025; Lai et al., 2026). By learning patterns from large collections of simulated chaotic systems, such models aim to generalize to unseen target systems with minimal or no in-distribution data. Panda (Lai et al., 2026) extends patch-based transformers with channel attention and dynamics-informed embeddings to handle diverse multivariate dynamics, while DynaMix (Hemmer & Durstewitz, 2025) utilizes a CNN-based encoder to interpret context signals and adaptively select regime-specific RNN experts.

However, existing works overlook the unique spectral characteristics of chaotic dynamics, making their solutions less effective. As illustrated in Figure 1, these characteristics present distinct modeling challenges. First, Figure 1(a) reveals that chaotic systems exhibit significantly higher spectral entropy compared to general time series. Unlike periodic data where information concentrates in narrow bands, chaotic systems distribute information across a continuum of scales. Standard Transformers like Panda, with fixed patch sizes, inevitably suffer from an information bottleneck on this continuum, either truncating long-range dependencies or overlooking fine-grained, high-frequency fluctuations. Second, Figure 1(b–c) highlights the profound heterogeneity among chaotic systems. Systems like the Lorenz-63 and Lorenz-96 display markedly different power spectra, implying that a single shared parameterization is suboptimal.

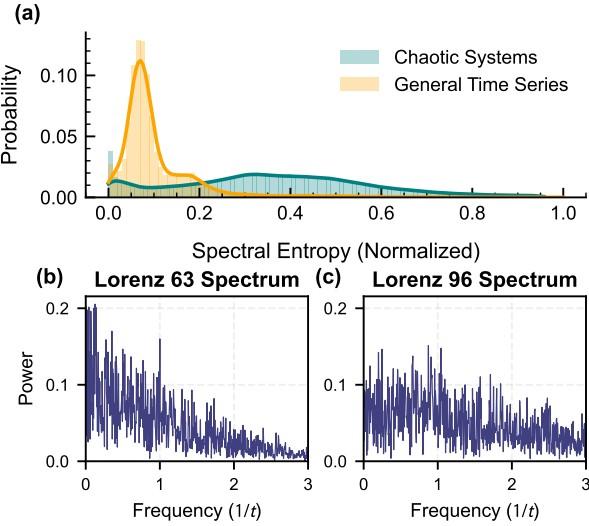

*Figure 1.* Motivating observations. (a) Spectral entropy distributions of synthetic chaotic systems (Lai et al., 2026) versus general time series (Liu et al., 2024) (including Electricity, ETT, and Exchange Rate). (b–c) Power spectra of representatives from Lorenz-63 and Lorenz-96 systems.

Although DynaMix attempts to address this heterogeneity by leveraging a pool of predictors, it relies on a simple CNN encoder to process context. Restricted by its local receptive field, the encoder fails to capture the global spectral signatures essential for distinguishing such diverse regimes.

To bridge this gap, we introduce ChaosNexus, a foundation model designed to adapt to the multi-scale, heterogeneous nature of ODE-based chaotic dynamics. At its core is our proposed ScaleFormer, a U-Net-inspired architecture that hierarchically processes temporal contexts. Its encoder progressively merges patches to capture coarse-grained global attractors, while the decoder reconstructs fine-grained details via patch expansion, ensuring fidelity across different frequency bands. To tackle the cross-system heterogeneity, we incorporate a Mixture-of-Experts (MoE) mechanism into each ScaleFormer block. Crucially, recognizing that distinct power spectra define the system's identity, we condition the model on a frequency fingerprint derived from a wavelet scattering transform. It adaptively modulates the fusion of multi-scale representations that align with the target system's intrinsic energy distribution.

ChaosNexus is pretrained on the chaotic-system corpus introduced by Panda (Lai et al., 2026), consisting of approximately 20,000 synthetically generated ODE systems. Training is guided by a composite objective that jointly enforces short-term predictive accuracy and the preservation of long-term statistical properties. Through extensive experiments, we show that ChaosNexus sets a new state-of-the-art in zero-shot forecasting on chaotic benchmarks. Its remarkable

efficiency is further highlighted on real-world weather forecasting: ChaosNexus achieves zero-shot temperature MAE below $1°C$, outperforming competitive baselines even when they are fine-tuned on more than $470K$ samples from the target system. Our contributions are summarized as follows:

- We propose ChaosNexus, a foundation model for chaotic system forecasting built upon hierarchical ScaleFormer blocks. It disentangles global attractor geometries from local fluctuations, enabling robust zero-shot generalization to unseen systems with distinct spectral characteristics.

- To address the heterogeneity across chaotic systems, we augment the ScaleFormer blocks with MoE layers and also construct the frequency fingerprint of the system. These designs enable the model to allocate specialized parameters to distinct dynamical regimes while explicitly conditioning forecasts on global spectral statistics.

- We show that ChaosNexus achieves state-of-the-art zero-shot forecasting performance on thousands of synthetic chaotic systems and real-world data from weather observation stations. We also provide illustrative analysis to validate the efficacy of our architectural designs.

## 2. Related Works

**Chaotic System Forecasting.** Forecasting chaotic systems is a central challenge in science and engineering. Reservoir computing (RC)-based methods (Srinivasan et al., 2022; Gauthier et al., 2021; Li et al., 2024) employ fixed read-in weights to lift inputs into the high-dimensional state space of a randomly initialized reservoir, while training only a linear readout. Deep learning models like recurrent neural networks (RNNs) often require techniques such as teacher forcing to counteract training instabilities on chaotic trajectories (Brenner et al., 2022; Hess et al., 2023). More recent works aim to preserve the geometric and statistical properties of system attractors within neural operators. This is achieved through evolution regularization with optimal transport and Maximum Mean Discrepancy (MMD), or by imposing mathematical constraints such as unitarity that leverage system ergodicity (Cheng et al., 2025; He et al., 2025). These frameworks are trained for a single, specific system. This inherent lack of generalization renders them impractical for unseen chaotic systems, precluding their application in zero-shot or few-shot forecasting.

**Out-of-distribution Generalization in Dynamical Systems.** Norton et al. (2025) show that reservoir computers can generalize to unobserved basins of attraction in multistable systems when trained on sufficiently rich transient dynamics. Another prominent strategy involves decomposing system dynamics into shared and specific components, where a base model captures common physical laws and low-dimensional vectors encode system-specific character-

istics, leveraging data from multiple regimes to learn fundamental representations of the underlying dynamics (Brenner et al., 2025; Wang et al., 2025; Huang et al., 2023). A complementary paradigm focuses on pretraining foundation models on large synthetic datasets encompassing diverse governing equations, parameter regimes, and initial conditions (Nzoyem et al., 2025; Subramanian et al., 2023; Herde et al., 2024; McCabe et al., 2024; Seifner et al., 2025), and most of these works target PDEs with rich spatiotemporal structure. Within the domain of ODE-based chaotic systems, Panda (Lai et al., 2026) trains Transformer blocks on a large-scale corpus of synthetic chaotic systems and demonstrates strong zero-shot forecasting performance on many unseen systems. DynaMix (Hemmer & Durstewitz, 2025) instead employs a mixture of almost-linear RNN experts with a CNN encoder to process context. Although these works clearly demonstrate the benefits of pretraining for generalization, their architectures either overlook the inherent multi-scale temporal structure of chaotic dynamics or fail to capture the global spectral signatures.

## 3. Methodology

**Problem Statement and Model Overview.** We address the problem of ODE-based chaotic system forecasting: given historical observations $\boldsymbol{X}_{1:T} = (\boldsymbol{x}_1, \boldsymbol{x}_2, \cdots, \boldsymbol{x}_T) \in \mathbb{R}^{T \times V}$ spanning $T$ times of a chaotic system with $V$ variables, we forecast its successive $H$ steps, *i.e.*, $\hat{\boldsymbol{X}}_{T+1:T+H} = f_\theta(\boldsymbol{X}_{1:T}) \in \mathbb{R}^{H \times V}$, where $f_\theta$ denotes the forecasting model. Here, we aim to design a foundation model $f_\theta$ that can directly produce faithful forecasting results based on historical observations, with little or no further in-distribution data required for training. We demonstrate the overall architecture of ChaosNexus in Figure 2, which comprises three key components: (i) input dynamics embedding, (ii) the ScaleFormer backbone, and (iii) frequency-enhanced joint scale readout. The details are shown as follows.

### 3.1. Input Dynamics Embedding

In chaotic systems, instantaneous observations are often noisy and insufficient to reveal the governing dynamics. We therefore segment the input trajectory $\boldsymbol{X} \in \mathbb{R}^{T \times V}$ into $S = \lfloor \frac{T}{D} \rfloor + 1$ non-overlapping temporal patches of length $D$. Each patch $\boldsymbol{P} \in \mathbb{R}^{D \times V}$ encapsulates a short-time trajectory segment, thereby providing essential local dynamical context. Motivated by Koopman theory (Koopman, 1931; Mauroy et al., 2020; Brunton et al., 2022) that nonlinear dynamics can be linearized by lifting them to a high-dimensional space of observables, we first enrich each patch with random polynomial and Fourier features (Appendix C.1) (Lai et al., 2026). The augmented patch is mapped to an embedding $\boldsymbol{u}$ of dimension $d_e$ by a linear layer.

### 3.2. ScaleFormer Architecture

**Architecture Overview and Theoretical Foundation.** Motivated by observations in Figure 1, we propose ScaleFormer, which is instantiated as a U-Net-style encoder-decoder architecture designed to explicitly model the multi-scale structure of chaotic systems. Its defining characteristic is the progressive coarsening of temporal resolution: the encoder recursively doubles the effective patch size $L$ via patch merging layers, while the decoder reconstructs fine-grained details via symmetric patch expansion, bridged by skip connections. We reinterpret this structural design through the lens of signal processing, specifically governed by the Nyquist-Shannon Sampling Theorem (Shannon, 2006). It postulates that the effective sampling rate scales inversely with $L$, imposing a physical limit on spectral capacity $f_{\max} \propto (2L \cdot \Delta t)^{-1}$. Our design naturally constructs a spectral hierarchy: The input level, operating at the initial patch size ($L = D$), functions as a broadband encoder that preserves maximum bandwidth to resolve the rapid, high-frequency fluctuations characteristic of local chaotic divergence. As the encoder deepens and $L$ increases ($L > D$), the reduced effective sampling rate physically forces the layers to act as low-pass filters, attenuating noise to distill the system's slow manifold—the robust, low-dimensional geometric structure governing long-term evolution.

**Patch Merging at Encoder Blocks.** Following each encoder block at level $i$, a patch merging layer reduces the temporal resolution by a factor of two to enforce the low-pass filtering effect. Given the input $\boldsymbol{H}_{\mathrm{enc}}^{(i)} \in \mathbb{R}^{S/2^{i-1} \times V \times d_i}$, we separate features at even and odd time steps and concatenate them along the channel dimension. The merged output is obtained via a linear projection $\boldsymbol{W}_{\mathrm{enc}}^{(i)}$:

$$\boldsymbol{H}_{\mathrm{enc}}^{'(i)} = [\boldsymbol{H}_{\mathrm{even}}^{(i)} \mid \boldsymbol{H}_{\mathrm{odd}}^{(i)}]\boldsymbol{W}_{\mathrm{enc}}^{(i)} + \boldsymbol{b}_{\mathrm{enc}}^{(i)}, \qquad (1)$$

where $\boldsymbol{H}_{\mathrm{enc}}^{'(i)} \in \mathbb{R}^{S/2^i \times V \times 2d_i}$. This process progressively widens the receptive field, culminating in a bottleneck layer that captures coarsened global structures.

**Patch Expansion at Decoder Blocks.** Mirroring the encoder, the decoder reconstructs high-resolution dynamics. Each decoder block is followed by a patch expansion layer that doubles the temporal resolution. For the $i$-th decoder level, the input $\boldsymbol{H}_{\mathrm{dec}}^{(i)}$ is up-sampled via a linear transformation and reshape operation:

$$\boldsymbol{H}_{\mathrm{dec}}^{'(i)} = \mathrm{Reshape}(\boldsymbol{W}_{\mathrm{dec}}^{(i)} \boldsymbol{H}_{\mathrm{dec}}^{(i)} + \boldsymbol{b}_{\mathrm{dec}}^{(i)}). \qquad (2)$$

**Skip Connections.** To ensure precise reconstruction, we bridge the encoder and decoder with skip connections. The output of the $i$-th encoder layer is processed by a 1D convolution block and fused with the corresponding decoder features, allowing the model to recover fine-grained details

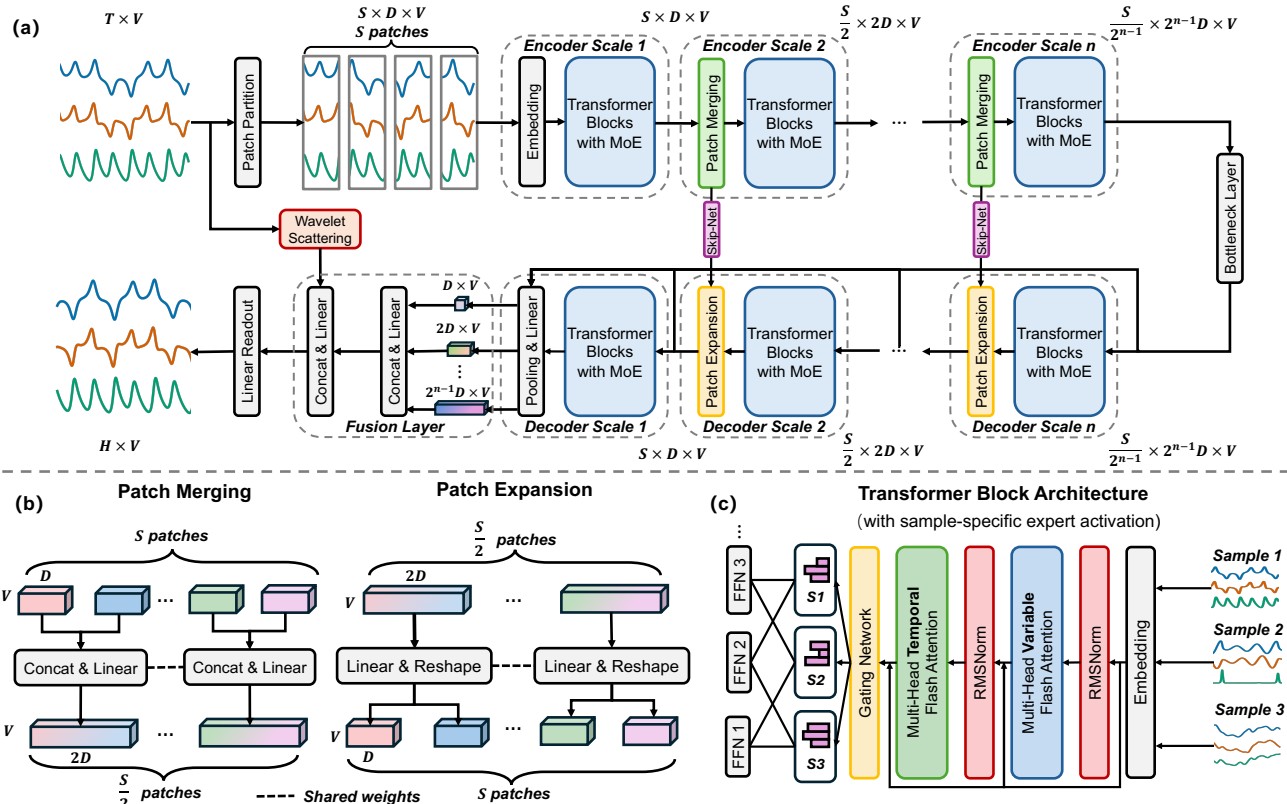

*Figure 2.* Overview of our ChaosNexus framework, with details of patch merging and expansion operations, and the Transformer block architecture with mixture-of-experts layers.

that are filtered out during encoding. Further details are provided in Appendix C.2.

**MoE-Enhanced Transformer Block.** Within each scale, the latent representations are processed by a modified Transformer block, specifically equipped with Mixture-of-Experts (MoE) to master the heterogeneity of chaotic systems. We first employ dual axial attention to factorize computation into sequential variable (VA) and temporal (TA) axes, reducing complexity from $\mathcal{O}(S^2V^2)$ to $\mathcal{O}(S^2 + V^2)$. Crucially, VA explicitly models the inter-variable coupling—a fundamental property of chaotic synchronization often neglected in standard time series models. We employ rotary positional embeddings (RoPE) (Su et al., 2024) to accommodate varying sequence lengths, and we utilize pre-normalization for training stability and FlashAttention (Dao et al., 2022) for efficiency. To distinguish diverse dynamical regimes, we replace the standard feed-forward network with a sparse MoE layer (Dai et al., 2024). Let $\hat{x}$ denote the RMSNorm-normalized input, the computational flow is as follows:

$$\boldsymbol{z}_p = \boldsymbol{u}_p + \text{VA}(\hat{\boldsymbol{u}}_p), \quad \boldsymbol{h}_p = \boldsymbol{z}_p + \text{TA}(\hat{\boldsymbol{z}}_p), \quad (3)$$

$$\boldsymbol{o}_p = \boldsymbol{h}_p + \text{MoE}(\hat{\boldsymbol{h}}_p). \quad (4)$$

The MoE layer comprises $M$ specialist experts $\{E_j\}_{j=1}^M$ and one shared expert $E_s$. A gating network selects the top-$K$

specialists based on the input's dynamical characteristics:

$$\text{MoE}(\hat{\boldsymbol{h}}_p) = g_{s,p}E_s(\hat{\boldsymbol{h}}_p) + \sum_{i \in \mathcal{I}_p} s_{i,p}E_i(\hat{\boldsymbol{h}}_p), \quad (5)$$

$$\mathcal{I}_p = \text{TopK}(\boldsymbol{s}_p, K), \quad (6)$$

$$\boldsymbol{s}_p = \text{Softmax}(\boldsymbol{W}\hat{\boldsymbol{h}}_p), \quad g_{s,p} = \sigma(\boldsymbol{W}_s\hat{\boldsymbol{h}}_p), \quad (7)$$

where $\boldsymbol{s}_p \in \mathbb{R}^M$ represents the routing scores, $g_{s,p}$ is the shared gate, and $\sigma(\cdot)$ is the Sigmoid function.

### 3.3. Frequency-enhanced Joint Scale Readout

The decoder of ScaleFormer produces a set of representations $\{\boldsymbol{H}_{\text{dec}}^{(i)}\}_{i=1}^L$ capturing system dynamics at $L$ different temporal scales. To synthesize these into a single, comprehensive representation for forecasting, we first apply temporal mean pooling to each decoder output to obtain system-level features $\bar{\boldsymbol{H}}^{(i)}$ for each scale. These features are then concatenated and projected through a linear fusion layer to produce a unified dynamics representation $\boldsymbol{H}_{\text{uni}} \in \mathbb{R}^{d_e \times V}$, which integrates multi-scale information:

$$\boldsymbol{H}_{\text{uni}} = [\bar{\boldsymbol{H}}^{(1)} \| \dots \| \bar{\boldsymbol{H}}^{(L)}]\boldsymbol{W}_f + \boldsymbol{b}_f. \quad (8)$$

A robust foundation model must not only model temporal

evolution but also identify the underlying dynamical system or its current regime. To this end, we condition our model on frequency-domain information, which serves as a fingerprint for the system's dynamics. We employ the wavelet scattering transform on the historical observations $\boldsymbol{X}$ to extract a stable, multi-scale summary of its spectral content (Appendix C.3). The resulting scattering coefficients, $\boldsymbol{F}_w \in \mathbb{R}^{C \times T' \times V}$, are temporally pooled to yield a single frequency fingerprint $\bar{\boldsymbol{F}}_w \in \mathbb{R}^{C \times V}$. It distills the system's intrinsic oscillatory and modulatory behaviors into a fixed-size representation, enhancing the model's ability to distinguish between different dynamical systems. Since $\bar{\boldsymbol{F}}_w$ is a system-level descriptor after temporal pooling, we use it as a late-stage conditioning signal for regime-level calibration of the final decoded dynamics. Broadcasting the same global fingerprint to every patch-level MoE router would introduce a granularity mismatch, because the fingerprint no longer provides patch-specific evidence for local expert selection. The final multi-step forecast is produced by a linear prediction head that combines the unified dynamics $\boldsymbol{H}_{\text{uni}}$ and the frequency fingerprint $\bar{\boldsymbol{F}}_w$:

$$\hat{\boldsymbol{X}}_{T+1:T+H} = [\boldsymbol{H}_{\text{uni}} \, \| \, \bar{\boldsymbol{F}}_w]\boldsymbol{W}_o + \boldsymbol{b}_o, \qquad (9)$$

where $\boldsymbol{W}_o$ and $\boldsymbol{b}_o$ are learnable parameters. This allows the model to leverage both learned multi-scale temporal patterns and the system's intrinsic spectral properties to make accurate predictions. Appendix B.10 further validates this design against router-level conditioning and nonlinear readout alternatives.

### 3.4. Training Objective

The total objective function for ChaosNexus comprises three components: a primary forecasting loss, an auxiliary load-balancing loss for the MoE layers, and a distributional regularization term to preserve the system's statistical properties. The primary training objective is the Mean Squared Error (MSE), formulated as:

$$\mathcal{L}_{\text{mse}} = \frac{1}{B} \sum_{n=1}^{B} ||\hat{\boldsymbol{X}}_{T+1:T+H}^n - \boldsymbol{X}_{T+1:T+H}^n||_2^2, \qquad (10)$$

where $\hat{\boldsymbol{X}}^n$ and $\boldsymbol{X}^n$ are the predicted and ground-truth of the $n$-th trajectory in a batch with size $B$.

As is standard for Mixture-of-Experts (MoE) models, relying solely on the prediction loss can lead to expert load imbalance, where the gating network disproportionately favors a small subset of experts (Shazeer et al., 2017). This leaves other experts undertrained and limits the model's overall capacity. To mitigate this, we incorporate an auxiliary load balancing loss from Dai et al. (2024):

$$\mathcal{L}_{\text{balance}} = M \sum_{i=1}^{M} f_i r_i, \qquad (11)$$

where $f_i$ is the fraction of patches routed to expert $i$, and $r_i$ is the average routing probability assigned to it. This encourages more uniform expert utilization.

Due to the sensitive dependence on initial conditions in chaotic systems, point-wise accuracy is often insufficient for long-horizon forecasting. A robust forecast must also reproduce the geometric and statistical properties of the system's attractor. To enforce this, we introduce a regularization term based on the Maximum Mean Discrepancy (MMD, Appendix C.4), which minimizes the divergence between the state distributions:

$$\mathcal{L}_{\text{reg}} = \frac{1}{B^2} \sum_{i,j=1}^{B} \left[ \kappa(\hat{\boldsymbol{X}}^i, \hat{\boldsymbol{X}}^j) + \kappa(\boldsymbol{X}^i, \boldsymbol{X}^j) - 2\kappa(\hat{\boldsymbol{X}}^i, \boldsymbol{X}^j) \right]. \qquad (12)$$

Following prior work, we use a mixture of rational quadratic kernels for $\kappa$ (Schiff et al., 2024). The final objective is:

$$\mathcal{L} = \mathcal{L}_{\text{mse}} + \lambda_1 \mathcal{L}_{\text{balance}} + \lambda_2 \mathcal{L}_{\text{reg}}, \qquad (13)$$

where $\lambda_{1,2}$ are weighting hyperparameters.

## 4. Experiments

In this section, we present comprehensive experiments to evaluate the forecasting capabilities of our proposed model. We demonstrate the hyperparameter setting used in experiments in Appendix D. Appendix E describes the training setups and computational infrastructure.

### 4.1. Evaluation on Synthetic Chaotic Systems

**Setups.** We utilize the benchmark dataset consisting of synthetic chaotic systems from Panda (Lai et al., 2026). Its training set contains 20K novel chaotic ODEs. The held-out test set used for evaluation comprises 9.3K systems (Appendix F.1). Although ChaosNexus is designed for ODE-based chaotic system forecasting, spatially discretized PDE trajectories also induce finite-dimensional dynamical systems, making them a natural additional evaluation of whether the learned model transfers beyond the pretraining family. We therefore also evaluate performance on two PDE systems with increasing spatial complexity: the 1D Kuramoto–Sivashinsky (KS) dynamics and the 2D Von Kármán Vortex Street (VKVS) dynamics, as detailed in Appendix B.6. We use the symmetric mean absolute percentage error (sMAPE) (Lai et al., 2026) with 128- and 512-timestep horizons to evaluate point-wise forecasting accuracy. We consider the correlation dimension error ($D_{\text{frac}}$), the Kullback–Leibler (KL) divergence between system attractors ($D_{\text{stsp}}$), the largest Lyapunov exponent error ($D_{\text{Lyap}}$), and the weighted mean energy error ($\text{ME}_{\text{LRw}}$) to evaluate the fidelity in key statistical properties of system attractors (Zhang & Gilpin, 2025a). We compare our proposed method against

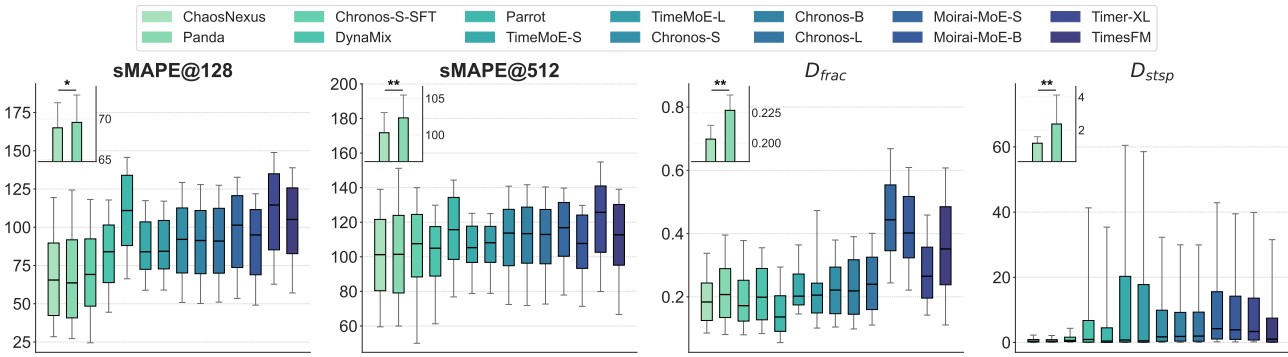

*Figure 3.* Zero-shot forecasting performances of models on synthetic chaotic systems. Each box shows the median (center line), the middle 50% of results (box), and the overall range (whiskers). The inset plot shows the mean performance with the 95% CI of ChaosNexus and Panda. Asterisks indicate statistically significant differences determined by the Wilcoxon signed-rank test (*: $p < 0.05$, **: $p < 0.01$).

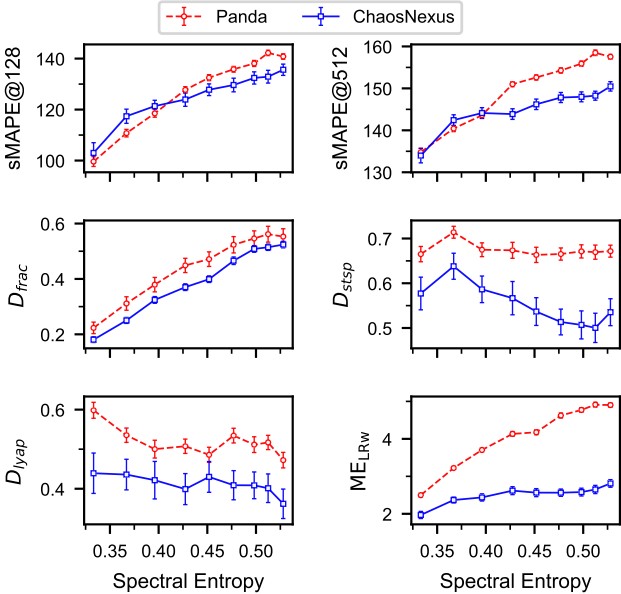

*Figure 4.* Results on Lorenz96 systems with different spectral entropy. We vary parameter $F$ to modulate spectral entropy. The panels display the mean performance with the 95% CI.

several state-of-the-art time series foundation models with different parameter sizes, including Panda (Lai et al., 2026), Time-MoE (Shi et al., 2025), TimesFM (Das et al., 2024), Chronos (Ansari et al., 2024), Moirai-MoE (Liu et al., 2025a), Timer-XL (Liu et al., 2025b), DynaMix (Hemmer & Durstewitz, 2025), Parrot (Zhang & Gilpin, 2025b), where "-S", "-B", and "-L" refer to small, base, and large parameter sizes, respectively. To assess the adaptability of general-purpose models to this specific domain, we also fine-tune the Chronos-S, Chronos-B, and Chronos-L on our chaotic systems training corpus. For all other baseline models, we load their officially released pre-trained weights for evaluation. We choose these baselines because they are all foundation models intended for generalization, aligning with our zero-shot evaluation on previously unseen chaotic systems. Appendix F details the experimental setups.

**Results of Zero-shot Forecasting.** We conduct a zero-shot evaluation on the held-out test set of chaotic systems. The results are shown in Figure 3 and detailed in Table 2. All models use a context length of 512 to autoregressively forecast 512 steps ahead. While ChaosNexus, Panda, and DynaMix are pretrained on the chaotic systems corpus, other baselines are general-purpose time-series foundation models, for which we employ the official pretrained weights. Chaos-Nexus demonstrates point-wise accuracy competitive with the baseline, achieving an average sMAPE@128 of 68.901. For long-term dynamics, ChaosNexus exhibits superior fidelity. It reduces $D_{\text{frac}}$ to 0.203. Notably, it attains $D_{\text{stsp}}$ of 1.206. Table 2 in Appendix B.1 further demonstrates the superior performance of ChaosNexus on $D_{\text{Lyap}}$ and $\text{ME}_{\text{LRw}}$. Given that the sensitive dependence on initial conditions renders any long-term point-wise forecast of a chaotic system ultimately unreliable (Li et al., 2022; Jiang et al., 2023; Schiff et al., 2024), the strong performance of ChaosNexus on long-term statistical metrics is compelling evidence that it can infer the intrinsic dynamics of new systems from context. Notably, leading general-purpose time-series foundation models, despite being pretrained on larger time-series datasets than ours (Appendix F.4), struggle with forecasting chaotic systems. We also observe that their generalization capabilities improve after further fine-tuning on the chaotic systems corpus, as evidenced by the results of Chronos-SFT. This contrast provides compelling evidence for our claim that chaotic dynamics differ from general time series. It also validates the need to build domain-specific foundation models on chaotic data. We further evaluate ChaosNexus on higher-dimensional ODE systems constructed from coupled Lorenz-63 dynamics with 768 and 1536 state dimensions, where it consistently outperforms baselines across point-wise and attractor-level metrics (Appendix B.2).

**Performance with Varying Spectral Entropy.** To verify our hypothesis that multi-scale representations are essential for capturing complex chaotic dynamics, we conduct a controlled experiment using the Lorenz-96

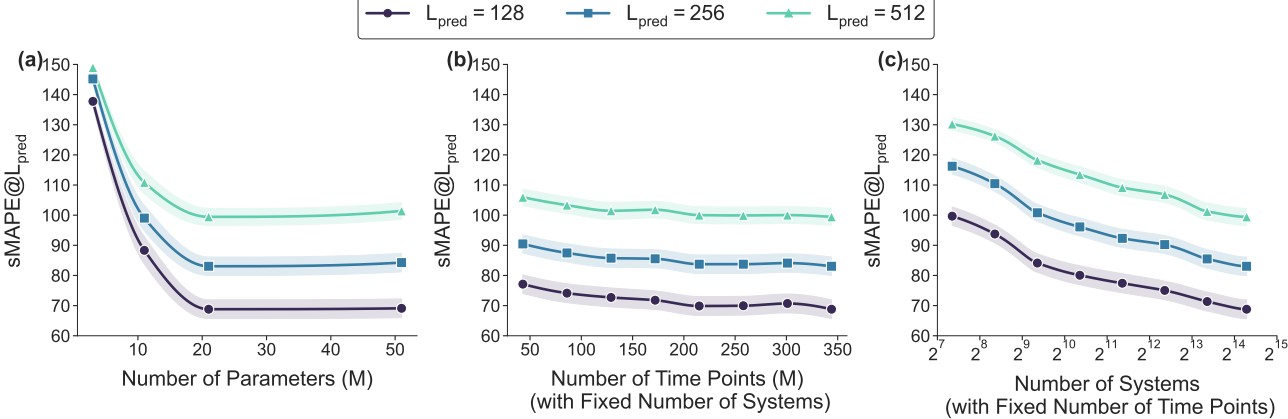

*Figure 5.* Scaling behavior of ChaosNexus. We demonstrate zero-shot sMAPE on synthetic chaotic systems varying: (a) the number of parameters; (b) the number of time points while holding the system diversity constant; and (c) the number of systems while holding the trajectories per system constant. Lines depict the average value, with shaded regions representing the 95% CI.

system (see Appendix F.2 for details). By systematically varying the external forcing parameter $F \in \{14, 20, 26, 32, 38, 44, 50, 56, 62\}$, we generate a spectrum of datasets ranging from weak to strong chaotic regimes. As $F$ increases, the system exhibits a monotonic rise in spectral entropy, indicating a transition toward increasingly rich multi-scale temporal structures that require capturing the hierarchy of time scales. The results in Figure 4 underscore the suitability of ChaosNexus for chaotic systems with inherent multi-scale structures. In terms of point-wise error (sMAPE@512), both models perform comparably in low-entropy regimes. As the spectral entropy increases, Panda exhibits a more rapid performance degradation than ChaosNexus. The advantage of our architecture is more pronounced in the energy and geometric metrics. The weighted mean energy error of Panda escalates dramatically to near $4.5$ as spectral entropy increases, and ChaosNexus maintains a nearly flat error profile with $\mathrm{ME}_{\mathrm{LRw}} \approx 2.5$. Similarly, ChaosNexus consistently achieves lower errors in $D_{\mathrm{frac}}$, $D_{\mathrm{stsp}}$, and $D_{\mathrm{lyap}}$ across all spectral entropy levels. This underscores the critical role of hierarchical modeling in forecasting high-entropy chaotic systems.

**Ablation Studies.** We conduct ablation studies to validate the effectiveness of our proposed architecture and training strategy. We evaluate four variants of our model by removing (i) patch merging and expansion operations, (ii) MoE layers, (iii) MMD-based auxiliary regularization, and (iv) frequency fingerprint. The results in Table 1 show that the full model achieves an effective balance between short-term point-wise accuracy and the preservation of long-term statistical properties. Removing patch merging and expansion causes a broad degradation across both point-wise and attractor-level metrics, confirming the importance of explicit multi-scale processing. Replacing MoE layers with standard feed-forward layers also weakens performance,

indicating that input-adaptive expert routing helps handle heterogeneous chaotic dynamics. Removing the MMD regularizer particularly harms long-horizon and distributional fidelity, while removing the frequency fingerprint consistently reduces performance by depriving the model of complementary spectral context. We further provide progressive component ablations in Appendix B.3.

*Table 1.* Model performances when removing each design. Reported values represent the mean ± 95% CI. (PME: Patch Merging and Expansion; MoE: Mixture-of-Experts Layers; MMD: MMD-based Auxiliary Regularization; FF: Frequency Fingerprint.)

| Model
Metrics | Full | w/o PME | w/o MoE | w/o MMD | w/o FF |
|---|---|---|---|---|---|
| sMAPE@128 | **68.901 ± 3.086** | 74.161 ± 3.082 | 73.076 ± 3.069 | 80.702 ± 3.217 | 72.699 ± 3.179 |
| sMAPE@512 | **100.293 ± 2.767** | 106.542 ± 2.516 | 105.298 ± 2.694 | 110.228 ± 2.771 | 104.002 ± 2.930 |
| $D_{\mathrm{frac}}$ | **0.203 ± 0.011** | 0.240 ± 0.010 | 0.220 ± 0.012 | 0.220 ± 0.010 | 0.215 ± 0.010 |
| $D_{\mathrm{stsp}}$ | **1.206 ± 0.392** | 1.820 ± 0.620 | 1.560 ± 0.310 | 1.460 ± 0.490 | 1.360 ± 0.440 |
| $\mathrm{ME}_{\mathrm{LRw}}$ | **1.562 ± 0.115** | 2.218 ± 0.152 | 1.870 ± 0.122 | 2.571 ± 0.164 | 1.771 ± 0.132 |
| $D_{\mathrm{Lyap}}$ | **0.065 ± 0.025** | 0.075 ± 0.019 | 0.072 ± 0.011 | 0.103 ± 0.032 | 0.082 ± 0.013 |

### 4.2. Scaling Behavior

Understanding scaling behavior is central to building foundation models, since it clarifies whether performance improves with model capacity and what kind of data expansion drives cross-system generalization. Figure 5 summarizes the scaling trends of ChaosNexus under changes in model size, per-system data volume, and system diversity.

**Parameter Scaling.** We first explore the impact of model size on performance. We generate models with varying parameter counts, ranging from $2.88M$ to $52.68M$, by systematically adjusting the number of encoder and decoder layers, as well as the dimension $d_e$ of the embedding space. The results in Figure 5(a) reveal a consistent trend: increasing the model's parameter count yields steady improvements in performance. For instance, scaling the model from $2.88M$ to $52.68M$ parameters improves sMAPE@128 by 49.83%, indicating that larger models possess a greater capacity to

*Figure 6.* Forecasting performance for global temperature on the WEATHER-5K dataset. The Mean Absolute Error (MAE) of ChaosNexus and baseline models is compared across multiple prediction horizons after fine-tuning on 85K (0.1%) and 473K (0.5%) samples. The zero-shot performance of ChaosNexus is shown as a dashed line for reference.

capture the complex dynamics inherent in the data.

**Data Scaling.** We further investigate performance as a function of training data size under two distinct settings. First, we fix the diversity, *i.e.,* the total number, of training systems, while varying the number of trajectories sampled from each system. Second, we increase the diversity of systems while holding the number of training time points constant. Figure 5(b) shows that merely increasing the number of time points for a fixed set of systems does not lead to a significant enhancement in zero-shot performance. In contrast, Figure 5(c) demonstrates that increasing the number of distinct systems substantially improves generalization. These findings support established research (Norton et al., 2025; Lai et al., 2026) on system-diversity scaling while refining it: effective cross-system generalization is driven primarily by corpus-level diversity, rather than by deeper sampling of the same systems.

### 4.3. Sim-to-Real Generalization on Weather Forecasting

**Setups.** Weather is an inherently chaotic system (Lorenz, 1969; 1982; 2017). For a rigorous evaluation on a real-world chaotic system, we utilize the WEATHER-5K dataset (Han et al., 2026). This dataset comprises hourly meteorological data from 5,672 weather stations worldwide over 10 years (2014-2023). It is chronologically split, with data from 2014 to 2021 used for training, 2022 for validation, and 2023 for testing. Each sample includes five variables: temperature, dew point, wind speed, wind direction, and sea-level pressure. Given the profound real-world importance of forecasting absolute values, we employ the gold-standard metric, Mean Absolute Error (MAE), to directly measure the discrepancy between predicted and ground-truth observations. The forecasting task is to predict the subsequent 120 hours of all variables given 512 hours of historical context. To assess few-shot performance under data-scarce conditions, we fine-tune models on two subsets of the training data: 0.1% (85K samples) and 0.5% (473K samples). ChaosNexus is first pretrained on the synthetic chaotic systems corpus and then fine-tuned on exactly the same WEATHER-5K subsets

as the baselines, which are trained from scratch without pretraining. Besides foundation models included in Section 4.1, we select strong deep learning baselines for this single-system, real-world benchmark, including FEDformer, CrossFormer, PatchTST, Koopa, and iTransformer. We report the zero-shot performance of our model. Further details of setups are provided in Appendix G.

**Results.** Figure 6 presents the forecasting results for the temperature variable. Remarkably, zero-shot ChaosNexus surpasses all baselines in their few-shot configurations. It achieves a mean error strictly below 1°C for 5-day (120-hour) global temperature forecasts. In contrast, the baseline models exhibit an MAE of at least 3°C, even when fine-tuned on the same data. The performance of ChaosNexus further improves with few-shot fine-tuning. This suggests that while pre-training endows the model with a robust, universal understanding of chaotic behavior, fine-tuning allows it to adapt these principles to the specific physical constraints and periodicities inherent in meteorological systems. This process grounds the model's abstract dynamical representations in real-world physics, enhancing its ability to generate accurate and stable long-term forecasts. Detailed results of all weather variables and performances of foundation models are shown in the Appendix B.14. We find that foundation models designed for chaotic system forecasting and trained on our corpus of synthetic chaotic dynamics, including ChaosNexus, Panda, and Chronos-SFT perform significantly better than those trained on general time series, even though they use a much larger corpus (see Table 13). It demonstrates that pretraining specifically on chaotic systems provides a more relevant inductive bias.

**Empirical Alignment between Synthetic and Real Data.** To investigate the origin of robust generalization performance on the WEATHER-5K dataset, we identify three systems from our synthetic pre-training corpus that are mathematically derived from or related to atmospheric models: *LorenzCoupled-Coullet*, *LorenzStenflo-VallisElNino*, and *SprottB-VallisElNino*. Lorenz systems are foundational models for atmospheric convection, while Vallis-El Nino

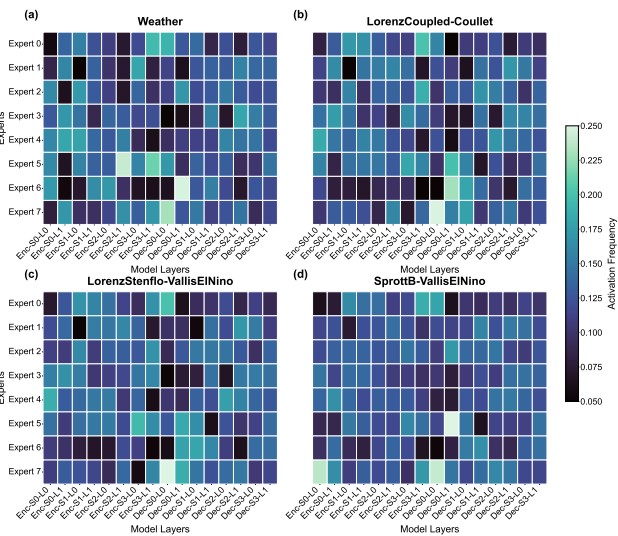

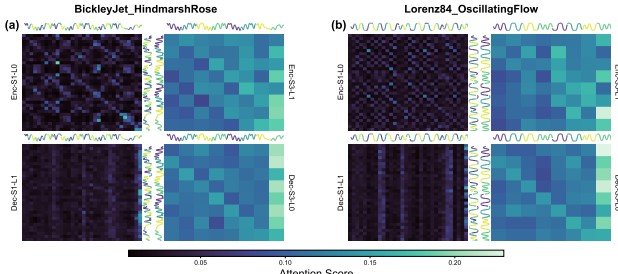

*Figure 8.* Visualization of input patch partitioning and multi-scale temporal attention for two chaotic systems. Each panel displays attention maps for the shallow (left) and deep (right) layers of the encoder (top) and decoder (bottom). S and L denote the scale level and the block within each level, respectively.

*Figure 7.* Expert activation patterns between Weather dataset and three synthetic systems derived from atmospheric physics. S and L denote the scale level and the block within each level, respectively.

variants are reduced-order models describing the El Nino-Southern Oscillation (ENSO). We visualize the gating probability distributions of expert activations elicited by these three synthetic systems alongside the activation pattern produced by the real-world Weather dataset in Figure 7, and observe a high degree of similarity in their expert utilization of different encoder and decoder layers. Crucially, this alignment is non-trivial; as detailed in Appendix B.14.3, unrelated chaotic systems exhibit markedly distinct activation patterns, confirming that the experts specialize in specific dynamical regimes rather than generic noise. This provides an empirical basis for ChaosNexus's robust generalization, confirming the transfer of physical priors and dynamical laws from synthetic counterparts to unseen real-world data.

### 4.4. Visualizing Multi-scale Temporal Attention

To investigate the inner workings of ScaleFormer, we visualize the input signal's patch partitioning alongside the temporal attention maps from shallow and deep layers of both the encoder and decoder. As illustrated in Figure 8 and 14, we select two systems from the test set with progressively weaker regularity (left to right in Figure 8).

**Patch Partition Patterns.** We find that shallow layers, operating on smaller patches, are adept at capturing high-frequency fluctuations. In contrast, deeper layers, processing merged patches of longer time intervals, capture long-term global structures. Particularly in Figure 8(b), a shallow-layer patch may encompass only a peak or a trough, whereas a deep-layer patch spans an entire peak-valley cycle.

**Temporal Attention Patterns of Encoder Layers.** The encoder's attention patterns distinctly reflect this multi-scale

processing. The deep encoder layers (upper right of each subfigure) consistently exhibit globalized attention distributions, indicating a focus on synthesizing long-range dependencies. The shallow encoder layers (upper left), however, display system-specific patterns. For the highly regular system in Figure 8(a), the model applies fixed-pattern filters to scan the time series. For the more complex system in Figure 8(b), the attention forms distinct blocks, indicating that the model concentrates on specific temporal segments whose interplay is critical for understanding the system's state.

**Temporal Attention Patterns of Decoder Layers.** Decoder's attention mechanisms operate differently, functioning primarily as a selector. It aligns with our design that the decoder's outputs are mean-pooled over the temporal dimension for the final forecast. The model should learn to select and combine specific patterns from the historical context to support its predictions. Deep decoder layers show a pronounced focus on the final patch, capturing the most recent temporal dependencies crucial for autoregressive prediction. Shallow decoder layers, conversely, appear to anticipate future dynamics; for instance, in Figure 8(b), after observing a descending phase, the model intensifies its attention on historical ascending patterns, selectively weighting the most relevant context for the anticipated future trajectory.

## 5. Conclusions

We introduce ChaosNexus, a foundation model designed to master the inherent multi-scale temporal structures and spectral heterogeneity of chaotic systems. By integrating the hierarchical ScaleFormer backbone with MoE layers and a wavelet-based frequency fingerprint, it achieves state-of-the-art zero-shot performance on both synthetic chaotic systems and real-world weather forecasting applications. Extensive ablation studies and in-depth analysis further validate the efficacy of these architectural components.

## Acknowledgements

This work is supported in part by National Natural Science Foundation of China under grant 62476152 and U23B2030.

## Impact Statement

The research presented in this paper focuses on the modeling of chaotic systems, with primary applications in scientific domains such as meteorology. All data used for training and evaluation is either synthetically generated from mathematical principles or derived from publicly available, non-personal scientific datasets, ensuring no privacy concerns. This work does not involve human subjects, and we do not foresee any direct negative societal impacts or risks of perpetuating social biases. Our aim is to advance the scientific understanding and predictive capabilities for complex physical systems for the benefit of the scientific community.

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

## A. Analysis Process of Figure 1

### A.1. Data Sources

To construct the comparative distributions shown in Figure 1, we aggregate data from two distinct domains:

**Chaotic system corpus.** It derives from the synthetic chaotic system dataset (both train and test splits) from Panda (Lai et al., 2026). We process each trajectory using a sliding-window approach with a window size $L = 1024$ and a stride of 512. Windows containing NaNs or infinite values are filtered out.

**Representative systems.** We numerically integrate two canonical chaotic systems, Lorenz63 and Lorenz96, over 20,000 time steps with a sampling interval of $\Delta t = 0.01$. The first 1,000 steps are discarded as a burn-in period to ensure convergence to the strange attractor. For Lorenz63 system, we simulate with standard parameters $\sigma = 10.0, \rho = 28.0, \beta = 8/3$ and initial conditions $[1.0, 1.0, 1.0]$. For Lorenz96 system, we simulate with dimension $N = 4$ and forcing parameter $F = 18.0$. The initial state is set to $x_i = 8.0$ for all $i$, with a small perturbation added to $x_2$.

**General Time Series.** We aggregate standard benchmarks widely used in Long-Term Time Series Forecasting (Liu et al., 2024), including Electricity, ETTh2, ETTm2, and Exchange Rate. These datasets represent typical empirical data characterized by strong periodicity and lower-dimensional dynamics, rather than deterministic chaos. We also process each trajectory using a sliding-window approach with a window size $L = 1024$ and a stride of 512. Windows containing NaNs or infinite values are filtered out.

### A.2. Spectral Entropy Calculation

Spectral entropy is calculated for each window as follows:

- **Standardization:** Each local window is standardized to zero mean and unit variance.

- **Power Spectral Density (PSD):** We apply the Fast Fourier Transform (FFT) to compute the power spectrum $S(f) = |\mathcal{F}(x)|^2$.

- **Probability Distribution:** The power spectrum is normalized to form a probability distribution $P(f_i)$:

$$P(f_i) = \frac{S(f_i)}{\sum_k S(f_k)}. \quad (14)$$

- **Normalized Shannon Entropy.** The spectral entropy is calculated as the Shannon entropy of $P(f)$, normalized by the maximum possible entropy (corresponding to white noise) to bound the value between $[0, 1]$:

$$SE = -\frac{\sum_i P(f_i) \log_2 P(f_i)}{\log_2(L/2)}, \quad (15)$$

where the normalization factor is the logarithm of the number of positive frequency components.

### A.3. Visualization Details

- **Distribution Plot:** The spectral entropy distributions are estimated using Kernel Density Estimation (KDE) with 100 bins.

- **Power Spectra:** Frequencies for the chaotic systems are normalized by the characteristic time scale ($1/t$).

## B. Supplementary Experimental Results

### B.1. Numerical Results on Synthetic Chaotic Systems

We demonstrate detailed numerical results corresponding to Figure 3 in Table 2 for reference.

### B.2. High-Dimensional ODE Generalization

To examine whether ChaosNexus remains effective beyond low-dimensional chaotic systems, we evaluate it on Erdős–Rényi-coupled Lorenz-63 systems (Li et al., 2024). We consider systems with 256 and 512 coupled Lorenz-63 subsystems, corresponding to 768 and 1536 state dimensions, respectively. The evaluation follows the zero-shot protocol used in Section 4.1: the model observes a 512-step context and forecasts future trajectories without system-specific fine-tuning.

As shown in Table 3, ChaosNexus consistently outperforms Panda in both high-dimensional settings. These results indicate that the learned multi-scale and heterogeneous dynamics representations transfer to substantially larger ODE systems, rather than being limited to low-dimensional chaotic benchmarks.

### B.3. Progressive Component Ablation

The main removal-based ablation is presented in Table 1. Here, we further conduct a progressive ablation in Table 4, where PME, MoE, and MMD are incrementally combined while keeping the frequency fingerprint disabled. This setting isolates the complementarity among the multi-scale architecture, heterogeneous expert routing, and distributional regularization; the final row therefore corresponds to the w/o FF variant in Table 1.

The progressive results show that combining PME with either MMD or MoE improves over using each component alone on most metrics. Adding both MoE and MMD yields the strongest variant in this comparison, with particularly clear gains on long-horizon dynamical metrics such as $ME_{LRw}$ and $D_{stsp}$. These results support that PME, MoE, and MMD address complementary aspects of chaotic system forecasting.

*Table 2.* Detailed numerical results of model performance on synthetic chaotic systems. The best performance of each metric is marked in **bold**, and the second-best performance is underlined. Reported values represent the mean ± 95% CI.

| Model
Metric | ChaosNexus | Panda | Chronos-S-SFT | Chronos-B-SFT | Chronos-L-SFT | Chronos-S | Chronos-B | Chronos-L |
|---|---|---|---|---|---|---|---|---|
| sMAPE@128 ($\downarrow$) | 68.901 ± 3.086 | 69.567 ± 3.358 | 70.510 ± 11.356 | 70.124 ± 12.761 | 69.765 ± 11.514 | 86.323 ± 33.031 | 86.883 ± 33.122 | 82.730 ± 32.165 |
| sMAPE@512 ($\downarrow$) | 100.293 ± 2.767 | 102.333 ± 3.123 | 101.947 ± 10.226 | 101.215 ± 13.497 | 100.824 ± 11.058 | 104.826 ± 32.191 | 104.156 ± 31.964 | 102.967 ± 31.827 |
| $D_{frac}$ ($\downarrow$) | 0.203 ± 0.011 | 0.227 ± 0.013 | 0.233 ± 0.165 | 0.224 ± 0.085 | 0.210 ± 0.053 | 0.233 ± 0.135 | 0.246 ± 0.143 | 0.219 ± 0.120 |
| $D_{stsp}$ ($\downarrow$) | 1.206 ± 0.392 | 2.369 ± 1.751 | 2.391 ± 10.651 | 2.837 ± 1.978 | 2.685 ± 1.652 | 11.498 ± 25.207 | 11.255 ± 24.561 | 11.731 ± 27.171 |
| $ME_{LRw}$ ($\downarrow$) | 1.562 ± 2.015 | 1.649 ± 0.413 | 1.580 ± 0.350 | 1.602 ± 0.260 | 1.571 ± 0.302 | 2.397 ± 2.698 | 2.373 ± 2.804 | 2.385 ± 2.871 |
| $D_{Lyap}$ ($\downarrow$) | 0.065 ± 0.025 | 0.067 ± 0.047 | 0.072 ± 0.023 | 0.068 ± 0.021 | 0.069 ± 0.024 | 0.082 ± 0.007 | 0.074 ± 0.008 | 0.072 ± 0.007 |

| Model
Metric | Moirai-MoE-S | Moirai-MoE-L | TimeMoE-L | TimeMoE-S | TimerXL | TimesFM | Parrot | DynaMix |
|---|---|---|---|---|---|---|---|---|
| sMAPE@128 ($\downarrow$) | 92.223 ± 35.279 | 95.103 ± 53.000 | 87.426 ± 13.411 | 87.186 ± 13.790 | 105.379 ± 36.289 | 100.933 ± 15.372 | 92.084 ± 16.764 | 70.381 ± 12.148 |
| sMAPE@512 ($\downarrow$) | 108.493 ± 30.777 | 109.446 ± 31.755 | 103.489 ± 12.238 | 103.143 ± 12.757 | 115.239 ± 34.773 | 108.211 ± 13.381 | 114.368 ± 14.724 | 102.966 ± 14.945 |
| $D_{frac}$ ($\downarrow$) | 0.423 ± 0.204 | 0.372 ± 0.209 | 0.230 ± 0.164 | 0.256 ± 0.310 | $\infty \pm$ nan | 0.364 ± 0.076 | 0.106 ± 0.157 | 0.145 ± 0.182 |
| $D_{stsp}$ ($\downarrow$) | 13.613 ± 27.323 | 13.581 ± 27.593 | 10.651 ± 25.348 | 11.542 ± 28.004 | 14.534 ± 30.619 | 9.655 ± 11.048 | 6.085 ± 17.528 | 6.904 ± 19.824 |
| $ME_{LRw}$ ($\downarrow$) | 3.181 ± 2.168 | 6.803 ± 4.842 | 8.700 ± 1.029 | 8.965 ± 1.013 | 3.925 ± 2.648 | 11.122 ± 0.606 | 0.654 ± 1.067 | 1.638 ± 2.372 |
| $D_{Lyap}$ ($\downarrow$) | 0.081 ± 0.012 | 0.075 ± 0.042 | 0.072 ± 0.014 | 0.068 ± 0.002 | 0.075 ± 0.009 | 0.069 ± 0.008 | 0.065 ± 0.012 | 0.067 ± 0.014 |

*Table 3.* Zero-shot performance on high-dimensional coupled Lorenz-63 systems. Results are reported as mean ± 95% CI.

| Metric | Dim=768 ChaosNexus | Dim=768 Panda | Dim=1536 ChaosNexus | Dim=1536 Panda |
|---|---|---|---|---|
| sMAPE@128 | **55.907 ± 0.438** | 69.962 ± 1.241 | **55.486 ± 0.369** | 67.033 ± 1.430 |
| sMAPE@512 | **117.901 ± 0.613** | 125.244 ± 0.508 | **117.396 ± 0.509** | 125.186 ± 0.554 |
| $D_{frac}$ | **0.080 ± 0.004** | 0.131 ± 0.005 | **0.092 ± 0.004** | 0.135 ± 0.005 |
| $ME_{LRw}$ | **1.148 ± 0.034** | 1.562 ± 0.022 | **1.101 ± 0.032** | 1.710 ± 0.024 |

## B.4. Sensitivity to Context and Prediction Length

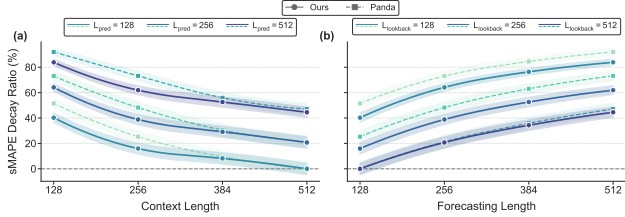

*Figure 9.* Performance Sensitivity of ChaosNexus and Panda to different (a) context length and (b) forecasting length. Lines depict the average value, with shaded regions representing the 95% CI.

**Performance with Different Context Length.** We evaluate our model across a range of input context lengths. As shown in Figure 9(a), our model's performance consistently improves with a longer context and consistently surpasses the baseline Panda model. It also shows less sensitivity to the specific context length chosen. These advantages of our model stem from its multi-scale architecture, which effectively leverages information across different temporal scales to build a more stable representation of the system's dynamics.

**Performance with Different Prediction Length.** Long-horizon forecasting serves as a crucial test of a model's capacity to learn the intrinsic dynamics of a chaotic system. Accordingly, our model's performance advantage over Panda becomes substantially larger at longer prediction horizons, as shown in Figure 9(b). It validates our design philosophy, which prioritizes multi-scale feature extraction and

dynamics discrimination to build a more faithful representation of the underlying system.

### B.5. Inference Efficiency

Table 5 demonstrates the computational efficiency of various foundation models in a long-term forecasting scenario. Specifically, we report the inference latency required to generate a prediction horizon of 512 time steps with a context length of 512 time steps. To ensure the statistical reliability of our results, the reported values are the mean and standard deviation derived from 1,000 independent runs. Chaos-Nexus exhibits an inference latency approximately 0.017s higher than Panda per forecast. This moderate increase is an expected trade-off adhering to the "no free lunch" principle, attributable to our hierarchical architecture of ScaleFormer, MoE routing, and frequency-domain modeling. Given that the faithful reproduction of complex chaotic dynamics is paramount and this latency remains well within practical limits for this task, we consider the computational cost well-justified by the substantial performance gains. Regarding general-purpose baselines, their inference speeds are largely dictated by specific architectural configurations, such as patch granularity and architectural complexity. For instance, Timer-XL achieves high efficiency through large-patch processing (e.g., patch size of 96), whereas Moirai-MoE incurs significant overhead due to its smaller patch size, intricate expert routing and gating clustering mechanisms. However, we emphasize that lower latency cannot compensate for poor generalization. Since these baselines fail to capture chaotic dynamics effectively, their speed advantage offers no practical utility.

### B.6. Forecasting Performance on PDE Systems

We evaluate ChaosNexus on two PDE datasets with increasing spatial complexity: first the 1D Kuramoto–Sivashinsky (KS) dynamics, and then the 2D Von Kármán Vortex Street

*Table 4.* Progressive ablation of PME, MoE, and MMD components. All metrics are lower-better and reported as mean $\pm$ 95% CI.

| Variant | sMAPE@128 | sMAPE@512 | $D_{\text{frac}}$ | $\text{ME}_{\text{LRw}}$ | $D_{\text{Lyap}}$ | $D_{\text{stsp}}$ |
|---|---|---|---|---|---|---|
| + PME | $75.786 \pm 3.181$ | $105.127 \pm 3.010$ | $0.222 \pm 0.012$ | $2.187 \pm 0.172$ | $0.097 \pm 0.011$ | $1.561 \pm 0.597$ |
| + MMD | $87.672 \pm 2.931$ | $113.881 \pm 2.304$ | $0.231 \pm 0.013$ | $2.421 \pm 0.179$ | $0.099 \pm 0.010$ | $1.752 \pm 0.668$ |
| + PME & MMD | $75.332 \pm 3.088$ | $104.867 \pm 2.689$ | $0.220 \pm 0.011$ | $2.092 \pm 0.219$ | $0.095 \pm 0.014$ | $1.490 \pm 1.297$ |
| + PME & MoE | $75.028 \pm 3.151$ | $104.400 \pm 2.979$ | $0.219 \pm 0.012$ | $2.055 \pm 0.167$ | $0.093 \pm 0.045$ | $1.421 \pm 0.653$ |
| **+ PME & MoE & MMD** | $\mathbf{72.699 \pm 3.179}$ | $\mathbf{104.002 \pm 2.930}$ | $\mathbf{0.215 \pm 0.010}$ | $\mathbf{1.771 \pm 0.132}$ | $\mathbf{0.082 \pm 0.013}$ | $\mathbf{1.360 \pm 0.440}$ |

*Table 5.* Inference time comparison of foundation models when forecasting 512 time steps. Reported values represent the mean $\pm$ standard deviation over 1000 runs.

| Model | Time (s) |
|---|---|
| ChaosNexus | $0.119 \pm 0.036$ |
| Panda | $0.048 \pm 0.004$ |
| Chronos-S | $0.081 \pm 0.022$ |
| Chronos-B | $0.095 \pm 0.012$ |
| Chronos-L | $0.173 \pm 0.022$ |
| Moirai-MoE-S | $1.677 \pm 0.377$ |
| Moirai-MoE-L | $3.124 \pm 0.201$ |
| TimeMoE-S | $0.038 \pm 0.019$ |
| TimeMoE-L | $0.042 \pm 0.020$ |
| TimesFM | $0.143 \pm 0.026$ |
| Timer-XL | $0.005 \pm 0.002$ |

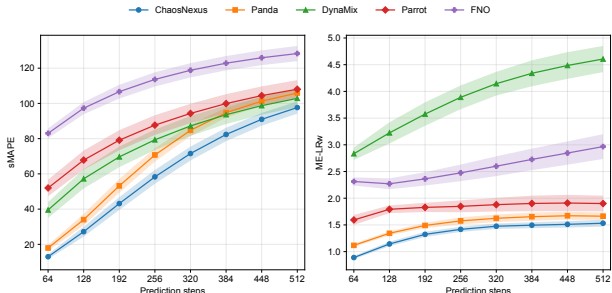

*Figure 10.* Forecasting performance on the 1D Kuramoto–Sivashinsky PDE dataset across prediction horizons. Lines depict the average value, with shaded regions representing the 95% CI.

(VKVS) dynamics.

### B.6.1. 1D KURAMOTO–SIVASHINSKY DYNAMICS

**Simulation Setup.** We consider the Kuramoto–Sivashinsky equation, a canonical 1D spatiotemporal chaotic PDE. We set the spatial domain length to $L = 100$ and solve the system with a pseudo-spectral solver, retaining 1024 nonzero Fourier modes spanning both positive and negative frequencies. Together with the zero-frequency mode involved in the inverse transform, this yields a 1025-dimensional state space. We integrate the system from $t = 0$ to $t = 100$ with relative and absolute tolerances of $10^{-8}$, and save 4096 uniformly spaced snapshots for evaluation.

**Results.** We perform 512-step zero-shot forecasting with 512 context steps, and compare ChaosNexus with Panda, DynaMix, Parrot, and an FNO baseline trained on the same contexts. The full metric curves in Figure 10 show that ChaosNexus achieves the best performance across rollout lengths, including both point-wise forecasting accuracy and spectral energy fidelity. Figure 11 further provides representative qualitative forecasts at 512 steps.

### B.6.2. 2D VON KÁRMÁN VORTEX STREET DYNAMICS

**Simulation Setup.** We consider the 2D Navier-Stokes equations modeled via the Lattice Boltzmann Method (LBM) using a standard D2Q9 topology. The simulation is configured to generate Von Kármán Vortex Street (VKVS) dynamics past a cylindrical obstacle. The simulation domain is a rectangular channel with dimensions $420 \times 180$ lattice units.

A cylindrical obstacle with radius $r = 20$ is positioned at $(x, y) = (105, 90)$ to induce flow separation. We impose a parabolic velocity profile at the inlet with a maximum characteristic velocity $u_{LB} = 0.04$, and a standard bounce-back condition on the obstacle surface. The viscosity is adjusted to achieve a Reynolds number ($Re$) of 450, placing the system in a regime characterized by unsteady, periodic vortex shedding and chaotic turbulence in the wake.

**Data Collection.** To ensure the flow reaches a statistically stationary state, we discard the initial $90,000$ simulation steps as a burn-in period. Subsequently, we collect a dataset of $T = 4096$ frames, sampled at a temporal interval of $\Delta t = 250$ LBM steps.

**Preprocessing.** Instead of raw velocity fields, we focus on the vorticity dynamics ($\omega = \partial_x v_y - \partial_y v_x$), computed via central differences, as it better highlights the coherent structures of the fluid. The spatial domain is cropped to remove the laminar inlet region (removing the first 40 columns), resulting in an effective resolution of $380 \times 180$. To enable efficient forecasting, we project the high-dimensional vorticity fields into a low-dimensional latent space using Principal Component Analysis (PCA), retaining the top $d = 16$ principal components.

**Results.** We compare the zero-shot forecasting performance of ChaosNexus with other foundation models on ODE-based chaotic dynamics, including Panda, Parrot (Zhang & Gilpin, 2025b), and DynaMix (Hemmer & Durstewitz, 2025). While the forecasting processes operate within a low-dimensional PCA latent space, we apply the in-

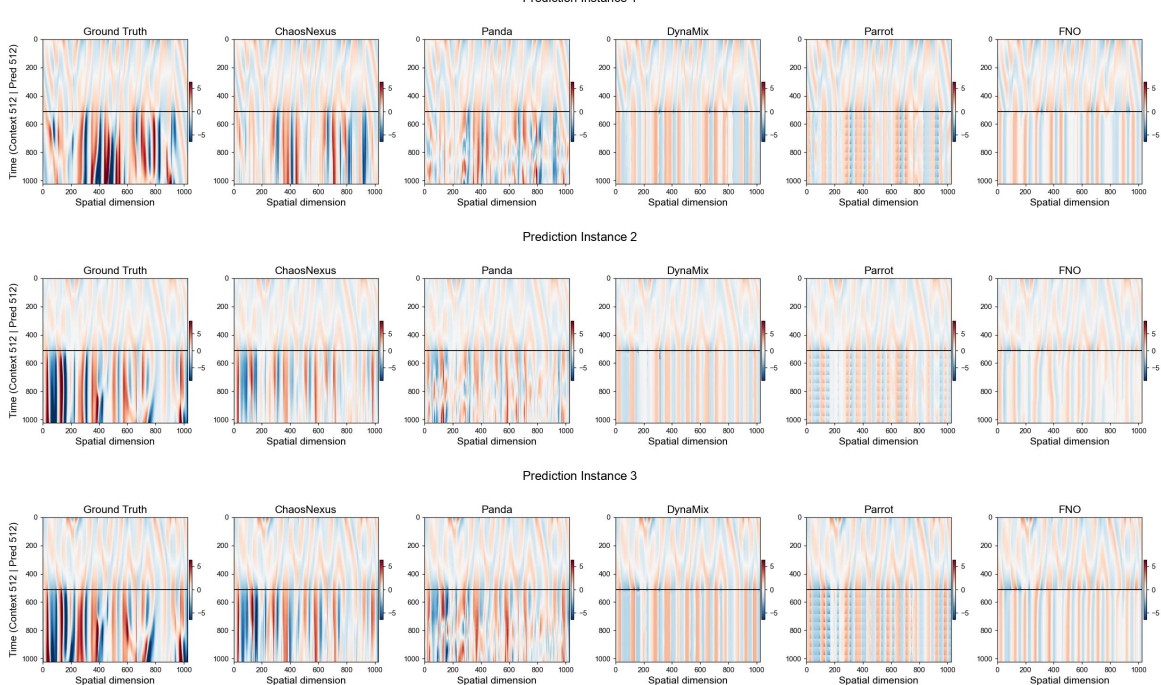

*Figure 11.* Qualitative 512-step forecasting examples on the 1D Kuramoto–Sivashinsky PDE dataset.

verse transformation to map predictions back to the original observation space for metric evaluation. The context length is 512 steps, and the forecasting horizons are $\{64, 128, 192, 256, 320, 384, 448, 512\}$ steps. We also demonstrate illustrative forecasting samples in Figure 12, and report sMAPE metrics in Figure 13. We find that ChaosNexus achieves superior forecasting performance on this PDE system, despite being trained solely on ODEs. PCA projects spatiotemporal dynamics onto a latent manifold that resembles our ODE training corpus. Crucially, our ScaleFormer architecture excels at modeling the resulting multi-scale temporal dynamics, effectively capturing both the dominant periodic vortex shedding and the fine-grained chaotic fluctuations in the turbulent wake.

### B.7. Additional Results on Multi-scale Feature Analysis

We demonstrate temporal attention maps for each encoder and decoder level of ScaleFormer in Figure 14.

### B.8. Expert Activation Visualization

We visualize the expert activation patterns within the encoder and decoder for selected test systems in Figure 15. We find that systems derived from the same foundation dynamics (Appendix F.1) trigger analogous routing profiles across all layers and scales. This provides direct evidence that the MoE framework has learned to partition the problem space, systematically assigning inputs to specialized

experts based on their dynamical properties to effectively process and differentiate between complex systems. We also provide quantitative results in Appendix B.9 to further support our findings.

### B.9. Analysis of MoE Routing Specialization and Robustness

#### B.9.1. EXPERT ACTIVATION CLUSTERING

To investigate the underlying specialization mechanisms within the Mixture of Experts (MoE) architecture, we analyze the gating activation patterns, *i.e.,* expert selection probabilities, across different depths of the network. Specifically, we aggregate the expert activation probabilities of context trajectories from three canonical chaotic dynamical systems, including Lorenz63, Rossler, and Lorenz96 systems, to determine whether the router implicitly learns to distinguish systems based on their governing physical laws.

Figure 16 demonstrates t-SNE projections of the high-dimensional gating distributions from various Encoder and Decoder MoE layers (Depths 1 through 4). To quantify the degree of system-specific specialization in the routing mechanism, we calculate the Adjusted Rand Index (ARI) for each projection, which measures the similarity between the obtained clustering and the ground-truth labels. A score of 1.0 signifies perfect alignment where experts are exclusively specialized for specific systems, whereas a score near 0.0 indicates random assignment.

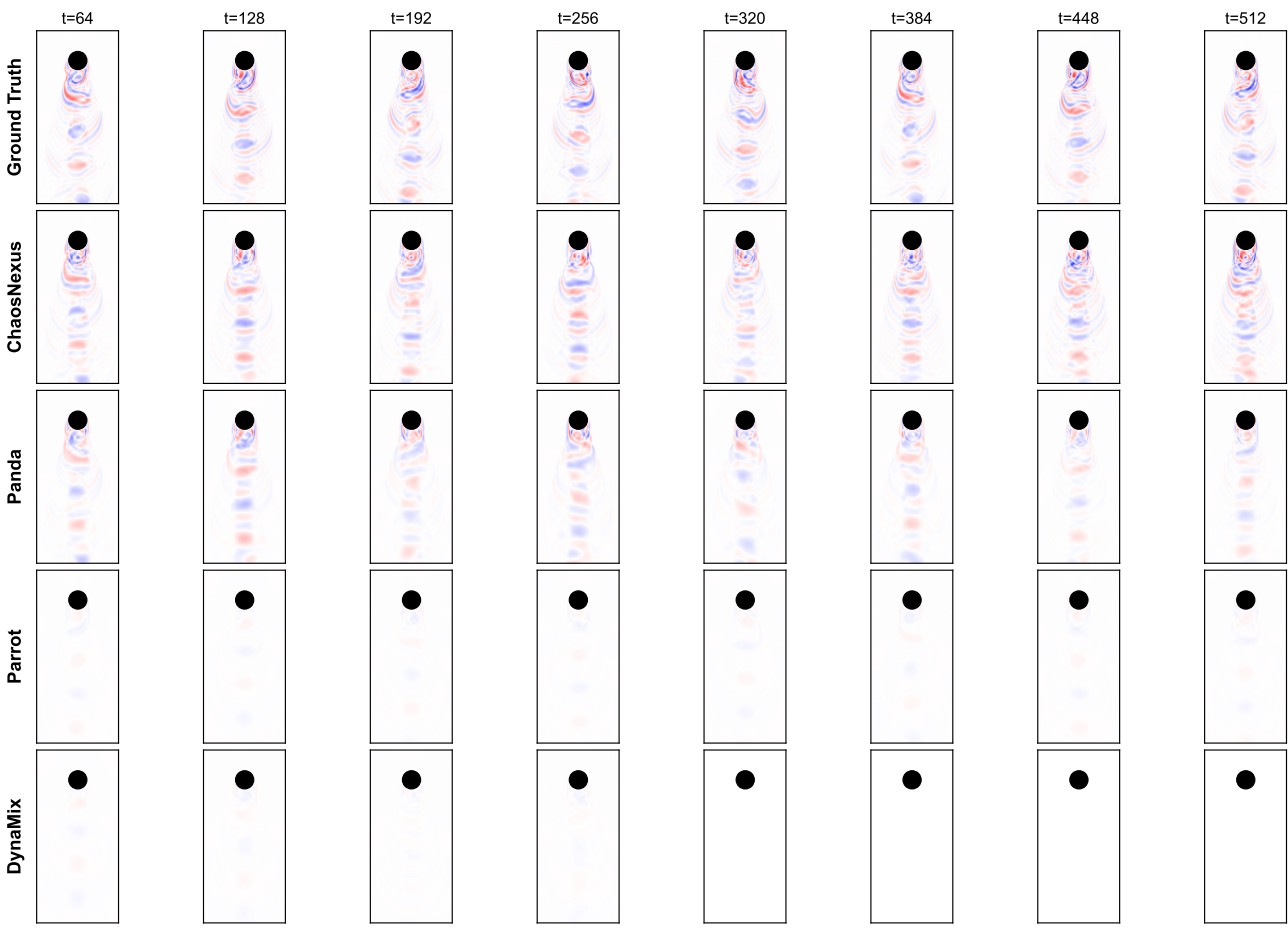

*Figure 12.* Qualitative 512-step forecasting examples on the 2D Von Kármán Vortex Street PDE dataset.

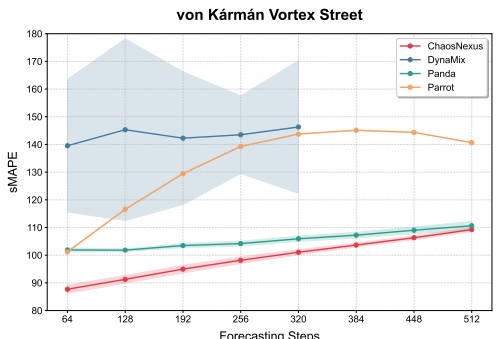

*Figure 13.* Forecasting performance on Von Kármán Vortex Street (VKVS) dynamics. Lines depict the average value, with shaded regions representing the 95% CI. DynaMix produces NaN values from 320 forecasting steps; therefore, its performance after longer horizons cannot be reported.

The visualization reveals that the router's gating decisions are highly structured and system-dependent. In the vast majority of MoE layers, the expert activation patterns form distinct clusters that correspond precisely to the Lorenz63, Rossler, and Lorenz96 systems. This observation is substantiated by the quantitative metrics, where the ARI scores consistently remain high—exceeding 0.5 in most layers and peaking at 0.9933 in the encoder. These results statistically confirm that the experts exhibit strong system-level specialization, implying that the router implicitly learns to distinguish and dispatch data based on the distinct underlying physical mechanisms of each dynamical system.

### B.9.2. ENTROPY OF GATING DISTRIBUTION

Figure 17 depicts the layer-wise evolution of the gating entropy of three canonical systems, including Lorenz63, Rossler, and Lorenz96. Scatter points represent the entropy of the gating distribution from a specific sample, and box plots encapsulate the aggregate statistical dispersion, i.e., the median and interquartile range. The results are summarized as follows:

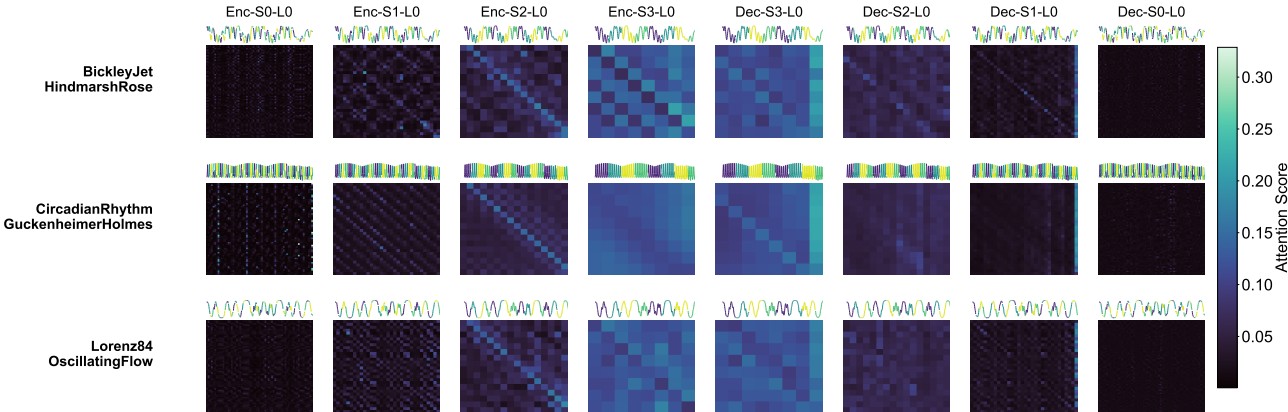

*Figure 14.* Visualization of input patch partitioning and multi-scale temporal attention for three chaotic systems. S and L denote the scale level and the block within each level, respectively.

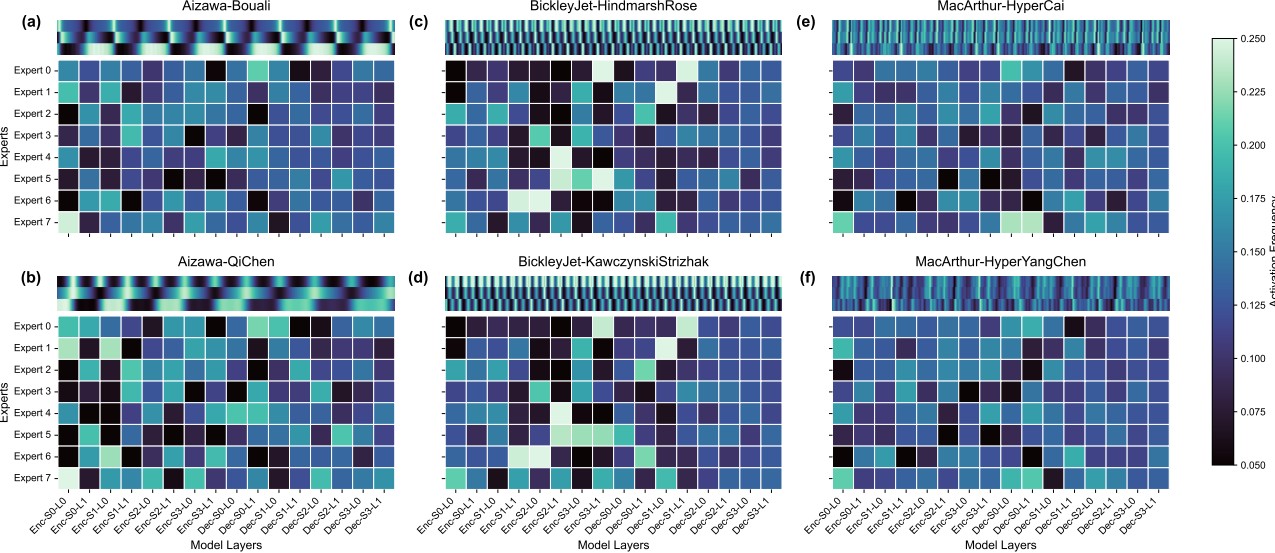

*Figure 15.* Expert activation visualization for six discovered chaotic systems by the evolutionary framework from three common foundation chaotic systems. S and L denote the scale level and the block within each level, respectively.

- **Shallow Encoder.** In the initial encoder layers (Enc-D1 to Enc-D3), the gating distribution exhibits consistently high entropy. This indicates that the router utilizes a diverse mixture of experts to process raw input patches.

- **Bottleneck.** Entropy decreases substantially as the information propagates to the network bottleneck (Enc-D4 and Dec-D4). Here, the entropy minimizes, signifying a regime of high specialization. The model abstracts the input into core dynamical representations, and the router demonstrates high confidence, assigning specific expert modules to handle distinct underlying patterns. This drop in entropy confirms that the model has successfully disentangled the latent semantics, prioritizing specific experts for specific dynamical behaviors.

- **Shallow Decoder.** In the final decoding stages, entropy rises back to higher levels, which implies collaborative synthesis. To reconstruct accurate continuous trajectories from abstract representations, the decoder must integrate the semantic guidance from both the bottleneck and the high-frequency details retrieved via skip connections. The router therefore employs an ensembling strategy, aggregating outputs from multiple experts to ensure robust, smooth, and precise signal reconstruction.

- **Discussion on Load Balancing Loss.** The results demonstrate that the router establishes a dynamic equilibrium: it yields to the regularization pressure in the shallow layers to maintain generalizability, but prioritizes semantic specialization in the deep layers where distinguishing physical mechanisms is critical. Thus, the load balancing loss serves as a flexible regularizer, preventing mode col-

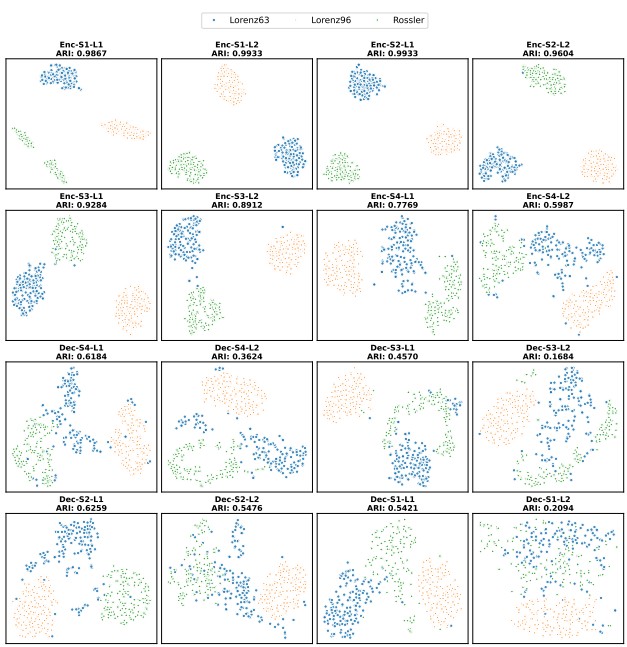

*Figure 16.* Layer-wise expert activation clustered by system type.

lapse without suppressing the necessary concentration of attention required to model complex chaotic dynamics.

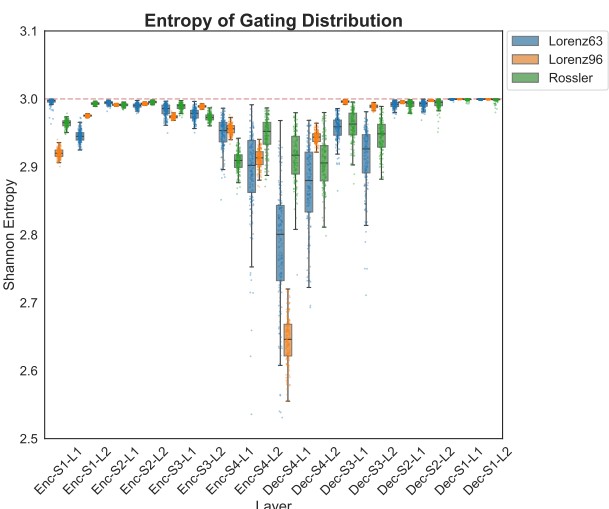

*Figure 17.* Layer-wise gating distribution entropy for canonical systems.

### B.9.3. EXPERT PRUNING IMPACT

To validate the distinct functional specialization within our Mixture-of-Experts architecture, we conduct an expert pruning experiment on three canonical chaotic systems, including Lorenz63, Rossler, and Lorenz96. Specifically, we identify the top-2 most frequently activated experts for each system per layer and deactivate them during the inference phase. As evidenced by the results in Table 6, this targeted

*Table 6.* Expert pruning impact on three canonical chaotic systems. Each reported value indicates the mean ± 95% CI.

| Experiment | sMAPE@128 | sMAPE@512 | $D_{\mathrm{frac}}$ | $D_{\mathrm{stsp}}$ |
|---|---|---|---|---|
| **Lorenz63 w/o Pruning** | **62.105 ± 0.964** | 115.545 ± 0.651 | 0.132 ± 0.003 | 0.204 ± 0.019 |
| Lorenz63 w/ Pruning | 79.698 ± 1.002 | 123.342 ± 0.592 | 0.147 ± 0.003 | 0.247 ± 0.019 |
| **Lorenz96 w/o Pruning** | **154.140 ± 0.091** | **157.518 ± 0.070** | **6.022 ± 0.014** | **20.554 ± 0.049** |
| Lorenz96 w/ Pruning | 154.160 ± 0.092 | 157.577 ± 0.070 | 6.159 ± 0.014 | 20.627 ± 0.049 |
| **Rossler w/o Pruning** | **30.458 ± 0.525** | **55.677 ± 0.590** | **0.159 ± 0.005** | **0.074 ± 0.003** |
| Rossler w/Pruning | 37.818 ± 0.579 | 64.831 ± 0.604 | 0.160 ± 0.005 | 0.102 ± 0.004 |

*Table 7.* sMAPE comparison between models with and without MoE across system-heterogeneity partitions. Δ denotes the absolute error reduction from adding MoE.

| Split | sMAPE@128 w/ MoE | sMAPE@128 w/o MoE | Δ ↓ | sMAPE@512 w/ MoE | sMAPE@512 w/o MoE | Δ ↓ |
|---|---|---|---|---|---|---|
| Easy | **41.492 ± 2.169** | 46.029 ± 2.069 | 4.537 | **76.743 ± 2.889** | 81.682 ± 2.723 | 4.939 |
| Medium | **68.325 ± 2.572** | 75.184 ± 2.526 | 6.859 | **100.951 ± 2.713** | 106.988 ± 2.560 | 6.037 |
| Hard | **99.377 ± 3.362** | 106.153 ± 3.283 | 6.776 | **122.965 ± 2.966** | 128.899 ± 2.607 | 5.934 |
| Easy → Hard | +57.885 | +60.124 | – | +46.222 | +47.217 | – |

pruning leads to a consistent degradation across both point-wise forecasting accuracy (sMAPE) and long-term attractor fidelity metrics ($D_{\mathrm{frac}}$ and $D_{\mathrm{stsp}}$). This performance drop substantiates that the model relies on specific, specialized experts to capture distinct dynamical regimes, rather than utilizing a generalized ensemble for all inputs.

### B.9.4. PARTITIONED ANALYSIS UNDER SYSTEM HETEROGENEITY

To further examine whether MoE helps handle heterogeneous dynamics, we partition the evaluation systems according to their distance from the training systems. Since the MoE layers operate on latent representations encoded from observed trajectories, we measure heterogeneity using Dynamic Time Warping (DTW) distance (Berndt & Clifford, 1994), which compares trajectory evolution patterns more directly than pointwise distance. For each evaluation system, we compute its nearest-neighbor DTW distance to the training systems and divide the evaluation set into Easy, Medium, and Hard groups from low to high distance.

Table 7 shows that MoE consistently reduces sMAPE across all three partitions. The gains remain clear in the Hard group, where the evaluation systems are farthest from the training systems, indicating that MoE is not merely improving average performance on easy cases. Moreover, the Easy-to-Hard degradation is smaller with MoE than without MoE for both 128-step and 512-step forecasts, suggesting that input-adaptive expert routing makes the model more robust as system heterogeneity increases.

### B.10. Detailed Analysis on Frequency Fingerprint

**Spectral representation design.** We explore using the STFT and learnable Fourier features as alternative designs for the system fingerprint. Specifically, to implement STFT, we replace the WST module with an STFT encoding, flattening the time-frequency features into the same dimension as our fingerprint. For the learnable fingerprint, we replace

*Table 8.* Comparison between alternative spectral representations.

| Experiment | sMAPE@128 | sMAPE@512 | $D_{\text{frac}}$ | $D_{\text{stsp}}$ |
|---|---|---|---|---|
| **WST (Ours)** | **68.901 ± 3.086** | **100.293 ± 2.767** | 0.203 ± 0.011 | **1.206 ± 0.392** |
| STFT | 77.096 ± 11.502 | 102.205 ± 11.247 | **0.201 ± 0.056** | 1.370 ± 1.240 |
| Learnable | 83.550 ± 11.122 | 107.300 ± 9.950 | 0.215 ± 0.057 | 2.032 ± 1.287 |

*Table 9.* Ablation of frequency fingerprint conditioning and readout designs. All metrics are lower-better and reported as mean ± 95% CI.

| Variant | sMAPE@128 | sMAPE@512 | $ME_{\text{LRw}}$ | $D_{\text{frac}}$ |
|---|---|---|---|---|
| **Late conditioning / linear readout** | **68.901 ± 3.085** | **100.293 ± 2.766** | 1.562 ± 0.115 | **0.203 ± 0.011** |
| MoE routing modulated by fingerprint | 78.774 ± 3.095 | 103.949 ± 2.442 | 2.106 ± 0.255 | 0.226 ± 0.014 |
| w/o frequency fingerprint | 72.699 ± 3.179 | 104.002 ± 2.930 | 1.771 ± 0.132 | 0.215 ± 0.010 |
| FiLM-style readout | 71.951 ± 3.078 | 101.343 ± 2.399 | 1.694 ± 0.235 | 0.206 ± 0.013 |
| MLP readout | 72.177 ± 3.103 | 101.489 ± 2.548 | 1.721 ± 0.249 | 0.208 ± 0.016 |

the fixed wavelet filters with learnable spectral filters (1D convolutional layer) followed by the same pooling operations, allowing the model to adaptively learn frequency representations. The results are shown in Table 8. From the results, we have the following conclusions:

- **First**, the WST achieves significantly lower point-wise errors and better attractor reconstruction compared to STFT. We attribute this to the fact that chaotic systems exhibit dynamics across a continuum of scales. WST naturally captures multi-scale interactions through its hierarchical cascade, making it more robust for diverse chaotic dynamics. In contrast, STFT suffers from the fixed window size limitation.

- **Second**, Learnable variant performs the worst. Given the vast diversity of our training corpus, learning a single set of spectral filters that generalizes universally is highly difficult. The WST provides a strong inductive bias with its mathematical properties of translation invariance and stability to deformations, offering a stable fingerprint that requires no training, thus enhancing zero-shot generalization.

**Conditioning and readout design.** We further examine whether the frequency fingerprint should modulate intermediate MoE routing or be used only at the final readout. The fingerprint is temporally pooled and therefore summarizes the global spectral identity of the observed system; it does not encode patch-specific information that would help each local router make token-level decisions. Thus, broadcasting the same vector into every MoE gate can create a granularity mismatch, while late conditioning uses the fingerprint as a regime-level calibrator after the ScaleFormer backbone has already extracted multi-scale temporal representations.

Table 9 supports this design choice. Modulating MoE routing with the fingerprint performs worse than the original late-conditioning design, indicating that injecting the global descriptor into patch-level gates can weaken temporal representation learning. Removing the fingerprint also degrades

*Table 10.* Sensitivity analysis to the weighting coefficient $\lambda_2$ of MMD regularization.

| $\lambda_2$ | sMAPE@128 | sMAPE@512 | $D_{\text{frac}}$ | $D_{\text{stsp}}$ |
|---|---|---|---|---|
| 0.01 | 80.093 ± 3.213 | 109.596 ± 2.809 | 0.231 ± 0.012 | 1.331 ± 0.381 |
| 0.05 | 80.139 ± 3.169 | 107.743 ± 2.744 | 0.216 ± 0.012 | 1.434 ± 0.435 |
| 0.10 | 79.107 ± 3.112 | 105.665 ± 2.731 | 0.210 ± 0.012 | 1.287 ± 0.400 |
| **0.50** | **68.901 ± 3.086** | **100.293 ± 2.767** | **0.203 ± 0.011** | **1.206 ± 0.392** |
| 1.00 | 78.474 ± 2.923 | 102.550 ± 2.412 | 0.208 ± 0.011 | 1.329 ± 0.395 |
| 5.00 | 80.928 ± 2.760 | 103.572 ± 2.320 | 0.210 ± 0.012 | 1.385 ± 0.309 |
| 10.00 | 81.280 ± 2.724 | 103.668 ± 2.319 | 0.209 ± 0.012 | 1.318 ± 0.333 |

all reported metrics, confirming that the spectral descriptor contributes useful regime information. Finally, replacing the linear readout with FiLM-style or MLP readouts does not improve performance, suggesting that the nonlinear ScaleFormer representation and wavelet fingerprint already provide sufficient features for a simple conditional decoder.

### B.11. Impact of MMD Regularization

#### B.11.1. SENSITIVITY TO THE WEIGHTING COEFFICIENT

We set $\lambda_2 = 0.5$ in our experiments. Here we demonstrate the sensitivity to the weighting coefficient $\lambda_2$. Specifically, we choose $\lambda_2$ at different scales: $\{0.01, 0.05, 0.1, 0.5, 1, 5, 10\}$. Table 10 presents the results. From the results, we draw the following conclusions:

- **First**, our observations indicate that $\lambda_2 = 0.5$ represents a robust optimum, effectively balancing the point-wise accuracy required for short-term forecasting with the distributional fidelity needed for long-term stability.

- **Second**, when $\lambda_2$ is small (0.01-0.1), we observe a marked degradation in both point-wise accuracy and attractor fidelity. This confirms that explicitly enforcing attractor geometry aids the model in learning the underlying dynamics. Pure MSE minimization is insufficient for chaotic systems as it lacks the global constraints to prevent divergence.

- **Third**, excessively large weights ($\lambda_2 \geq 5.0$) lead to a performance drop on point-wise accuracy, as the distributional constraint begins to dominate the loss landscape, impeding the model's ability to minimize local prediction errors.

#### B.11.2. SENSITIVITY TO KERNEL FUNCTION

We conduct additional experiments to compare our default mixture of rational quadratic kernels against three alternatives: mixture of Gaussian kernel, a linear kernel, and a polynomial kernel, which take the following forms:

- **Mixture of Gaussian kernel**. To ensure a fair comparison with the multi-scale nature of our default mixture of rational quadratic kernel, we implement a mixture

*Table 11.* Sensitivity analysis of the kernel function selection of MMD regularization.

| Kernel | sMAPE@128 | sMAPE@512 | $D_{\text{frac}}$ | $D_{\text{stsp}}$ |
|---|---|---|---|---|
| **Mixture of rational quadratic kernel** | **68.901 ± 3.086** | **100.293 ± 2.767** | **0.203 ± 0.011** | **1.206 ± 0.392** |
| Mixture of Gaussian kernel | 80.329 ± 3.198 | 109.577 ± 2.780 | 0.227 ± 0.012 | 1.431 ± 0.515 |
| Linear kernel | 82.293 ± 3.145 | 109.282 ± 2.750 | 0.217 ± 0.012 | 1.276 ± 0.313 |
| Polynomial kernel | 83.126 ± 3.033 | 107.908 ± 2.533 | 0.215 ± 0.011 | 1.309 ± 0.366 |

of Gaussian kernels over the same set of length scales $\boldsymbol{\sigma} = \{0.2, 0.5, 0.9, 1.3\}$,

$$\kappa(\boldsymbol{u}, \boldsymbol{v}) = \sum_{\sigma \in \boldsymbol{\sigma}} \exp - \frac{||\boldsymbol{u} - \boldsymbol{v}||_2^2}{2\sigma^2}. \qquad (16)$$

- **Linear kernel**. The linear kernel captures similarity through a direct dot product in the input space, implying a linear relationship between the governing features of the attractors:

$$\kappa(\boldsymbol{u}, \boldsymbol{v}) = \boldsymbol{u}^T \boldsymbol{v}. \qquad (17)$$

- **Polynomial kernel**. The polynomial kernel projects the inputs into a higher-dimensional feature space determined by the degree $d$ and a bias term $c$:

$$\kappa(\boldsymbol{u}, \boldsymbol{v}) = (\boldsymbol{u}^T \boldsymbol{v} + c)^d, \qquad (18)$$

where we set $d = 2$ and $c = 1$.

The experimental results are shown in Table 11. We find that the mixture of rational quadratic kernels consistently yields superior performance across both short-term forecasting (sMAPE) and long-term attractor reconstruction. It outperforms the Gaussian, linear, and polynomial kernels by a wide margin in both point-wise accuracy and attractor reconstruction fidelity. This aligns with the theoretical motivation in Appendix C.4, a rational quadratic kernel can be viewed as an infinite mixture of Gaussian kernels with varying length scales (Rasmussen & Williams, 2006). This property is crucial for capturing the multi-scale temporal and spectral structures inherent in chaotic systems, which single-scale Gaussian kernels fail to represent adequately.

### B.11.3. VISUALIZATION EXAMPLES

We further provide illustrative forecasting cases that isolate the contribution of the MMD-based auxiliary loss. The results are shown in Figure 18. As shown, the removal of the distributional constraint causes the predicted trajectories to drift significantly from the underlying manifold, failing to reproduce the complex geometry of the strange attractor. In contrast, the MMD-regularized model effectively preserves the attractor structure, ensuring that the forecasted dynamics faithfully align with the ground-truth.

### B.12. Forecast Showcases

We demonstrate forecasting showcases of six representative systems in Figure 19.

## B.13. Performance Collapse of System-Specific Models in Zero-Shot Forecasting

To demonstrate the necessity of designing and training a foundation model for zero-shot chaotic system forecasting, we conduct a controlled experiment where a system-specific model, FEDFormer (Zhou et al., 2022), is trained on the exact training corpus as ChaosNexus. After the training process, we test the model on the canonical Lorenz63 system and demonstrate the results in Figure 20. We find that FEDFormer fails to capture the underlying chaotic dynamics given the context. The phenomenon indicates that without the specific design choices in ChaosNexus, system-specific models suffer from severe underfitting when exposed to highly heterogeneous dynamical systems, rendering them ineffective for zero-shot generalization.

## B.14. Additional Results on Weather Benchmark

### B.14.1. DETAILED RESULTS

We demonstrate the detailed forecasting results for all weather variables, including the temperature, dew point, sea level pressure, wind direction, and wind speed in Figure 21-25, respectively. Clearer results for ChaosNexus, Panda, and Chronos-S-SFT, which are trained on the corpus of synthetic chaotic systems, are shown in Figure 26-30. This strong performance paradigm is consistently replicated across the remaining meteorological variables. In the zero-shot setting, ChaosNexus substantially outperforms all baseline models, even when they are fine-tuned on up to 473K samples from the target weather system. The model's forecasting accuracy is further enhanced with few-shot fine-tuning, demonstrating remarkable data efficiency. This advantage is particularly pronounced at longer prediction horizons, highlighting the robustness of the representations learned during pre-training. Collectively, these results validate our central hypothesis: pre-training on a diverse corpus of chaotic systems endows the model with a universal understanding of complex dynamics. This allows ChaosNexus to achieve state-of-the-art performance on real-world forecasting tasks with minimal, or even zero, in-domain fine-tuning, thereby overcoming the critical challenge of data sparsity in scientific applications.

Besides comparison with system-specific models in Figure 6 of the main text, we also benchmark the forecasting performance of other foundation models on this dataset. We find that foundation models designed for chaotic system forecasting or trained on our corpus of synthetic chaotic dynamics, including ChaosNexus, Panda, and Chronos-S-SFT, perform significantly better than those trained on general time series, even though they use a much larger corpus (see Table 13). It demonstrates that pretraining specifically on chaotic systems provides a more relevant inductive bias for weather forecasting. Moreover, ChaosNexus also outperforms Panda on many variable forecasting tasks, highlighting the contri-

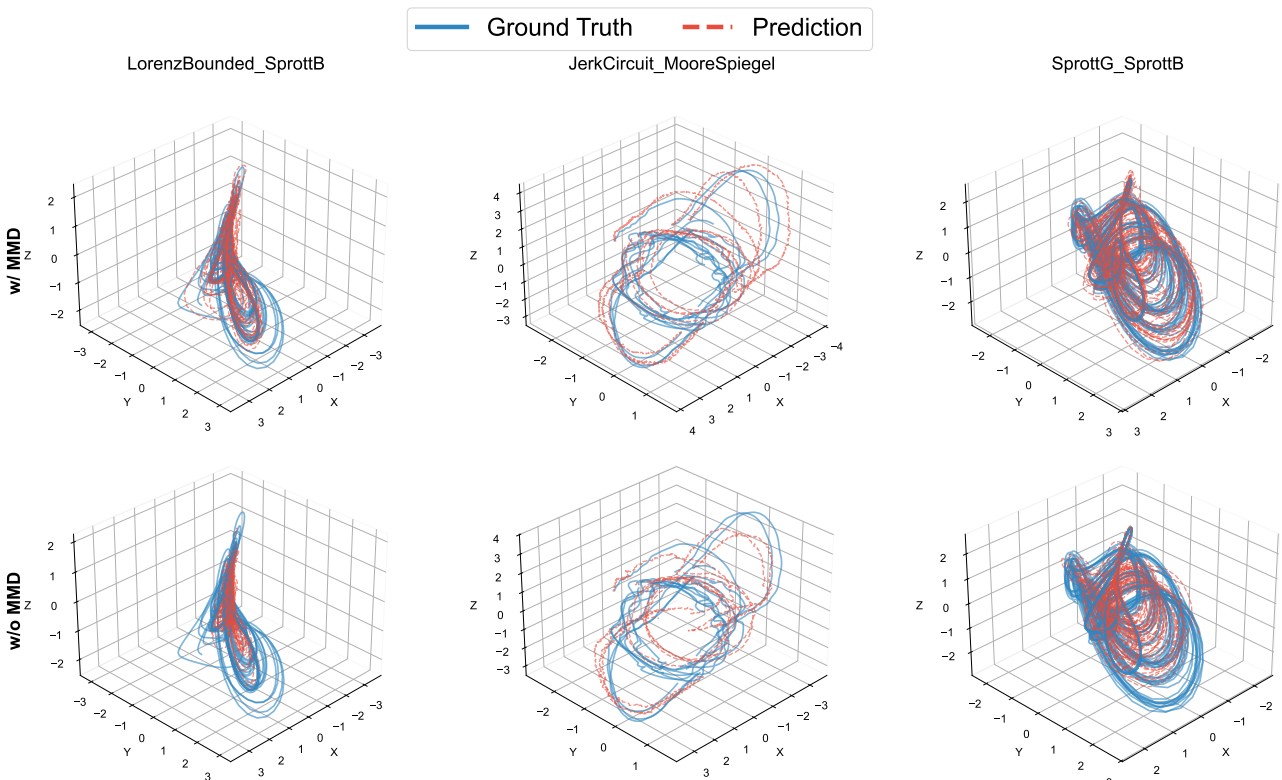

*Figure 18.* Visualization of the impact of MMD regularization on long-term forecasting.

bution of our multi-scale architectural designs.

### B.14.2. TEMPERATURE FORECASTING PERFORMANCE ACROSS LATITUDES

We conduct additional analysis and stratify weather stations into three latitude bands: low latitudes (30°N–30°S), mid-latitudes (30°N–60°N, 30°S–60°S), and high latitudes (60°N–90°N, 60°S–90°S). There are 1093, 4000, and 579 stations in low-latitude, mid-latitude, and high-latitude bands, respectively. For each band, we report the MAE on the 5-day temperature forecasting of our model and all baselines. The results are shown in Figure 31-33.

From the results, we can draw the following conclusions:

- **First**, ChaosNexus maintains a zero-shot MAE strictly below 1°C across all latitude bands at the 5-day (120h) horizon. Furthermore, fine-tuning yields consistent performance gains across all stations, for instance, in high-latitude regions, the 120h MAE decreases from 0.8124 to 0.6659 (an ∼ 18% improvement). This confirms that our foundation model serves as a robust universal prior capable of rapid adaptation to local climatic conditions.

- **Second**, the error distribution accurately reflects the inherent complexity of atmospheric dynamics. Zero-shot error is minimized in the tropics (MAE ≈ 0.59) due to

lower variability, and increases slightly in mid-to-high latitudes (MAE ≈ 0.74–0.81), regions characterized by chaotic frontal systems and baroclinic instability. Despite these challenges, the error remains tightly bounded.

- **Third**, ChaosNexus consistently outperforms all baselines across every latitude band. It surpasses strong system-specific baselines (e.g., Crossformer, PatchTST) by a substantial margin, avoiding catastrophic errors exceeding 3°C, and reliably outperforms the competing foundation model, Panda, in zero-shot settings. These results establish ChaosNexus as the state-of-the-art solution for chaotic forecasting.

### B.14.3. EXPERT ACTIVATION CONTRAST WITH RANDOM SAMPLED SYSTEMS

To further validate that the observed alignment between real-world weather data and synthetic atmospheric proxies is physically meaningful, we visualize the expert activation patterns of a randomly selected, unrelated chaotic system from the test set in Figure 34. Unlike the consistent activation signatures shared by the weather data and its canonical prototypes, the random system elicits a markedly distinct gating distribution across both encoder and decoder layers. This contrast provides counterfactual evidence that ChaosNexus effectively routes inputs based on intrinsic physical

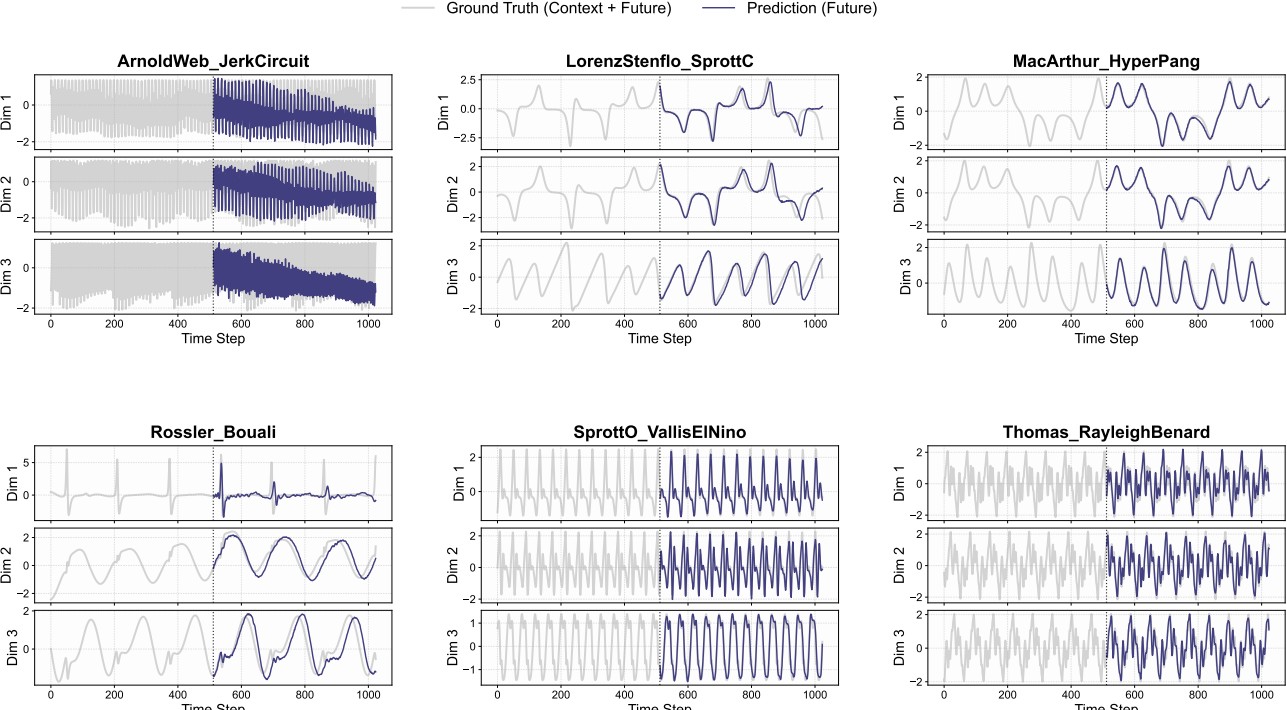

*Figure 19.* Forecasting showcases of representative chaotic systems.

properties, activating atmospheric experts only when the input data exhibits the corresponding convective or oscillatory dynamics. This selectivity underscores that the model's generalization capabilities stem from a precise matching of dynamical regimes.

## C. Implementation Details

### C.1. Input Augmentation Features

As stated in the main text, our approach to feature engineering is inspired by Koopman operator theory (Koopman, 1931), which suggests that a complex nonlinear dynamical system can be represented as a linear system in an infinite-dimensional space of observable functions. While this infinite-dimensional space is practically inaccessible, it can be effectively approximated by projecting the system's state into a higher-dimensional feature space. This process of lifting the dynamics is a cornerstone of methods like Extended Dynamic Mode Decomposition (eDMD) (Williams et al., 2015).

Following this principle, and adopting a technique from recent work on pretrained forecast models, we enrich the representation of each time series patch before it is processed by the main architecture. Instead of using the raw patch data alone, we construct an augmented feature vector by concatenating the original patch with two additional sets of randomly generated, nonlinear features.

- **Random Polynomial Features.** To capture nonlinear relationships within each patch, we generate a set of monomial features. For a given polynomial degree, $d$, this is achieved by first sampling a collection of $d$-tuples of indices. For each tuple, we compute a new feature by multiplying the patch elements corresponding to those indices. This creates a basis of polynomial observables that can approximate the underlying dynamics. For our model, we use polynomial features of degree $d \in \{2, 3\}$.

- **Random Fourier Features.** To approximate a universal kernel and capture periodic patterns, we employ random Fourier features, a widely-used technique for scaling up kernel methods. We implement this by projecting a patch onto a set of random vectors, whose components are sampled from a normal distribution. The resulting scalar values are then transformed using both sine and cosine functions, effectively creating a randomized spectral basis.

The final embedding for each patch is formed by concatenating the original patch vector with the generated polynomial and Fourier features. This lifted representation provides a much richer input to the model, allowing it to more easily learn and represent the complex, nonlinear evolution of the dynamical systems.

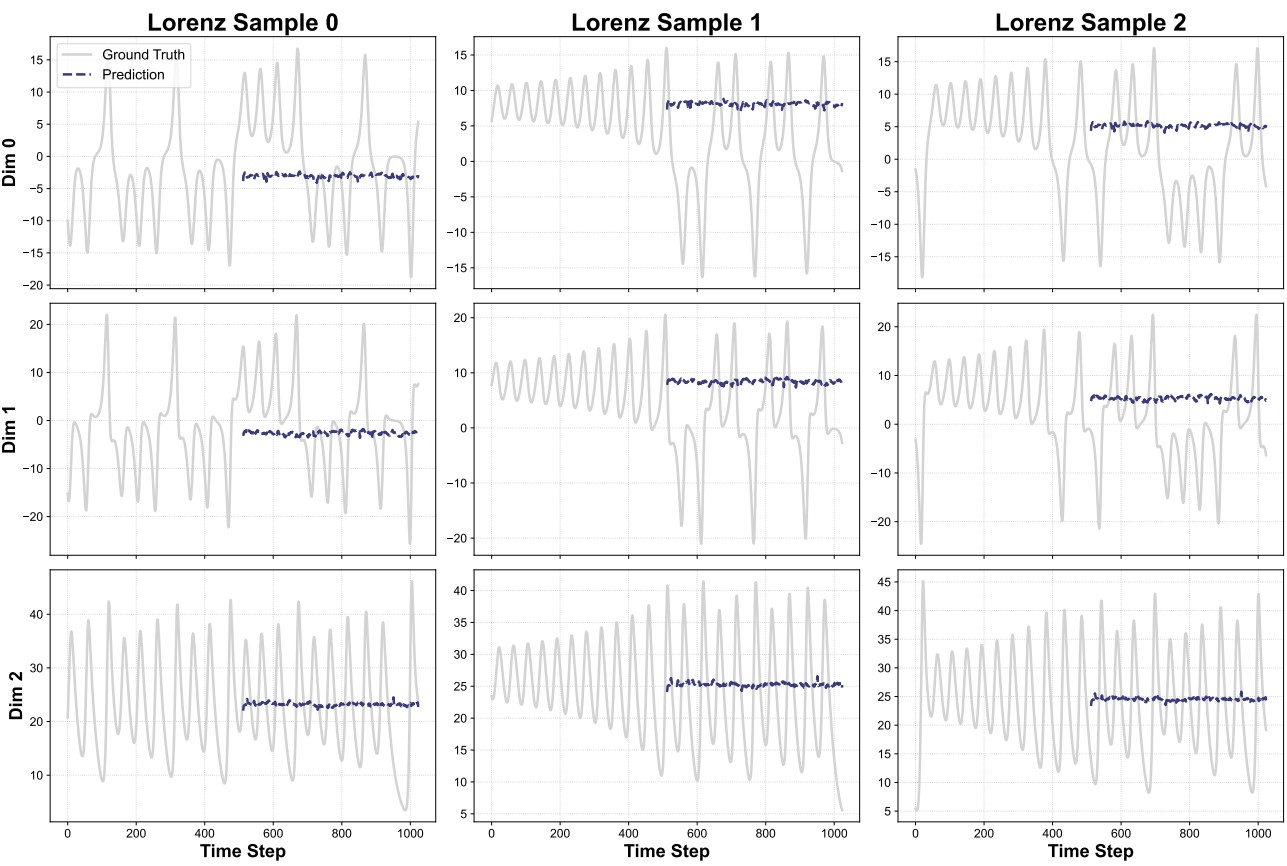

*Figure 20.* Performance collapse of FEDFormer in zero-shot forecasting of Lorenz63 system.

## C.2. Skip Connection Blocks

To mitigate the loss of fine-grained information during the down-sampling operations within the encoder, we employ a skip connection architecture that links encoder and decoder blocks at corresponding resolutions. This mechanism is crucial for providing the decoder with direct access to high-resolution feature maps from the encoder, thereby enhancing the model's ability to reconstruct the system's dynamics with high fidelity.

Our implementation for these skip connections is a specialized 1D residual convolutional block. Its design is inspired by modern convolutional networks that have successfully integrated principles from Transformer architectures, showing high efficiency and performance (Herde et al., 2024). The block operates on different variables independently. The forward pass consists of the following key operations:

- **Depthwise Convolution.** The core of the block is a 1D depthwise convolution with a large kernel size, which uses 7 in our experiments. This operation efficiently captures local spatio-temporal patterns across the patch sequence.

- **Normalization.** Following the convolution, a LayerNorm layer is applied to the features. This standardizes the

activations across the feature dimension, ensuring stable training dynamics.

- **Inverted Bottleneck.** The architecture employs an inverted bottleneck design, a hallmark of modern efficient networks. The normalized features are first passed through a point-wise convolution that expands the channel dimension by a factor of 4. This is followed by a GELU activation function, which introduces non-linearity. A second point-wise convolution then projects the features back to the original dimension. This expand-and-contract structure allows the model to learn complex interactions between channels in a higher-dimensional space.

- **Stability and Regularization.** For improved training, two advanced techniques are integrated. First, a learnable, per-channel scaling parameter is applied to the output of the inverted bottleneck. This allows the model to dynamically modulate the contribution of each residual block, which is particularly beneficial in deep architectures. Second, stochastic depth randomly sets the output of the block to zero during training, effectively bypassing it. This acts as a powerful regularizer, preventing feature co-adaptation and improving model generalization.

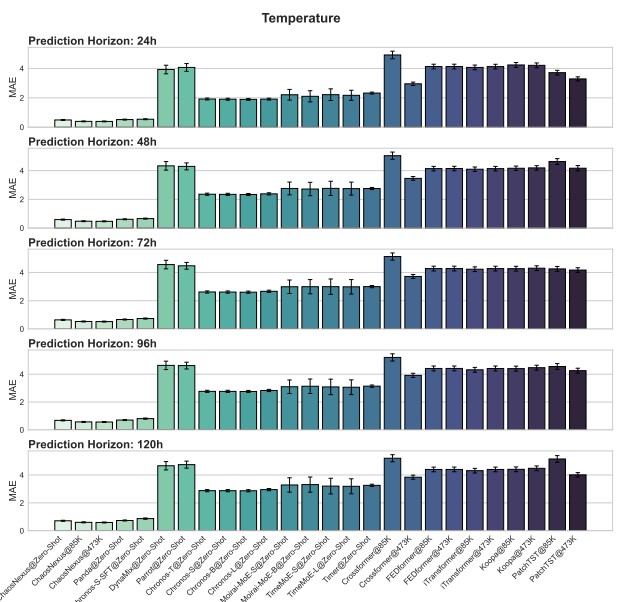

*Figure 21.* Forecasting performance for temperature on the WEATHER-5K dataset. The Mean Absolute Error (MAE) of ChaosNexus and baseline models is compared across multiple prediction horizons after fine-tuning on 85K (0.1%) and 473K (0.5%) samples.

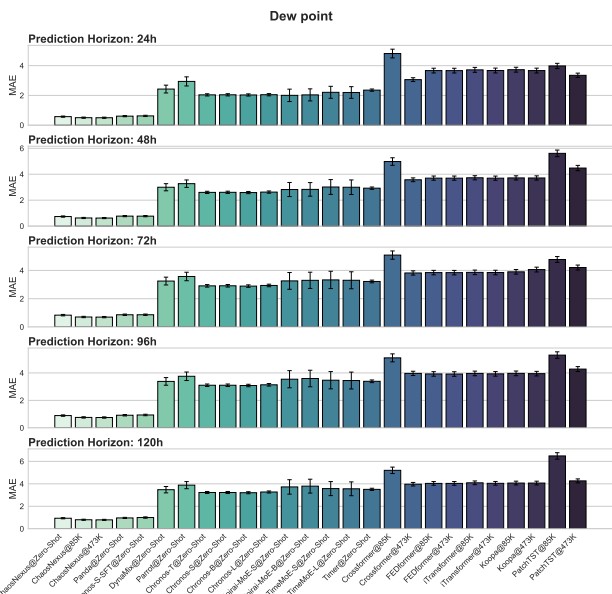

*Figure 22.* Forecasting performance for dew point on the WEATHER-5K dataset. The Mean Absolute Error (MAE) of ChaosNexus and baseline models is compared across multiple prediction horizons after fine-tuning on 85K (0.1%) and 473K (0.5%) samples.

- **Residual Connection.** Finally, the output of the processed branch is added to the original input tensor, forming the block's essential residual connection.

By integrating these blocks as skip connections, we ensure that the decoder has access to a rich, multi-scale representation of the input, enabling it to accurately reconstruct detailed system dynamics that might otherwise be lost in the encoder's hierarchical processing.

### C.3. Wavelet Scattering Transform

In our work, we employ the Wavelet Scattering Transform (WST) to extract a stable, multi-scale frequency representation from the historical observations $\boldsymbol{X}$. The WST (Mallat, 2012; Bruna & Mallat, 2013; Andén & Mallat, 2014) generates signal representations that are stable to small time shifts and deformations without sacrificing significant information. It achieves this by cascading wavelet convolutions with complex modulus non-linearities, followed by local averaging. This hierarchical structure is analogous to that of a Convolutional Neural Network (CNN), but with fixed, pre-defined wavelet filters instead of learned kernels. The transform is constructed by iteratively applying three fundamental operations: convolution with an analytic wavelet filter $\psi_\lambda(t)$, complex modulus non-linearity $|\cdot|$, and averaging via convolution with a low-pass filter $\phi_J(t)$.

For an input signal $x(t)$, the scattering transform up to the second order, denoted as $S_J x$, is a collection of coefficients

from different layers (or orders):

$$S_J x = [S_J^{(0)} x, S_J^{(1)} x, S_J^{(2)} x], \qquad (19)$$

where each order is defined as follows:

**Zero-Order Coefficients.** The zeroth-order coefficients capture the local mean of the signal. They are computed by convolving the input signal $x(t)$ with a wide low-pass filter $\phi_J(t)$, where $J$ defines the scale of temporal averaging, formulated as follows:

$$S_J^{(0)} x(t) = x \star \phi_J(t).$$

This provides the coarsest, most stable representation of the signal's energy.

**First-Order Coefficients.** The first-order coefficients form the core of the wavelet analysis. The calculation begins by convolving the signal $x(t)$ with a family of first-order analytic wavelets, $\psi_\lambda^{(1)}(t)$, to capture information around specific frequencies $\lambda$. The complex modulus of this result is then taken—a crucial step that demodulates the signal and ensures invariance to local phase shifts. Finally, this resulting envelope is smoothed by convolving it with the low-pass filter $\phi_J(t)$, which achieves local time-shift invariance through averaging. The complete operation is summarized by the formula:

$$S_J^{(1)} x(t, \lambda) = |x \star \psi_\lambda^{(1)}| \star \phi_J(t).$$

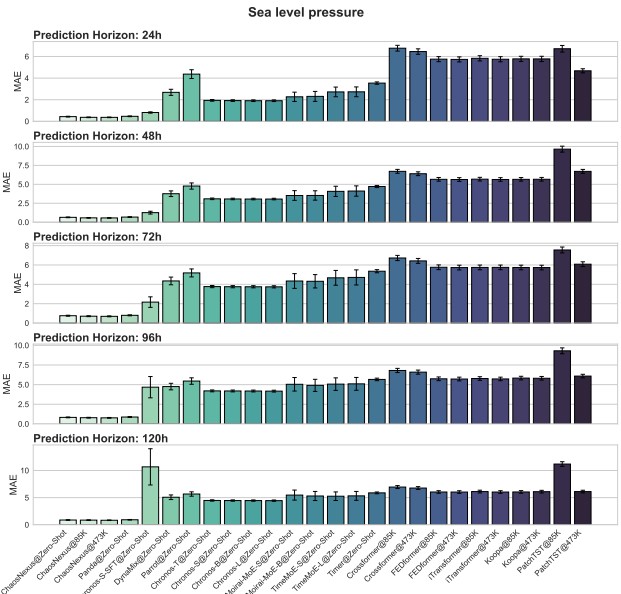

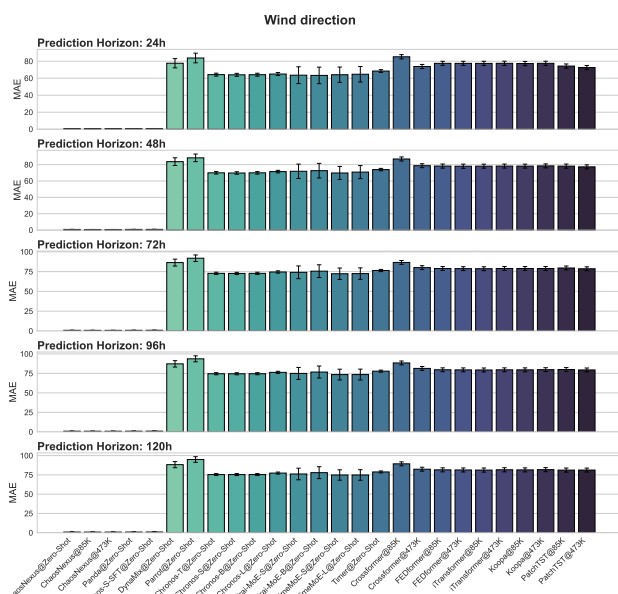

*Figure 23.* Forecasting performance for sea level pressure on the WEATHER-5K dataset. The Mean Absolute Error (MAE) of ChaosNexus and baseline models is compared across multiple prediction horizons after fine-tuning on 85K (0.1%) and 473K (0.5%) samples.

*Figure 24.* Forecasting performance for wind direction on the WEATHER-5K dataset. The Mean Absolute Error (MAE) of ChaosNexus and baseline models is compared across multiple prediction horizons after fine-tuning on 85K (0.1%) and 473K (0.5%) samples.

**Second-Order Coefficients.** To recover transient information, such as rapid amplitude modulations lost during first-order averaging, the transform recursively applies the wavelet decomposition. This process begins with the modulus envelopes, $|x \star \psi_\lambda^{(1)}|$, generated by the first order. These envelopes are then convolved with a second family of wavelets, $\psi_\mu^{(2)}(t)$, to extract their spectral content, which reveals interactions between the primary frequency bands. Following this, a second modulus operation is applied before the final averaging with the low-pass filter $\phi_J(t)$ stabilizes the representation. The entire cascade is encapsulated by the formula:

$$S_J^{(2)}x(t, \lambda, \mu) = ||x \star \psi_\lambda^{(1)}| \star \psi_\mu^{(2)}| \star \phi_J(t).$$

In our methodology, the collection of all scattering coefficients, $\{S_J^{(0)}, S_J^{(1)}, S_J^{(2)}\}$, forms the feature set $\boldsymbol{F}_w \in \mathbb{R}^{C \times T' \times V}$. Here, $C$ represents the total number of scattering paths (i.e., combinations of $\lambda$ and $\mu$), $T'$ is the reduced temporal dimension after averaging, and $V$ is the number of variables. To create a single, fixed-size fingerprint for the underlying dynamical system, we apply temporal pooling across the $T'$ dimension. This results in the final representation $\bar{\boldsymbol{F}}_w \in \mathbb{R}^{C \times V}$, which summarizes the intrinsic oscillatory and modulatory characteristics of the system, serving as a robust conditional input for our model.

## C.4. Maximum Mean Discrepancy

Forecasting the long-term evolution of chaotic systems necessitates metrics that extend beyond point-wise accuracy. To ensure our model reproduces not just a single trajectory but the system's intrinsic statistical and geometric structure, we employ a distributional loss based on the Maximum Mean Discrepancy (MMD).

As established in prior literature (Schiff et al., 2024), a suitable metric for comparing state distributions of trajectories should exhibit several essential characteristics. Specifically, it must: (i) respect the underlying geometry of the state space and be capable of comparing distributions with non-overlapping supports; (ii) provide an unbiased estimator that can be computed from finite samples; (iii) maintain low computational complexity with respect to both dimensionality and sample size; (iv) act as a true metric on the space of probability measures, ensuring that a vanishing distance implies convergence; and (v) feature parametric estimation rates, such that sample error is independent of the system's dimension.

The family of Integral Probability Metrics (IPMs) (Müller, 1997) provides a general framework that satisfies these desiderata. For any two probability distributions $p_1$ and $p_2$, an IPM is defined as the supremum of the difference between expectations over a class of functions $\mathcal{K}$:

$$\text{IPM}(p_1, p_2) = \sup_{\kappa \in \mathcal{K}} |\mathbb{E}_{\boldsymbol{u} \sim p_1}[\kappa(\boldsymbol{u})] - \mathbb{E}_{\boldsymbol{u}' \sim p_2}[\kappa(\boldsymbol{u}')]|. \quad (20)$$

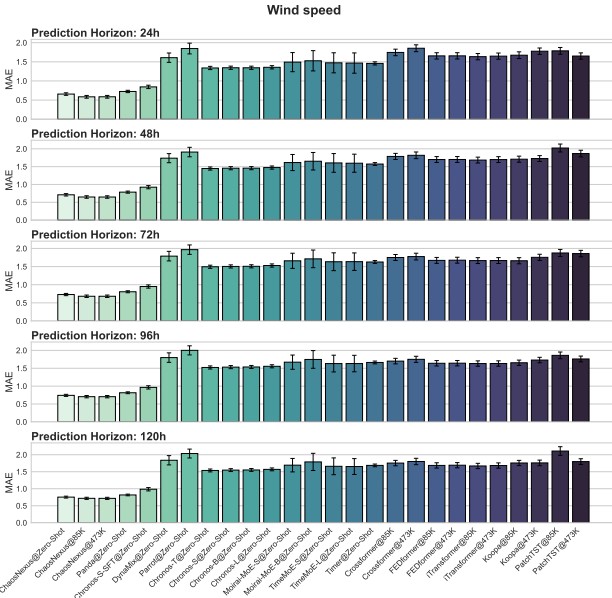

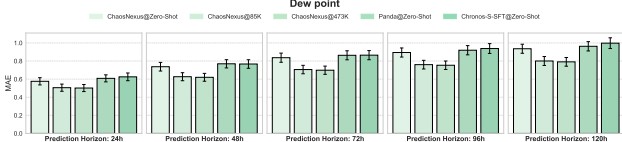

*Figure 27.* Forecasting performance for dew point on the WEATHER-5K dataset. The Mean Absolute Error (MAE) of ChaosNexus and baseline models is compared across multiple prediction horizons after fine-tuning on 85K (0.1%) and 473K (0.5%) samples. Only models previously trained with synthetic chaotic systems are reported.

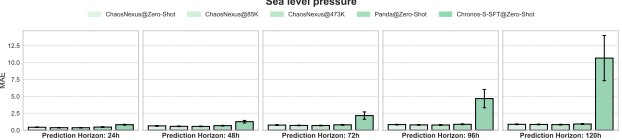

*Figure 28.* Forecasting performance for sea level pressure on the WEATHER-5K dataset. The Mean Absolute Error (MAE) of ChaosNexus and baseline models is compared across multiple prediction horizons after fine-tuning on 85K (0.1%) and 473K (0.5%) samples. Only models previously trained with synthetic chaotic systems are reported.

*Figure 25.* Forecasting performance for wind speed on the WEATHER-5K dataset. The Mean Absolute Error (MAE) of ChaosNexus and baseline models is compared across multiple prediction horizons after fine-tuning on 85K (0.1%) and 473K (0.5%) samples.

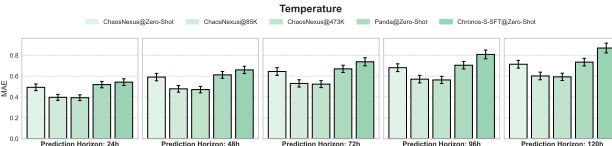

*Figure 26.* Forecasting performance for temperature on the WEATHER-5K dataset. The Mean Absolute Error (MAE) of ChaosNexus and baseline models is compared across multiple prediction horizons after fine-tuning on 85K (0.1%) and 473K (0.5%) samples. Only models previously trained with synthetic chaotic systems are reported.

Within this class, we select the Maximum Mean Discrepancy (MMD), which distinguishes itself by defining $\mathcal{K}$ as the unit ball in a Reproducing Kernel Hilbert Space (RKHS), denoted $\mathcal{H}$. The formal definition of MMD is thus:

$$\text{MMD}(p_1, p_2) := \sup_{||f||_{\mathcal{H}} \leq 1} |\mathbb{E}_{\boldsymbol{u} \sim p_1}[f(\boldsymbol{u})] - \mathbb{E}_{\boldsymbol{u}' \sim p_2}[f(\boldsymbol{u}')]|. \tag{21}$$

By leveraging the reproducing property of the RKHS and the Riesz representation theorem, the squared MMD can be expressed in a convenient analytical form using a kernel function $\kappa(\cdot, \cdot)$ that defines $\mathcal{H}$:

$$\text{MMD}^2(p_1, p_2) = \mathbb{E}_{\boldsymbol{u}, \boldsymbol{u}' \sim p_1}[\kappa(\boldsymbol{u}, \boldsymbol{u}')] + \mathbb{E}_{\boldsymbol{v}, \boldsymbol{v}' \sim p_2}[\kappa(\boldsymbol{v}, \boldsymbol{v}')]$$
$$- 2\mathbb{E}_{\boldsymbol{u} \sim p_1, \boldsymbol{v} \sim p_2}[\kappa(\boldsymbol{u}, \boldsymbol{v})]. \tag{22}$$

This expression leads directly to the unbiased empirical estimator used in our work as the regularization loss $\mathcal{L}_{\text{reg}}$.

For the kernel function $\kappa$, our implementation follows successful precedents (Rasmussen & Williams, 2006; Li et al., 2015; Schiff et al., 2024), employing a mixture of rational quadratic kernels. This choice ensures sensitivity to distributional discrepancies across multiple length scales. The composite kernel is formulated as:

$$\kappa(\boldsymbol{u}, \boldsymbol{v}) = \sum_{\sigma \in \boldsymbol{\sigma}} \frac{\sigma^2}{\sigma^2 + ||\boldsymbol{u} - \boldsymbol{v}||_2^2}, \tag{23}$$

where the set of scale parameters is chosen to be $\boldsymbol{\sigma} = \{0.2, 0.5, 0.9, 1.3\}$, consistent with these prior works.

## D. Hyperparameter Settings

Table 12 delineates the hyperparameter configurations for the suite of ChaosNexus models, spanning from Mini to Large scales. Please note that "ChaosNexus" refers to the "ChaosNexus-Base" variant in all analyses, figures, and tables (except for parameter scaling in Section 4.2), if not explicitly stated. For all model variants, we maintain a consistent 4 hierarchical scales, input context length of $T = 512$ and a prediction horizon of $H = 128$, with the input trajectory segmented into patches of length $D = 8$. The scaling of model capacity is primarily achieved by adjusting the embedding dimension $d_e$, the number of Transformer blocks at each hierarchical scale (Blocks), the corresponding number of attention heads (Heads), and the depth of the convolutional blocks within the skip connections (Skip Depths). Key parameters for our specialized components are kept constant across all scales: each Mixture-of-Experts (MoE) layer consists of $M = 8$ specialist experts, of which the

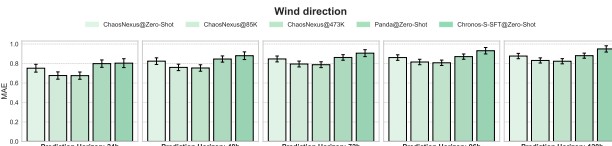

*Figure 29.* Forecasting performance for wind direction on the WEATHER-5K dataset. The Mean Absolute Error (MAE) of ChaosNexus and baseline models is compared across multiple prediction horizons after fine-tuning on 85K (0.1%) and 473K (0.5%) samples. Only models previously trained with synthetic chaotic systems are reported.

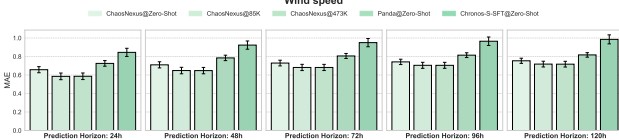

*Figure 30.* Forecasting performance for wind speed on the WEATHER-5K dataset. The Mean Absolute Error (MAE) of ChaosNexus and baseline models is compared across multiple prediction horizons after fine-tuning on 85K (0.1%) and 473K (0.5%) samples. Only models previously trained with synthetic chaotic systems are reported.

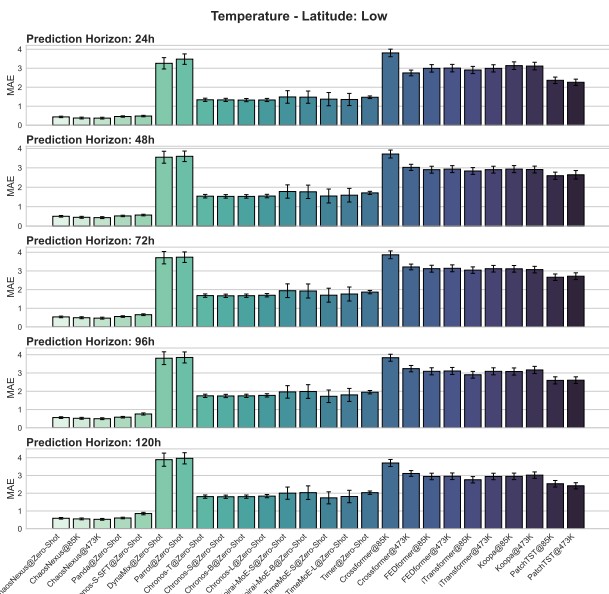

*Figure 31.* Forecasting performance for temperature of low latitude weather stations. The Mean Absolute Error (MAE) of ChaosNexus and baseline models is compared across multiple prediction horizons after fine-tuning on 85K (0.1%) and 473K (0.5%) samples.

top $K = 2$ are activated for each token, and the wavelet scattering transform produces a frequency fingerprint of dimension $C = 48$. This transform is configured with parameters $J = 8$ and $Q = 8$; as detailed in Appendix C.3, $J$ defines the scale of temporal averaging for the low-pass filter, while $Q$ represents the number of wavelet filters per octave (quality factor). The composite training objective is governed by the weights $\lambda_1 = 0.1$ for the MoE load balancing loss and $\lambda_2 = 0.5$ for the MMD-based distributional regularization. The final column reports both the number of activated and total parameters for each model configuration.

# E. Details of Training Setup and Computational Infrastructure

**Training Setup.** We train all ChaosNexus model variants for $100K$ iterations using a global batch size of 1024. The input context length is fixed at 512, and the model forecasts the subsequent 128 time steps. The initial patch size is set to 8. To enable efficient batching across heterogeneous systems, following the existing work (Lai et al., 2026), we randomly sample three channels from each multivariate trajectory to fix the training dimension at $d = 3$. This design aligns with the theoretical minimum of coupled variables required for continuous-time deterministic chaos (Strogatz, 2024). During inference, we process the full multivariate trajectories, since channel attention enables multivariate generalization. The training objective is a weighted sum of MSE, load balancing ($\lambda_1 = 0.1$), and MMD regularization ($\lambda_2 = 0.5$). To ensure convergence stability on chaotic data distributions, we employ the AdamW opti-

mizer. The learning rate is set to $10^{-3}$ and follows a cosine decay schedule with 10% linear warmup. We also apply gradient norm clipping to 1.0 to mitigate gradient explosion, a common challenge in chaotic system modeling. We provide a detailed hyperparameter setting and discussions in Appendix D. For the Panda baseline, we use the same training setup as ChaosNexus for fair performance comparison. To construct the Chronos-S-SFT baseline, we fine-tune the Chronos model for $300K$ iterations using the AdamW optimizer. The per-device batch size is set to 512. The learning rate is initialized at $10^{-3}$ and follows a cosine decay schedule with a 10% linear warmup to ensure stable convergence. We apply gradient norm clipping with a threshold of 1.0 to mitigate gradient explosion. Weight decay is set to 0.0. To enhance the model's robustness, we incorporate a diverse set of augmentations during training, including Random Takens Embedding and Random Fourier Series. The implementation utilizes the Hugging Face Trainer framework with 16 dataloader workers to optimize throughput. For system-specific models, we follow the standard training and evaluation protocols provided in the Time-Series-Library [1] to ensure a fair comparison.

**Computational Resources.** All training experiments are conducted on a node equipped with $8 \times$ NVIDIA A100 GPUs, each with 80GB memory. The training process requires approximately 10 hours without multi-GPU parallelization. Inference is performed on a single NVIDIA A100

---

[1] https://github.com/thuml/Time-Series-Library

*Table 12.* Hyperparameter configurations for ChaosNexus models.

| Method | $T$ | $H$ | $D$ | $d_e$ | Blocks | Attention Heads | Skip Depths | $M$ | $K$ | $C$ | $J$ | $Q$ | $\lambda_1$ | $\lambda_2$ | Params |
|---|---|---|---|---|---|---|---|---|---|---|---|---|---|---|---|
| **ChaosNexus-Mini** | 512 | 128 | 8 | 24 | [1,1,1,1] | [3,6,12,24] | [2,2,2,0] | 8 | 2 | 48 | 8 | 8 | 0.1 | 0.5 | **2.88M/7.60M** |
| **ChaosNexus-Small** | 512 | 128 | 8 | 48 | [1,1,1,1] | [3,6,12,24] | [2,2,2,0] | 8 | 2 | 48 | 8 | 8 | 0.1 | 0.5 | **10.88M/29.72M** |
| **ChaosNexus-Base** | 512 | 128 | 8 | 48 | [2,2,2,2] | [3,6,12,24] | [2,2,2,0] | 8 | 2 | 48 | 8 | 8 | 0.1 | 0.5 | **20.32M/58.01M** |
| **ChaosNexus-Large** | 512 | 128 | 8 | 64 | [3,3,3,3] | [4,8,16,32] | [2,2,2,0] | 8 | 2 | 48 | 8 | 8 | 0.1 | 0.5 | **52.68M/153.12M** |

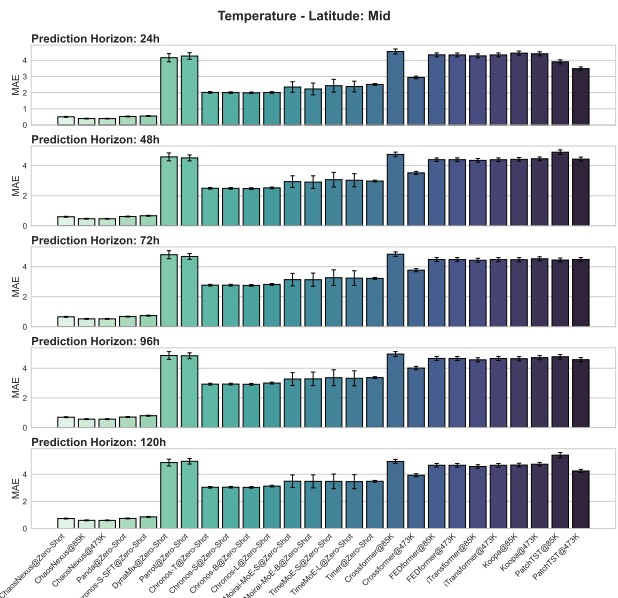

*Figure 32.* Forecasting performance for temperature of mid-latitude weather stations. The Mean Absolute Error (MAE) of ChaosNexus and baseline models is compared across multiple prediction horizons after fine-tuning on 85K (0.1%) and 473K (0.5%) samples.

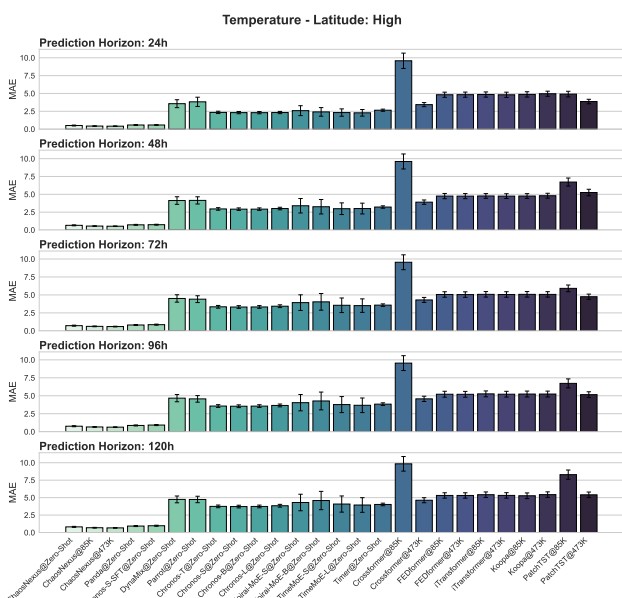

*Figure 33.* Forecasting performance for temperature of high latitude weather stations. The Mean Absolute Error (MAE) of Chaos-Nexus and baseline models is compared across multiple prediction horizons after fine-tuning on 85K (0.1%) and 473K (0.5%) samples.

GPU. Our implementation utilizes PyTorch with BF16 to optimize memory usage and throughput.

## F. Details of Experimental Settings for Evaluations on Synthetic Chaotic Systems

### F.1. Details of Synthetic Chaotic System Dataset

The study utilizes the large-scale synthetic dataset of chaotic dynamics introduced by Lai et al. (2026). This dataset is specifically designed to provide a vast and dynamically diverse corpus for pretraining a universal forecasting model, moving beyond reliance on a limited set of well-known systems. For completeness and the reader's convenience, we briefly summarize the methodology used by Lai et al. (2026) to create this dataset. Their generation pipeline is rooted in an evolutionary algorithm that discovers and validates novel chaotic ordinary differential equations (ODEs).

**Founding Population and Evolutionary Framework.** The algorithm begins with a founding population of 129 well-documented, human-curated, low-dimensional chaotic systems (Gilpin, 2021; 2023). For these foundational systems, which include canonical examples like the Lorenz equations, the parameters and initial conditions are meticulously tuned to ensure operation within their chaotic regimes, and their integration timescales are standardized based on invariant mathematical properties such as Lyapunov exponents. From this seed set, the evolutionary framework iteratively generates new candidate systems through a cycle of mutation and recombination. The mutation step introduces variation by randomly sampling pairs of parent systems $\dot{\mathbf{x}} = f_a(\mathbf{x}, t; \theta_a)$ and $\dot{\mathbf{y}} = f_b(\mathbf{y}, t; \theta_b)$ as well as applying a parameter jitter, where random Gaussian noise is added to the default parameters of the selected ODEs ($\tilde{\theta}'_a \sim \mathcal{N}(\theta_a, \sigma), \tilde{\theta}'_b \sim \mathcal{N}(\theta_b, \sigma)$). Subsequently, the recombination step combines the mutated parent systems to form a novel child system using a skew product construction:

$$\begin{cases} \dot{\mathbf{x}} = f_a(\mathbf{x}, t; \theta_a) \\ \dot{\mathbf{y}} = \kappa_b f_b(\mathbf{y}, t; \tilde{\theta}'_b) + \kappa_a f_a(\mathbf{x}, t; \tilde{\theta}'_a) \end{cases}$$

This method is chosen for its propensity to preserve

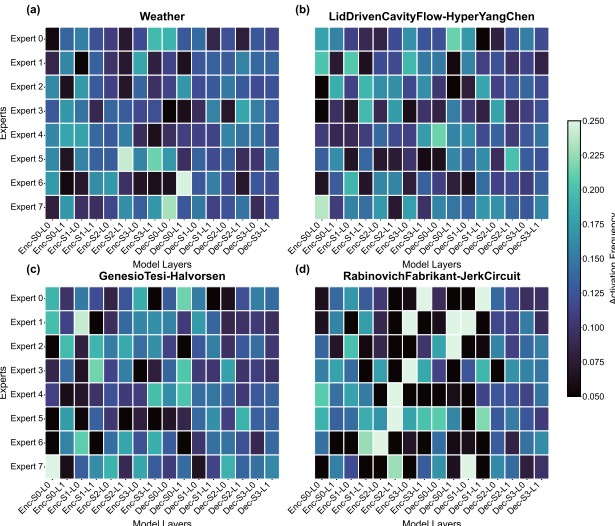

*Figure 34.* Expert activation patterns between the Weather dataset and randomly selected unrelated chaotic systems. S and L denote the scale level and the block within each level, respectively.

chaotic dynamics under sufficiently weak or strong coupling. The scaling factors, $\kappa_a$ and $\kappa_b$, are determined from the reciprocal of the root mean square (RMS), i.e., $\kappa = 1/\sqrt{\mathbb{E}||f(x,t)||^2}$ of a representative trajectory of the parent system.

**Selection for Chaoticity.** A critical and computationally intensive stage of the pipeline involves a rigorous, multi-step selection process that filters for genuine and sustained chaotic behavior, culling all other candidates. First, systems exhibiting trivial dynamics are rejected; the numerical integration is automatically terminated for any candidate that converges to a fixed point (indicated by an integration step size falling below $10^{-10}$), diverges to infinity (a coordinate value exceeding $10^4$), or fails to complete integration within a 5-minute time limit. Surviving candidates are then subjected to the 0-1 test, a standard method for distinguishing between chaotic and periodic or quasiperiodic dynamics. Finally, a further sequence of attractor tests is applied to ensure dynamical complexity. This includes a test based on near-recurrences to reject simple limit cycles, a power spectrum analysis to discard trajectories with only a few dominant frequencies, and an estimation of the largest Lyapunov exponent with the Rosenstein estimator (Rosenstein et al., 1993). This comprehensive discovery and validation process yields a final training corpus of $20K$ unique chaotic dynamical systems.

**Data Augmentation and Trajectory Generation.** To further expand the dataset's volume, several augmentations are applied to the generated trajectories. These transformations are selected because they preserve the underlying property that the resulting time series originates from a valid nonlinear dynamical system. The augmentations include random

time-delay embedding, justified by Takens' embedding theorem (Takens, 2006), convex combinations, and affine transforms. For the final dataset, trajectories of 4096 timesteps are generated for each system using a high-precision numerical integrator with relative and absolute tolerances of $1 \times 10^{-9}$ and $1 \times 10^{-10}$, respectively. Initial conditions are sampled from a preliminary, lower-tolerance integration run to approximate starting on the system's attractor.

**Held-Out Test Set.** For robust zero-shot evaluation, a distinct held-out test set of $9.3 \times 10^3$ systems is created. This set is generated from a reserved subset of 20 systems from the original 129 founding population that are never used in the training set generation. A strict separation is enforced by ensuring that none of these 20 systems, nor any of their mutations, appear as either a driver or a response in the skew product constructions for the training data, thereby preventing any data leakage.

**Statistical Properties of Synthetic Systems.** We conduct a comprehensive statistical analysis of the generated systems. Specifically, we compute the largest Lyapunov exponent for each system with the Rosenstein estimator (Rosenstein et al., 1993), and estimate the correlation dimension using the Grassberger-Procaccia (GP) algorithm (Grassberger & Procaccia, 1983). The histogram of these two critical invariants across synthetic chaotic systems is visualized in Figure 35. The heavy-tailed distribution of the largest Lyapunov exponent confirms that the dataset encompasses a broad spectrum of dynamical behaviors, ranging from weakly to strongly chaotic regimes. The correlation dimension displays a unimodal broad distribution, demonstrating the diversity of fractal geometries characterizing the synthetic strange attractors.

**Symbolic Divergence between Training and Held-Out Founding Systems.** To quantitatively clarify that our evaluation regime tests for true zero-shot generalization rather than mere parameter-shift adaptation, we analyze the structural distinctness of the held-out founding test systems relative to those used for constructing the training dataset. Specifically, we represent the differential equations of all systems as symbolic expression trees and utilize the Tree Edit Distance (TED) to quantify symbolic structural similarity. It measures the minimum number of node operations (insertions, deletions, or re-labeling) required to transform one symbolic tree into another. Crucially, a TED of zero indicates that two systems share an identical functional topology and differ solely in their numerical coefficients, while any nonzero value implies a difference in the equation's functional terms. We compute the minimum TED for each held-out system against the entire set of founding systems used to construct the training dataset. The resulting distribution shown in Figure 36 is concentrated around a distance of 6. This substantial structural gap confirms that the held-out

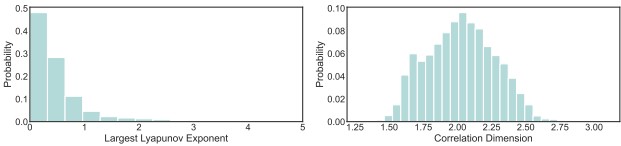

*Figure 35.* Distributions of the largest Lyapunov exponent and the correlation dimension of synthetic chaotic systems.

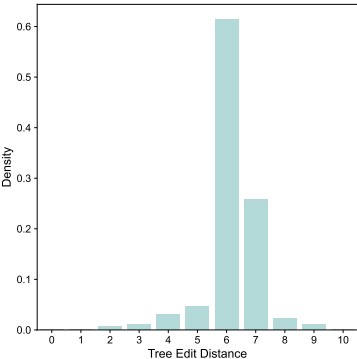

*Figure 36.* Distribution of minimum tree edit distance for each held-out founding system against the entire set of founding systems used to construct the training dataset.

systems belong to topologically distinct equation families, demonstrating that the model's performance relies on universal dynamical learning rather than parameter interpolation within known structures.

## F.2. Details of Controlled Experiment on Lorenz96 System

To rigorously evaluate the efficacy of hierarchical multi-scale modeling, we conduct a controlled experiment using the Lorenz-96 system. This experiment is specifically designed to examine model robustness as the target system transitions from a relatively simple regime to one characterized by increasingly rich multi-scale temporal structures.

The Lorenz-96 system is defined by the following set of coupled ordinary differential equations (ODEs):

$$\frac{dx_i}{dt} = (x_{i+1} - x_{i-2})x_{i-1} - x_i + F, \quad (24)$$

where $x_i$ represents a scalar quantity at $V = 4$ discrete sites. Crucially, we use the external forcing parameter $F$ as a control variable to modulate the dynamics' spectral profile. By systematically varying $F \in \{14, 20, 26, 32, 38, 44, 50, 56, 62\}$, we induce a monotonic increase in Spectral Entropy (SE), which serves as a quantitative proxy for the richness and breadth of the system's multi-scale interactions.

For each configuration of $F$, we generate trajectories of 20,000 time steps using a high-precision numerical inte-

grator with a step size of $dt = 0.01$. To ensure that the dynamics sufficiently converge to the strange attractor, the initial 2,000 steps are discarded as a burn-in period. The resulting data are segmented using a sliding window of 1,024 steps with a stride of 128. Within each window, the first 512 steps serve as the historical context, while the subsequent 512 steps define the prediction horizon.

We perform a zero-shot comparison between ChaosNexus and Panda. The evaluation focuses on the correlation between spectral entropy and forecasting error across multiple dimensions, including point-wise accuracy (sMAPE@128 and sMAPE@512), correlation dimension error ($D_{\text{frac}}$), KL divergence between state distributions ($D_{\text{stsp}}$), largest Lyapunov exponent error ($D_{\text{lyap}}$), and weighted mean energy error ($\text{ME}_{\text{LRw}}$).

### F.3. Details of Evaluation Metrics

To provide a comprehensive assessment of model performance, we employ a suite of evaluation metrics that quantify both short-term, point-wise prediction accuracy and the long-term fidelity of the reconstructed system dynamics. These metrics are designed to evaluate a model's ability to not only forecast the immediate future state but also to reproduce the intrinsic geometric and statistical properties of the chaotic attractor.

**sMAPE.** For evaluating short-term predictive quality, we utilize the Symmetric Mean Absolute Percentage Error (sMAPE) calculated over a forecast horizon of length $T$. The sMAPE provides a normalized, point-wise measure of the discrepancy between the predicted trajectory and the ground truth. It is defined as:

$$\text{sMAPE} \equiv \frac{200}{T} \sum_{t=1}^{T} \frac{\|\mathbf{x}_t - \hat{\mathbf{x}}_t\|_1}{\|\mathbf{x}_t\|_1 + \|\hat{\mathbf{x}}_t\|_1}, \quad (25)$$

where $\mathbf{x}_t$ and $\hat{\mathbf{x}}_t$ are the true and forecasted state vectors at time step $t$, respectively. This metric is particularly well-suited for this task as its percentage-based formulation is robust to the varying scales of different dynamical systems, and it is less sensitive to outliers than the Mean Absolute Error (MAE).

**Correlation Dimension Error $D_{\text{frac}}$.** To assess a model's ability to replicate the long-term geometric structure, we evaluate its reproduction of the system's strange attractor. In a chaotic dynamical system, long-term trajectories populate a fractal object known as a strange attractor, which possesses a unique and invariant fractal dimension that characterizes its space-filling properties. We use the correlation dimension as a non-parametric method to estimate this fractal dimension directly from the time series data (Grassberger & Procaccia, 1983). This method quantifies how the number of points on the attractor scales with distance by measuring,

for each point, the density of neighboring points within a given radius $r$. The fractal dimension is revealed by the power-law relationship between this point density and the radius $r$. We compute the correlation dimension for both the ground-truth trajectory and the attractor generated from the model's long-term forecast. The metric $D_{\text{frac}}$ is then the root mean square error (RMSE) between these two estimated dimensions. A smaller $D_{\text{frac}}$ value signifies that the model's generated dynamics faithfully reproduce the intrinsic geometric complexity of the true system's attractor.

**Kullback–Leibler Divergence between System Attractors ($D_{\text{stsp}}$).** Beyond geometric structure, a successful long-term forecast must also capture the statistical properties of the attractor. We quantify this using the Kullback-Leibler (KL) divergence ($D_{\text{stsp}}$) between the probability distributions of the true and reconstructed attractors (Hess et al., 2023; Göring et al., 2024). The long-term behavior of a chaotic system can be described by an invariant probability measure over its phase space, which represents the likelihood of finding the system in a particular state. Operationally, we approximate this invariant measure for both the true and forecasted trajectories by fitting Gaussian Mixture Models (GMMs) to points sampled from each attractor. The $D_{\text{stsp}}$ is then the estimated KL divergence between these two GMMs (Hershey & Olsen, 2007). A lower value indicates that the reconstructed attractor more accurately captures the statistical and density profile of the true system's dynamics.

**Largest Lyapunov Exponent Error ($D_{\text{Lyap}}$).** While geometric and statistical metrics ($D_{\text{frac}}$ and $D_{\text{stsp}}$) assess the static shape and density of the attractor, they do not explicitly measure the temporal dynamics of system instability. To verify if the model captures the hallmark of chaos—sensitivity to initial conditions—we evaluate the Largest Lyapunov Exponent (LLE). The LLE quantifies the average exponential rate of divergence of infinitesimally close trajectories. We estimate the LLE for both the ground-truth trajectory and the model's long-term forecast using the Rosenstein estimator (Rosenstein et al., 1993). The metric $D_{\text{Lyap}}$ is defined as the absolute difference between these two estimated exponents. A low $D_{\text{Lyap}}$ value indicates that the model has successfully internalized the governing physical laws that drive the chaotic evolution, rather than merely memorizing superficial patterns.

**Weighted Mean Energy Error ($\text{ME}_{\text{LRw}}$).** To rigorously evaluate the spectral fidelity of the forecasted trajectories, we assess the model's ability to reproduce the system's energy distribution across the frequency domain. While standard time-domain metrics may overlook spectral distortions hidden within smooth predictions, $\text{ME}_{\text{LRw}}$ explicitly quantifies the deviation in the Power Spectral Density (PSD). To prioritize these dynamically significant components over background noise, we employ a weighted formulation de-

fined as:

$$\text{ME}_{\text{LRw}} = \sum_i w_i |\log(\frac{P_{\text{pred}}(f_i)}{P_{\text{true}}(f_i)})|, \tag{26}$$

where $P_{\text{pred}}(f_i)$ and $P_{\text{true}}(f_i)$ represent the PSD values of the predicted and ground-truth trajectories at frequency $f_i$, respectively. The weighting coefficient $w_i$ is normalized by the total energy of the ground truth signal:

$$w_i = \frac{P_{\text{true}}(f_i)}{\sum_j P_{\text{true}}(f_j)}. \tag{27}$$

This weighting mechanism ensures that the metric is sensitive to errors in high-energy frequency bands while being robust to negligible fluctuations in low-energy regimes. A lower $\text{ME}_{\text{LRw}}$ indicates that the model has faithfully reconstructed the intrinsic oscillatory properties and energy profile of the chaotic system.

### F.4. Details of Baselines

We compare our proposed method against several state-of-the-art time series foundation models, including Panda (Lai et al., 2026), Time-MoE (Shi et al., 2025), TimesFM (Das et al., 2024), Chronos (Ansari et al., 2024), Moirai-MoE (Liu et al., 2025a), and Timer-XL (Liu et al., 2025b). To assess the adaptability of general-purpose models to this specific domain, we also include Chronos-S-SFT, a variant of the Chronos-S model that has been fine-tuned on our chaotic systems training corpus. The key characteristics of each baseline are detailed below.

- **Panda** is a pretrained, encoder-only Transformer model designed for forecasting chaotic dynamics. Based on the PatchTST (Nie et al., 2023) architecture, it introduces interleaved channel and temporal attention layers to capture variable coupling, alongside a dynamics embedding layer that uses polynomial and Fourier features inspired by Koopman operator theory.

- **Time-MoE** is a family of billion-scale, decoder-only Transformer foundation models that utilize a sparse Mixture-of-Experts (MoE) architecture to enhance scalability and computational efficiency. The model tokenizes the input time series point-wise and employs multiple forecasting heads to predict at different resolutions simultaneously through multi-task optimization. Time-MoE is pre-trained on Time-300B, a large-scale collection of over 300 billion time points from diverse domains, to achieve universal forecasting capabilities.

- **TimesFM** is a decoder-only Transformer-based foundation model for zero-shot time series forecasting. It processes time series data by breaking it into patches and is trained autoregressively to predict the next patch based

*Table 13.* The number of time points within the pretraining corpus of different methods.

| Method | ChaosNexus | Panda | Time-MoE | TimesFM | Moirai-MoE | Timer-XL |
|---|---|---|---|---|---|---|
| **# Time Points** | $\sim$0.35B | $\sim$0.35B | $\sim$300B | $\sim$100B | $\sim$231B | $\sim$232B (LOSTA & UTSD) |

*Table 14.* The number of parameters of baseline methods. For methods with mixture-of-experts layers, we demonstrate activated parameter counts/total parameter counts.

| Method | ChaosNexus | Panda | Chronos-S | Chronos-B | Chronos-L | Moirai-MoE-S | Moirai-MoE-L | TimeMoE-S | TimeMoE-L | TimerXL | TimesFM |
|---|---|---|---|---|---|---|---|---|---|---|---|
| **# Parameters** | 21M/58M | 21M | 21M | 48M | 205M | 11M/117M | 86M/935M | 50M/113M | 200M/453M | 84M | 500M |

on the preceding context. A key design feature is using an output patch length that is longer than the input patch length to reduce the number of autoregressive steps required for long-horizon forecasting. The model is pretrained on a large corpus of approximately 100 billion time points, combining real-world data from Google Trends and Wikipedia with synthetic data.

- **Chronos** is a framework that adapts existing language model architectures, such as the T5 family, for probabilistic time series forecasting. Its core innovation is the tokenization of continuous time series values into a fixed vocabulary using a simple process of mean scaling and uniform quantization. By treating time series as a sequence of discrete tokens, Chronos is trained from scratch using the standard cross-entropy loss objective common to language models. The training corpus consists of a large collection of public datasets, augmented by synthetic data generated via Gaussian processes and a mixup strategy.

- **Moirai-MoE** is a decoder-only Transformer that improves upon its predecessor, Moirai (Woo et al., 2024), by incorporating a sparse Mixture-of-Experts (MoE) architecture. It replaces heuristic-driven, frequency-specific input/output layers with a single projection layer, delegating the task of modeling diverse time series patterns to specialized experts within the MoE layers, thereby enabling automatic token-level specialization. It also introduces a novel gating function that uses cluster centroids from a pretrained model to guide expert assignments. Moirai-MoE is trained on the LOTSA dataset using a decoder-only objective.

- **Timer-XL** is a causal, decoder-only Transformer designed for unified, long-context time series forecasting. It generalizes the next token prediction paradigm to multivariate time series by flattening 2D time series data into a unified context of patch tokens. Its central architectural innovation is TimeAttention, a causal self-attention mechanism that uses a Kronecker product-based mask and specialized position embeddings to effectively model both intra- and inter-series dependencies. Timer-XL is pretrained on large-scale datasets, such as UTSD and LOTSA, to achieve state-of-the-art zero-shot performance.

- **Chronos-SFT.** To investigate the domain adaptability of general-purpose models, we create a specialized version of Chronos by fine-tuning the publicly available Chronos weights on our chaotic systems training set. This process, referred to as Supervised Fine-Tuning (SFT), allows the model to adapt its learned representations from general time-series data to the specific, complex patterns inherent in chaotic dynamics. This baseline helps to disentangle the effects of model architecture from the benefits of domain-specific training data.

- **DynaMix.** It is a foundation architecture specifically engineered for zero-shot dynamical systems reconstruction (DSR). It employs a Mixture-of-Experts (MoE) framework where the individual experts are Almost-Linear RNNs (AL-RNNs), capable of learning parsimonious dynamical representations. A context-aware gating network dynamically selects experts to generalize across diverse attractors without fine-tuning. To ensure the preservation of long-term invariant statistics, DynaMix is pretrained using sparse teacher forcing on a curated corpus of low-dimensional chaotic and cyclic systems, utilizing delay embeddings to reconstruct the underlying state space geometry.

- **Parrot**. It serves as a robust, non-parametric baseline designed to probe the efficacy of learned representations in foundation models. It operates as an efficient in-context nearest-neighbor algorithm: by scanning the provided history for motifs that minimize Euclidean distance to the immediate context, it identifies the closest recurrence and directly copies the subsequent trajectory as the forecast. This approach exploits the determinism and recurrence inherent in strange attractors, demonstrating that simple pattern-matching strategies can often outperform complex deep learning models on chaotic benchmarks.

We summarize the number of time points within the pretraining corpus in Table 13 for comparison. We demonstrate the parameter count in Table 14.

# G. Details of Experimental Settings for Generalization on Weather Forecasting

## G.1. Details of Weather Dataset

WEATHER-5K is a large-scale, public benchmark dataset designed to advance research in Global Station Weather Forecasting (GSWF) and broader time-series analysis. The dataset derives from the Integrated Surface Database (ISD), a global repository of surface observations managed by the National Centers for Environmental Information (NCEI). While the full ISD contains data from over 20,000 stations, many are unsuitable for machine learning applications due to being non-operational, having inconsistent reporting intervals, or containing significant missing values for key variables. The creation of WEATHER-5K involves a meticulous selection process to curate a high-quality subset of stations that are currently operational and provide long-term, hourly reporting of essential weather elements. After the preprocessing stages, the final dataset contains hourly meteorological data from 5,672 stations worldwide over a 10-year period (2014–2023), providing a rich and extensive resource for developing and benchmarking sophisticated forecasting models. Each station's data includes five primary meteorological variables: Temperature, Dew Point, Wind Speed, Wind Direction, and Sea-Level Pressure.

For reproducibility and standardized evaluation, the WEATHER-5K dataset is chronologically divided into three subsets: a training set, a validation set, and a testing set. The training set consists of weather data from 2014 to 2021, the validation set includes data from the year 2022, and the testing set comprises data from 2023. This division follows an 8:1:1 ratio, which supports model training on sufficient historical data, validation on a separate year, and testing on the most recent data for an accurate evaluation. For our experiments under few-shot setting conditions, we use only 0.1% and 0.5% of the training data, respectively.

## G.2. Details of Baselines

We compare ChaosNexus against several strong deep learning baselines in this benchmark, including FEDformer (Zhou et al., 2022), CrossFormer (Zhang & Yan, 2023), PatchTST (Nie et al., 2023), Koopa (Liu et al., 2023), and iTransformer (Liu et al., 2024). The details are as follows:

- **FEDformer** is a Transformer architecture designed for long-term forecasting that addresses the tendency of standard Transformers to neglect global series properties, such as overall trends. It incorporates a seasonal-trend decomposition framework to disentangle the global profile of the series, which is processed separately from the more detailed components. Its core innovation is the replacement of the standard self-attention mechanism with frequency-domain operations. These Frequency Enhanced Blocks (FEB) and Frequency Enhanced Attention (FEA) modules operate on a randomly selected subset of Fourier or Wavelet basis functions, which not only captures the series' global properties more effectively but also achieves linear computational complexity.

- **CrossFormer** explicitly models the cross-dimension dependencies in multivariate time series, a factor often overlooked by models that focus primarily on temporal relationships. Its architecture is defined by three key components. First, a Dimension-Segment-Wise (DSW) embedding partitions each time series variable into segments, creating a 2D vector array that preserves both temporal and dimensional information. Second, a Two-Stage Attention (TSA) layer processes this array by first applying attention across the time axis and subsequently across the dimension axis. To handle a large number of variables efficiently, the cross-dimension stage uses a router mechanism to achieve linear complexity. Finally, these modules are integrated into a Hierarchical Encoder-Decoder (HED) that processes information at multiple scales to generate the final forecast.

- **PatchTST** introduces an efficient Transformer design centered on two principles: patching and channel-independence. The model first segments each univariate time series into patches, which serve as input tokens. This patching strategy retains local semantic information and quadratically reduces the computational and memory complexity of the attention mechanism, which in turn allows the model to process longer historical sequences. Subsequently, the model employs a channel-independent architecture, where each univariate series (channel) is processed individually by a shared vanilla Transformer encoder, thereby learning temporal patterns without explicit cross-channel mixing in the attention layers.

- **Koopa** is a forecasting model built on Koopman theory, specifically designed to handle non-stationary time series by linearizing their underlying dynamics. The model first employs a Fourier Filter to disentangle the series into time-invariant and time-variant components based on their frequency domain characteristics. It then applies distinct Koopman Predictors (KPs) to each component: a globally learned, parametric operator for the time-invariant dynamics, and locally computed, adaptive operators for the time-variant dynamics. These components are organized into stackable Koopman Blocks within a residual architecture, enabling hierarchical learning and end-to-end optimization of the forecasting objective without a reconstruction loss.

- **iTransformer** proposes a novel inversion of the Transformer structure by embedding the entire historical series of each variate independently as a token. This design

repurposes the self-attention mechanism to capture multivariate correlations among different variates, while utilizing the feed-forward network to encode non-linear temporal representations. By treating each variate as a unified token, the model effectively learns complex cross-variate dependencies and avoids the loss of temporal information associated with point-wise embeddings.

# H. Comparison between Chaotic Systems and General Time Series

To elucidate the fundamental dynamical distinctions between chaotic systems and general real-world time series, we conduct a comparative spectral analysis juxtaposing the Lorenz63 system and the Lorenz96 system against representative empirical time series of Electricity and Traffic that are considered by system-specific time-series forecasting models such as FEDFormer (Zhou et al., 2022). To ensure rigorous comparability across these disparate physical scales, we standardize and align all time series to visualize approximately 25 characteristic cycles, with the chaotic system time units calibrated against the daily periodicity of the empirical data. We then compute the Power Spectral Density (PSD) via Fast Fourier Transform (FFT) to map these temporal evolutions into a unified frequency domain ($1/t$ versus $1/day$), thereby isolating their underlying structural frequencies.

We demonstrate the results in Figure 37. The analysis reveals a stark topological dichotomy between the two system classes. Chaotic systems exhibit a continuous broadband spectrum, with energy distributed across a continuum of low frequencies without distinct isolated peaks, a hallmark of intrinsic aperiodicity. In contrast, the general time series exhibits a sparse line-spectrum structure, dominated almost entirely by a few fundamental frequencies (the daily cycle), with negligible energy in the intervening bands. This finding demonstrates that while real-world time series are typically governed by sparse, discrete periodic forcing, chaotic systems are fundamentally characterized by a continuous, multi-scale structure, in which dynamic complexity arises from a rich information density distributed across a broad temporal continuum rather than isolated frequencies.

# I. Relations to Chaotic System Theories

## I.1. Cross-system Generalization

We provide the mathematical intuition for why these components enable generalization across heterogeneous systems:

- **ScaleFormer architecture implements a multi-scale analysis.** Chaotic systems often exhibit multiple distinct timescales, for example, fast oscillations superposed on slow manifolds. The shallow layers (i.e., fine scales) of

ScaleFormer can capture high-frequency dynamics driven by the largest positive Lyapunov exponents, and the deep layers (i.e., coarse scales) capture the global attractor geometry associated with negative exponents. This architecture forces the model to learn the coupling mechanisms between timescales. Since diverse chaotic systems often share similar structural couplings (e.g., relaxational oscillations or bursting patterns) despite differing parameters and equations, explicitly disentangling these scales allows the model to transfer these learned dynamical patterns to unseen systems.

- **MoE layers serve as a basis expansion of local vector fields.** The evolution of a chaotic system can be described by $\dot{\mathbf{x}} = F(\mathbf{x})$. We hypothesize that while global attractors are varied across systems, local vector fields $F(\mathbf{x})$ can be decomposed into a set of local dynamical patterns (e.g., local saddle, spiral, or fold geometries). Mathematically, the MoE layer acts as a functional basis expansion. We view the experts $\{E_k\}_{k=1}^M$ as learned basis functions for local dynamics, MoE approximates the unknown vector field $F_{new}(\cdot)$ of an unseen system as:

$$F_{new}(\mathbf{x}) \approx \sum_{k=1}^M G_k(\mathbf{x}) \cdot E_k(\mathbf{x}), \qquad (28)$$

where $G_k(\cdot)$ denotes the gating coefficient. Generalization occurs because the model learns a reusable dictionary of experts $E_k$ during training. When encountering a new system, the model performs an online system identification by exploring the optimal combination weights $G_k(\mathbf{x})$ from the inputs, allowing it to reconstruct complex dynamics from these shared bases.

- **Wavelet fingerprints have Lipschitz continuity to diffeomorphisms.** If a novel target $x'$ is a deformed version of a source trajectory $x$, modeled by a diffeomorphism operator, the distance in our fingerprint $\Phi$ satisfies the bound:

$$||\Phi(x) - \Phi(x')|| \leq C||x' - x||. \qquad (29)$$

This bound theoretically guarantees that the mapping from the space of dynamical systems to our conditioning embedding space is stable and continuous. It ensures that structurally related systems, even if never seen during training, are mapped to a compact neighborhood in the feature space. It allows ChaosNexus to treat cross-system generalization as a smooth interpolation problem on a structured manifold.

## I.2. Relation to Operator Theory

We discuss the relation of ChaosNexus to operator theory as follows:

- **First**, as detailed in Section 3.1 and Appendix C.1, we pre-process input patches $\mathbf{P}$ using random polynomial

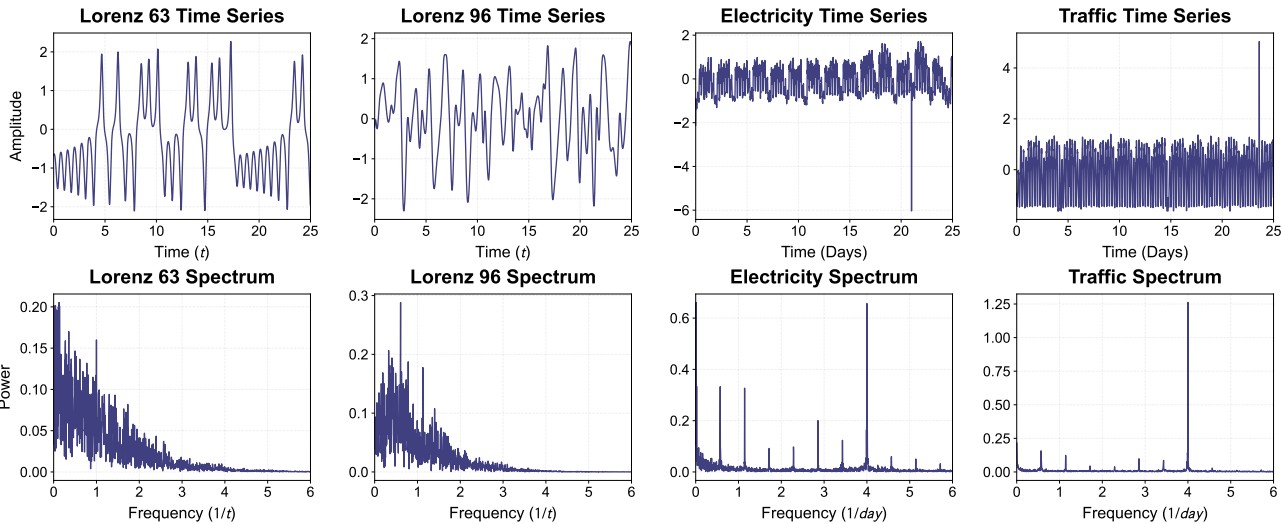

*Figure 37.* Comparison between chaotic systems and general time series.

and Fourier features. Mathematically, it corresponds to constructing a finite dictionary of observables $\Psi(\mathbf{P})$. This step explicitly mimics the lifting process in extended dynamic mode decomposition (eDMD), projecting the highly nonlinear state evolution onto a higher-dimensional manifold where the dynamics are more amenable to linear approximation.

- **Second**, in the lifted space, the time evolution is governed by the Koopman operator $\mathcal{K}$, such that $\Psi(\mathbf{P}_{t+1}) = \mathcal{K}\Psi(\mathbf{P}_t)$. Our ScaleFormer backbone can be theoretically interpreted as a learnable, finite-dimensional approximation of this operator. Unlike traditional eDMD which approximates $\mathcal{K}$ with a static matrix, our ScaleFormer uses the attention mechanism to learn a state-dependent spectral decomposition. The attention weights effectively perform a dynamic eigenvalue decomposition, attending to the specific eigenmodes most relevant for the current phase space region, thereby handling the continuous spectrum often present in chaotic systems.

### I.3. Relation to Invariants

We discuss the relation of ChaosNexus to invariants as follows:

- **First**, chaotic systems are characterized by a spectrum of Lyapunov exponents $\{\lambda_1, \lambda_2, \ldots, \lambda_d\}$. Positive exponents ($\lambda_i > 0$) drive exponential divergence, while negative exponents correspond to dissipative dynamics and attraction to the stable manifold. Our ScaleFormer architecture structurally aligns with this multi-scale dynamical structure. By processing input patches at progressively coarser resolutions, ScaleFormer explicitly disentangles these coupled timescales, where fine-scale layers capture high-frequency fluctuations and local error growth,

corresponding to the dynamics driven by the largest positive Lyapunov exponents, and coarse-scale layers capture long-range dependencies and the global attractor topology, governed by negative Lyapunov exponents. This separation prevents high-frequency chaotic mixing from obscuring the low-frequency invariant structure.

- **Second**, from the ergodic theory perspective, the long-term behavior of a chaotic system is characterized by an invariant physical measure. Our MMD loss minimizes the integral probability metric (IPM, Appendix C.4) between the predicted and true measures. Crucially, we instantiate the MMD with a mixture of rational quadratic (RQ) kernels. Since the RQ kernel is theoretically equivalent to an infinite-scale mixture of Gaussian kernels, it allows the metric to capture distributional discrepancies across a continuum of length scales. This capability ensures the model effectively learns the multi-scale geometry of the strange attractor, even when point-wise forecasting inevitably diverges.

