# OpenReview forum: "ChaosNexus: A Foundation Model for ODE-based Chaotic System Forecasting with Hierarchical Multi-scale Awareness"
_ICML.cc/2026/Conference — ICML 2026 regular_

### Official Review · Reviewer_UJiG · 2026-03-05

**Soundness:** 3
**Presentation:** 3
**Significance:** 3
**Originality:** 2
**Overall Recommendation:** 4
**Confidence:** 4

**Summary:**

This paper introduces a foundation model for low-dimensional chaotic time series featuring several architectural elements that are well suited for forecasting dynamical systems. The authors specifically emphasize capturing the multiscale and spectral properties of the chaotic training data to improve performance. The authors report gains on sMAPE and especially on long-term attractor metrics for their forecasts on an existing evaluation set of chaotic systems.

**Compliance With Llm Reviewing Policy:**

Affirmed.

**Final Justification:**

The authors put extensive effort into our discussion to provide additional experiments to reinforce their points. Moreover, I have independently verified their evaluations (none of the ablations results though as they require training runs). Throughout the discussion period I've been convinced of their claims and thus raise to an accept.

**Key Questions For Authors:**

1. Did you ensure that the representative systems (Lorenz63, Lorenz96, Rossler) are not contained in the training set for the publicly available chaotic systems data?
2. Are baselines for the WEATHER 5K evaluation in figure 5 parameter matched and hyperparameter tuned?
3. The performance discrepancy over the choice of kernel function for the MMD loss in Table 7 is quite large. How much of this delta can be attributed to the fact that the best choice is a mixture of kernels? And how would the table change if the other kernels were used as a mixture?
4. Did you systematically adjust the number of experts for the parameter scaling law in fig. 8a?

**Limitations:**

There is basically no discussion about the limitations of the proposed method. What are the failure modes? A section about the limitations would greatly increase transparency about how the model can fail or what dynamical regimes the model is not suited for. Even some discussion about failure cases within the evaluation set would be illuminating.

**Strengths And Weaknesses:**

#### Strengths

1. Experiments are very extensive and the appendix is extremely thorough
2. Results demonstrate strong gains on spectral metrics and attractor metrics
3. The paper contributes new work in the growing field of large-scale data-driven dynamics forecasting

#### Weaknesses

1. Primary training objective is MSE, but MSE is not included in the pointwise forecasting evaluations
2. Panda and Dynamix appear to provide evaluations longer than the maximum prediction horizon L=512 in this paper (especially Dynamix). Any long term metrics should be evaluated on much longer forecast horizons
3. The ablation experiments do not seem very extensive and most of the gains appear to be from the MMD loss. It is not convincing that all the architectural features are equally important and makes the method appear overly complicated
4. From table 2., the frequency fingerprint component seems to have the least impact across all of the metrics - especially the spectral and long-term statistics metrics which seems to contradict it's purpose

---

> ### Author Rebuttal · Authors · 2026-03-31
>
> > MSE metric.
>
> We supplement **MSE metric** on both synthetic chaotic systems ([link](https://reurl.cc/qpZbxD)) and real-world weather benchmark ([link](https://reurl.cc/3kYdW9)). We find that ChaosNexus also performs the best regarding this metric. We report sMAPE since it is appropriate for **comparing heterogeneous chaotic systems with different value scales**, also facilitating direct comparison with the competitive baseline Panda that uses the same metric.
>
> > Long term metrics for longer forecast horizons.
>
> We supplement long-term metrics for longer prediction horizon. Each test trajectory of our benchmark dataset has length **up to 4096 steps**. We use 512 steps as context and **forecast the subsequent 3584 steps**. This horizon is sufficiently long to fully expose whether the predicted trajectories preserve the underlying attractor-level statistics. The results are shown in [this link](https://reurl.cc/yOmb3y). We find that the **peformance comparison to the baselines remains consistent** with the evaluation setting of 512 steps.
>
> > Additional ablation studies and architectural contribution.
>
> We conduct additional progressive ablation where design components, including PME (patch merging/expansion), MoE, and MMD, are incrementally added. The results are summarized at [this link](https://reurl.cc/ovZj7v). The ablation results show that PME and MoE independently improve key metrics without MMD. Moreover, the combination of PME, MoE, and MMD achieves better performance, **indicating these components are complementary**.
>
> > Effect of the frequency fingerprint (FF).
>
> We evaluate the impact of FF across systems with varying spectral similarity to the training set. We select a subset of test systems, and compute the nearest-neighbor distance in the power spectral density (PSD) space relative to training systems and partitioned the test set into “Easy” (spectrally close), “Medium”, and “Hard” (spectrally far) groups. Statistics are in [this link](https://reurl.cc/8e5alR). Results on $ME_{\text{LRw}}$ are shown at [this link](https://reurl.cc/ovZjVD).
>
> The results show that **the full model consistently outperforms the variant without FF**, with improvements **increasing as spectral heterogeneity grows**: +9.7% for Easy, +11.6% for Medium, and +12.7% for Hard. This confirms that FF is more effective for systems that are spectrally distant from the training distribution, which aligns with its intended purpose of **enhancing generalization across heterogeneous chaotic dynamics**.
>
>
> > Clarification on representative systems inclusion in the training set.
>
> The training systems are generated as **novel child systems through mutation and recombination of parent systems** (Appendix F.1), rather than by directly including canonical systems such as Lorenz63, Lorenz96, or Rossler.
>
> > Baseline protocol for WEATHER-5K.
>
> We follow official implementations and hyperparameters for baselines [1] to ensure a fair comparison. ChaosNexus is evaluated in zero-shot setting and therefore does not require specific hyperparameter tuning.
>
> > Kernel selection and mixture effect.
>
> - First, "Gaussian kernel" in **Table 7** is implemented as **a mixture** over multiple length scales (**Appendix B.11.2**). We will revise the wording to avoid the ambiguity.
>
> - Second, we compare single kernels and their mixtures in [this link](https://reurl.cc/2a89ka). Using **a single rational quadratic kernel already achieves strong performance** (sMAPE@128 = 72.78, $D_{\text{stsp}}$ = 1.348), while the mixture of rational quadratic kernels further improves sMAPE@128 to 68.90 (≈4% relative gain) and $D_{\text{stsp}}$ to 1.206 (≈10% relative gain). In contrast, mixtures of Gaussian kernels do not reach comparable performance (sMAPE@128 = 80.33, $D_{\text{stsp}}$ = 1.431). Therefore, the kernel itself drives most of the improvement, and **the mixture contributes modest additional gain**.
>
> > Effect of the number of experts.
>
> - Parameter scaling law (Fig. 8a) is **not obtained by varying the number of experts**. Instead, we adjust the number of Transformer layers, the hidden dimension, and the number of attention heads as detailed in **Table 8**.
>
> - We **add experiments** that only varies the number of experts in 4,8,16,32. Results are shown in [this link](https://reurl.cc/aXaK73). We find that **increasing the number of experts contributes little to predictive performance** compared to the hidden dimension that directly controls the model’s representational capacity.
>
> > Discussion on failure modes.
>
> Our method is designed for **ODE-based chaotic systems**. For PDE-based spatiotemporal dynamics with grid-like inputs, such as the VKVS dynamics example in Appendix B.6, the data should be projected to a lower-dimensional latent space to match the input format for the model. In such dynamical regime, **our method can be less competitive than architectures specifically designed for native spatiotemporal grids**.
>
> [1] https://github.com/taohan10200/WEATHER-5K

---

> > ### Author Rebuttal · Reviewer_UJiG · 2026-04-01
> >
> > I would like to thank the authors for clarifying my questions and their extensive efforts to address my initial concerns with additional experiments. Overall, I think that the proposed model is a bit overcomplicated, but the authors do show that each feature seems to improve the overall performance. However, I would be interested in the PME + MMD only ablation as well which was not included in the new progressive ablations. I also think that the use of MoE is a promising direction for handling heterogeneity in foundation models.
> >
> > One remaining concern is that even though the authors propose MoE to handle heterogeneity in their training dataset, the reported metrics are presumably aggregated over the entire evaluation dataset and thus the metrics are too coarse to fully understand the model's performance across the eval systems. I would suggest the authors include a breakdown of the forecasting metrics like MSE or sMAPE across structured partitions of the eval systems to better highlight how MoE handles heterogeneity in this task.

---

> > > ### Author Response · Authors · 2026-04-04
> > >
> > > > Results on PME + MMD ablation.
> > >
> > > Following your comment, we **conduct additional experiment on the PME + MMD only ablation** and summarize the results below.
> > >
> > > | Method            | sMAPE@128        | sMAPE@512         | $D_{\text{frac}}$ | $ME_{\text{LRw}}$ | $D_{\text{Lyap}}$ | $D_{\text{stsp}}$ |
> > > | :---------------- | ---------------- | ----------------- | ----------------- | ----------------- | ----------------- | ----------------- |
> > > | + PME   | 75.7857 ± 3.1813 | 105.1268 ± 3.0098 | 0.2218 ± 0.0120   | 2.1871 ± 0.1715   | 0.0968 ± 0.0106   | 1.5609 ± 0.5970   |
> > > | + MMD  | 87.6723 ± 2.9311 | 113.8807 ± 2.3041 | 0.2312 ± 0.0130   | 2.4213 ± 0.1785   | 0.0988 ± 0.0095   | 1.7519 ± 0.6678   |
> > > | + PME & MMD  | 75.332 ± 3.0881  | 104.867 ± 2.6892  | 0.2198 ± 0.0112   | 2.0923 ± 0.2186   | 0.0945 ± 0.014    | 1.4902 ± 1.2970   |
> > > | + PME & MoE  | 75.0278 ± 3.1511 | 104.4003 ± 2.9786 | 0.2193 ± 0.0123   | 2.0554 ± 0.1671   | 0.0927 ± 0.0453   | 1.4211 ± 0.6532   |
> > > | + PME & MoE & MMD | 72.699 ± 3.179   | 104.002 ± 2.930   | 0.2150 ± 0.0100   | 1.771 ± 0.132     | 0.082 ± 0.013     | 1.360 ± 0.440     |
> > >
> > >
> > > Relative to PME only, adding MMD yields consistent improvements across all metrics. The improvements are **more evident on the long-horizon dynamical metrics**, for example, 4.3% (-0.0948) on $ME_{\text{LRw}}$ and 4.5% (-0.0707) on $D_{\text{stsp}}$, which is consistent with our design motivation that MMD mainly improves the preservation of long-term attractor statistics by aligning the predicted and ground-truth state distributions.
> > >
> > > > Detailed metrics related to MoE.
> > >
> > > To address this remaining concern, we perform **a partitioned analysis of the evaluation set** and compared the models with and without MoE on each partition.
> > >
> > > Since the MoE layers operate on latent representations encoded from the observed system trajectories, they primarily address the heterogeneity of trajectory evolution patterns. Based on this perspective, we use Dynamic Time Warping (DTW) distance [1] as our heterogeneity measure, since it is better suited to comparing trajectory evolution patterns than purely pointwise distances. Specifically, **for each evaluation system, we compute its nearest-neighbor DTW distance to the training systems, and then partition the evaluation set into three groups, Easy, Medium, and Hard, from low to high distance**. The statistics are shown in [this link](https://reurl.cc/r0513y). We then report MSE and sMAPE at both the 128-step and 512-step horizons for models with and without MoE. The detailed numerical results are as follows.
> > >
> > > | Split | MSE@128 w/ MoE | MSE@128 w/o MoE | **Δ↓**    | MSE@512 w/ MoE | MSE@512 w/o MoE | **Δ↓**    |
> > > | - | - | - | - | - | - | - |
> > > | Easy (closest) | 0.805 ± 0.411  | 0.886 ± 0.391   | **0.081** | 1.650 ± 0.716  | 1.747 ± 0.722   | **0.097** |
> > > | Medium | 1.512 ± 0.493  | 1.635 ± 0.402  | **0.123** | 2.345 ± 0.866  | 2.589 ± 0.773   | **0.244** |
> > > | Hard (farthest) | 2.279 ± 0.399  | 2.542 ± 0.495  | **0.263** | 2.793 ± 0.853  | 2.989 ± 0.854   | **0.196** |
> > > | **Easy → Hard** | **+1.474** | **+1.656**  | - | **+1.143** | **+1.242**  | - |
> > >
> > >
> > > | Split | sMAPE@128 w/ MoE | sMAPE@128 w/o MoE | **Δ↓**  | sMAPE@512 w/ MoE | sMAPE@512 w/o MoE | **Δ↓**    |
> > > | - | - | - | - | - | - | - |
> > > | Easy (closest) | 41.492 ± 2.169   | 46.029 ± 2.069   | **4.537** | 76.743 ± 2.889 | 81.682 ± 2.723 | **4.939** |
> > > | Medium   | 68.325 ± 2.572   | 75.184 ± 2.526   | **6.859** | 100.951 ± 2.713  | 106.988 ± 2.560 | **6.037** |
> > > | Hard (farthest) | 99.377 ± 3.362   | 106.153 ± 3.283  | **6.776** | 122.965 ± 2.966  | 128.899 ± 2.607   | **5.934** |
> > > | **Easy → Hard**  | **+57.885** | **+60.124**  | - | **+46.222** | **+47.217** | - |
> > >
> > > The results show that **MoE consistently improves performance across all three partitions and all reported metrics**. On the 128-step metrics, adding MoE yields roughly **7–10% (0.081-0.263) relative improvement in MSE and 6–10% relative improvement in sMAPE** across the three partitions. The improvements remain consistent at the 512-step horizon, with additional gains of about **3–7% in MSE and 4–6% in sMAPE**. Moreover, **from Easy to Hard, the degradation is smaller with MoE than without it**, for example, on MSE@128, the increase is 1.474 with MoE versus 1.656 without MoE. This suggests that MoE design is beneficial throughout, and **becomes particularly important when the test system is farther from the training distribution**, where a single shared function becomes more restrictive.
> > >
> > > Overall, this partitioned evaluation provides **more direct evidence that MoE is indeed a promising mechanism for handling heterogeneity** in foundation models for chaotic system forecasting. We will include this analysis, along with the detailed numerical results, in the revised manuscript.
> > >
> > > [1] Berndt, Donald J., and James Clifford. Using dynamic time warping to find patterns in time series. KDD 1994.

---

### Official Review · Reviewer_RADG · 2026-03-10

**Soundness:** 3
**Presentation:** 3
**Significance:** 3
**Originality:** 3
**Overall Recommendation:** 5
**Confidence:** 4

**Summary:**

The paper proposes ChaosNexus, a foundation model for forecasting ODE-based chaotic systems. The method introduces the ScaleFormer architecture to capture multi-scale temporal structures through hierarchical patch processing, and incorporates Mixture-of-Experts (MoE) layers together with a learned frequency fingerprint to address cross-system heterogeneity and encode global spectral characteristics. Experiments on more than 9,000 synthetic systems show improved preservation of long-term attractor statistics while maintaining competitive point-wise prediction accuracy.

**Compliance With Llm Reviewing Policy:**

Affirmed.

**Final Justification:**

The quality of this paper is sufficient for acceptance at ICML, and the authors have substantively addressed most of my concerns in the rebuttal.

**Key Questions For Authors:**

Please check the Weaknesses part.

**Limitations:**

yes

**Strengths And Weaknesses:**

Strengths:
1. This paper studies foundation models for chaotic systems, which is an interesting and worthwhile research topic.
2. This paper is clearly written, easy to read, and does not contain any obvious writing errors.

Weaknesses:
1. The role of the frequency fingerprint is not sufficiently clear. It is only used at the end of the network, where it is concatenated with the multi-scale representations and fed into a linear layer to produce the final prediction. This design may lack theoretical justification from a mathematical perspective, and the authors should further clarify this issue.
2. Although the paper provides relatively detailed statistical analyses of the datasets used, it does not present clear criteria for determining dataset suitability. What is the applicable data scope of ChaosNexus? If one intends to apply this method to other real-world time series datasets, how can one determine which datasets are suitable for ChaosNexus?

---

> ### Author Rebuttal · Authors · 2026-03-31
>
> > The role of the frequency fingerprint is not sufficiently clear. It is only used at the end of the network, where it is concatenated with the multi-scale representations and fed into a linear layer to produce the final prediction. This design may lack theoretical justification from a mathematical perspective, and the authors should further clarify this issue.
>
> - **Motivation.** Chaotic systems in our pretraining corpus exhibit substantial **cross-system spectral heterogeneity**. The frequency fingerprint is introduced as a compact **system-level spectral descriptor** extracted from the input history, which complements the multi-scale representation and helps disambiguate the final prediction under different dynamical regimes.
>
> - **Theoretical justification.** We view the final readout as a conditional decoder in the learned latent space. Let $s$ and $z$ denotes the multi-scale representation and the frequency fingerprint, respectively, we view the forecasting target in latent space as $$Y = G(s,z).$$ We consider a first-order local linearization of $G$ around a reference point $(s_0, z_0)$ $$G(s,z)\approx G(s_0,z_0)
> +\frac{\partial G}{\partial s}\Big|\_{(s_0,z_0)}(s-s_0)
> +\frac{\partial G}{\partial z}\Big|\_{(s_0,z_0)}(z-z_0).$$ Rearranging terms gives $$G(s,z) \approx W_hs+W_fz+b,$$ with approximation $$W_h := \frac{\partial G}{\partial s}\Big|\_{(s_0,z_0)},\quad
> W_f := \frac{\partial G}{\partial z}\Big|\_{(s_0,z_0)},\quad
> b := G(s_0,z_0)-W_h s_0-W_f z_0.$$ This matches our readout form $$\hat Y=[H_{\text{uni}}\|\bar F_w]W_o+b_o = W_hH_{\text{uni}}+W_f\bar F_w+b_o.$$
>
> We will revise the manuscript to make this motivation and interpretation explicit.
>
>
> > Although the paper provides relatively detailed statistical analyses of the datasets used, it does not present clear criteria for determining dataset suitability. What is the applicable data scope of ChaosNexus? If one intends to apply this method to other real-world time series datasets, how can one determine which datasets are suitable for ChaosNexus?
>
> We would like to clarify that ChaosNexus is most suitable for datasets satisfying the following three conditions.
>
> - First, **the observations should arise from an underlying continuous dynamical system**, rather than being dominated by purely discrete event rules or random noise. It is consistent with our problem formulation, which focuses on forecasting ODE-based chaotic systems from historical observations.
>
> - Second, the data should **exhibit multi-scale temporal structure** and **broadband spectral content**, rather than being dominated by only a few narrow periodic dynamical basis. This is the setting for which our hierarchical ScaleFormer architecture is designed. Our analysis in Figure 4 further shows that **the advantage of ChaosNexus becomes more pronounced as the system’s spectral entropy increases**.
>
> - Third, the system should **possess stable long-term statistical or attractor-level properties** that are meaningful forecasting targets beyond short-term point accuracy. Our training objective includes MMD regularization to align the predicted and true long-term state distributions. This design is most appropriate when long-term statistical fidelity matters, rather than when the task is purely short-term local extrapolation.
>
> We will add these discussions to the revised version of our manuscript.

---

> > ### Author Rebuttal · Reviewer_RADG · 2026-04-02
> >
> > I thanks the authors for their answers. The current design only concatenates the frequency fingerprint with the multi-scale decoder representation and passes it through a single linear head for prediction. This implies that the fingerprint can only exert a global linear influence on the output and does not modulate the internal MoE routing or attention layers. From a mathematical and architectural perspective, this late-stage, purely linear conditioning may under-utilize the theoretically motivated properties of the wavelet-based fingerprint. I recommend that the authors justify why such a simple readout suffices.

---

> > > ### Author Response · Authors · 2026-04-03
> > >
> > > We thank the reviewer for this insightful comment. We provide **the justification of our designs** and supplement **additional evidence by experiments** as follows.
> > >
> > > **Justification of late-stage conditioning for the frequency fingerprint:**
> > >
> > > - The frequency fingerprint is a **system-level descriptor**. It summarizes the global spectral identity of the observed system, but **no longer carries patch-specific information**. Therefore, applying such a descriptor to intermediate layers is not more effective, since the same global vector is simply broadcast to all patches and all layers, **without providing new patch-level local information** for expert selections. Therefore, we intend to use it as a late conditioning signal for final **regime-level calibration**.
> > >
> > > - We conduct additional experiments to **concatenate the fingerprint with the input representation of the gating network of the MoE layer** in every Transformer block. The results are shown as follows.
> > >
> > > | Method | sMAPE@128 | sMAPE@512 | $ME_{\text{LRw}}$ | $D_{\text{frac}}$ | $D_{\text{stsp}}$ | $D_{\text{Lyap}}$ |
> > > | :-: | :-: | :-: | :-: | :-: | :-: | :-: |
> > > | ChaosNexus (Late conditioning) | 68.901 ± 3.085 | 100.293 ± 2.766 | 1.562 ± 0.115 | 0.203 ± 0.011 | 1.206 ± 0.392 | 0.065 ± 0.025 |
> > > | MoE routing modulated by the fingerprint | 78.774 ± 3.095 | 103.949 ± 2.442 | 2.106 ± 0.255 | 0.226 ± 0.014 | 1.483 ± 0.835 | 0.081 ± 0.021 |
> > > | w/o FF | 72.699 ± 3.179 | 104.002 ± 2.930 | 1.771 ± 0.132 | 0.215 ± 0.010 | 1.360 ± 0.440 | 0.082 ± 0.013 |
> > >
> > > The results demonstrate that **early router conditioning performs worse than the original design** since it creates a granularity mismatch and weakens the backbone’s temporal modeling. It further supports our interpretation that the fingerprint is more effective as **a global system-level calibrator** during the prediction stage.
> > >
> > >
> > > **Justification of the linear readout function:**
> > >
> > > - We employ a linear readout because both multi-scale representations and the frequency fingerprint are already nonlinear representations of the input.
> > >
> > > - We conduct additional experiments to replace the final linear readout in Eq. (9) with either **an MLP**, i.e., $\hat{X}\_{T+1:T+H}=\mathrm{MLP}([H_{\mathrm{uni}}\|\bar{F}\_w])$, or **a FiLM-style [1] modulation layer**. For FiLM, $\gamma(\bar{F}\_w)=W_\gamma \bar{F}\_w+b_\gamma$ and $\beta(\bar{F}\_w)=W_\beta \bar{F}\_w+b_\beta$ denote learnable scale and shift functions generated from the fingerprint, and we apply $\tilde{H}=\gamma(\bar{F}\_w)\odot H_{\mathrm{uni}}+\beta(\bar{F}\_w)$, followed by $\hat{X}\_{T+1:T+H}=W\tilde{H}+b$. The results are as follows.
> > >
> > > | Method | sMAPE@128 | sMAPE@512 | $ME_{\text{LRw}}$ | $D_{\text{frac}}$ | $D_{\text{stsp}}$ | $D_{\text{Lyap}}$ |
> > > | :-: | :-: | :-: | :-: | :-: | :-: | :-: |
> > > | ChaosNexus (Linear readout) | 68.901 ± 3.085 | 100.293 ± 2.766 | 1.562 ± 0.115 | 0.203 ± 0.011 | 1.206 ± 0.392 | 0.065 ± 0.025 |
> > > | FiLM readout | 71.951 ± 3.078 | 101.343 ± 2.399 | 1.694 ± 0.235 | 0.206 ± 0.013 | 1.273 ± 0.928 | 0.069 ± 0.024 |
> > > | MLP readout | 72.177 ± 3.103 | 101.489 ± 2.548 | 1.721 ± 0.249 | 0.208 ± 0.016 | 1.329 ± 0.549 | 0.070 ± 0.021 |
> > >
> > > It demonstrates that **adding extra nonlinearity at the readout stage do not improve performance over the original design**. Moreover, introducing a nonlinear head could increase the difficulty of optimization, potentially leading to mild overfitting rather than improved generalization.
> > >
> > > Therefore, the current late-stage conditioning and readout function design are sufficient to extract the useful information from the wavelet-based fingerprint. We will include these discussions in the revised manuscript to clarify the rationale for our design choice.
> > >
> > > [1] Perez, Ethan, et al. Film: Visual reasoning with a general conditioning layer. AAAI 2018.

---

### Official Review · Reviewer_qRtZ · 2026-03-12

**Soundness:** 3
**Presentation:** 3
**Significance:** 2
**Originality:** 3
**Overall Recommendation:** 4
**Confidence:** 4

**Summary:**

The authors propose a foundation model for chaotic systems that uses a U-Net like high-level architecture and MoE transformer blocks. They train and test on a wide range of low-dimensional chaotic systems and then test on held out examples as well as a more realistic weather dataset.

**Compliance With Llm Reviewing Policy:**

Affirmed.

**Final Justification:**

The additional experiments in high-dimensional settings provide a useful demonstration of the value of training a foundation model in this setting. I have raised my score as a result.

One note on limitations: The design of the method requires a larger initial input trajectory to then predict future states. This makes it distinct from traditional surrogate models and more in line with time-series prediction, and therefore may limit its applicability for surrogate modeling applications that generally require prediction from a single initial condition.

**Key Questions For Authors:**

1. How does the inference speed and accuracy of this model compare with other models of specific systems? If the goal is to accelerate simulations, how does it compare with traditional ODE solvers? If the goal is to use experimental data, how does it compare with smaller models trained for specific systems?
2. How much benefit do you get from the model when you have noisy training data? Using primarily MSE tends to degrade in this context. Is the added regularization sufficient?
3. How well does this approach work for true high-dimensional spatiotemporal chaos, such as turbulence? Is the low-dimensional training set sufficient in that case?

**Limitations:**

The authors should better discuss some of the limitations of the architecture and training data. It seems to handle low-dimensional chaos well, but it's not clear whether it would deal with true high-dimensional chaos.

**Strengths And Weaknesses:**

**Soundness**
The paper is generally sound although highly empirical in nature. The generality is overstated due to the focus on low-dimensional chaos.

**Presentation**
The paper is generally well-written. I would suggest including a few additional citations of training ML surrogates on chaos:
https://proceedings.mlr.press/v202/hess23a/hess23a.pdf
https://openreview.net/pdf?id=86fUjQ3iVf
https://openreview.net/pdf?id=90qt3kVYaE

**Significance**
In many prediction tasks, speed is paramount and one of the primary benefits of ML surrogate models. For large foundation models, there is the danger of creating too large of a model that then is too slow during inference. It is hard to judge from the paper whether this model falls into this potential trap or not since there is no speed comparisons with non-foundation model baselines.

Furthermore, the experiment performed on 2D Navier-Stokes seems to benefit from the inherent low-dimensionality of the true dynamics. Performing PCA to reduce to 16 components effectively makes it a low-dimensional problem, so perhaps not the best example for PDE prediction.

**Originality**
There seem to be some novel combinations of existing architecture ideas combined with some interpretation in terms of chaotic systems.

---

> ### Author Rebuttal · Authors · 2026-03-31
>
> > The generality is overstated due to the focus on low-dimensional chaos.
>
> We clarify that our goal is to develop a foundation model for **ODE-based chaotic systems** rather than restricted to low-dimensional chaos. We add new experiments on Erdős–Rényi-coupled Lorenz-63 systems [1] with 256 and 512 coupled subsystems, corresponding to **768 and 1536 dimensions**, respectively. The main results are summarized below. These results show that ChaosNexus still remains effective on substantially higher-dimensional ODE systems, and the performance is **on the same order of magnitude as that of lower-dimensional systems**.
>
> | Metric | Dim=768 ChaosNexus | Dim=768 Panda | Dim=1536 ChaosNexus | Dim=1536 Panda |
> |:-:|:-:|:-:|:-:|:-:|
> | sMAPE@128 | **55.907 ± 0.438** | 69.962 ± 1.241 | **55.486 ± 0.369** | 67.033 ± 1.430 |
> | sMAPE@512 | **117.901 ± 0.613** | 125.244 ± 0.508 | **117.396 ± 0.509** | 125.186 ± 0.554 |
> | $D_{\text{frac}}$ | **0.080 ± 0.004** | 0.131 ± 0.005 | **0.092 ± 0.004** | 0.135 ± 0.005 |
> | $ME_{\text{LRw}}$ | **1.148 ± 0.034** | 1.562 ± 0.022 | **1.101 ± 0.032** | 1.710 ± 0.024 |
>
>
> > Regarding concerns on 2D Navier-Stokes experiments and applicability to spatiotemporal chaos.
>
> - **Experimental settings.** The 2D Navier–Stokes experiment is included to **align with the evaluation setting used in baseline work Panda** [2], which also builds a foundation model for ODE-based systems. The PCA reduction follows their practice to convert the PDE field into a hidden space representation that is compatible with our forecasting setup. We conduct this experiment to **show our superior performance** than existing baselines.
>
> - **Applicability to Spatiotemporal Chaos.** ChaosNexus is intended for **ODE-based chaotic systems**, consistent with **prior works in this line of research** [2,3], rather than for spatiotemporal modeling with grid-based inputs [4,5]. Such inputs are not naturally compatible without processing like PCA reduction. Therefore, we would not claim that pretraining only on ODE-based systems is sufficient for generalization to spatiotemporal chaos such as turbulence.
>
> > Including additional citations of training ML surrogates on chaos.
>
> We will incorporate these references accordingly. These works study **system-specific** chaotic system forecasting, spanning both ODE-based (first two papers) and PDE-based settings (last paper), whereas our work is focused on **cross-system zero-shot generalization for ODE-based chaotic system forecasting**.
>
> > Inference speed and accuracy compared with other models of specific systems.
>
> - **Clarification on the goal.** Our goal is not framed around **system-specific surrogates**, i.e., ODE solvers or system specific models trained on experiment data. Differently, our model is pretrained on a large collection of systems and then **generalizes to a previously unseen target system without system-specific retraining**.
>
> - **Inference speed.** We add an accuracy v.s. inference-speed plot on Weather-5K between ChaosNexus and non-foundation-model baselines in [this link](https://reurl.cc/M2XVzn). ChaosNexus achieves **comparable inference speed with time ~0.1s for 512-step prediction**. Importantly, ChaosNexus delivers substantially higher accuracy (MAE < 1) compared to these smaller models (MAE ≈ 4–4.5). Unlike system-specific models that require retraining for each new target system, ChaosNexus provides zero-shot generalization, also significantly reducing deployment cost when adapting to previously unseen systems.
>
>
> > Effect of regularization with noisy training data.
>
> We **add new experiments** in which the training data are **corrupted with noise** at 1%, 5%, 10%, 20%, and 30% of the data standard deviation, and we compare the model with and without the MMD regularization term. The results are reported in [this link](https://reurl.cc/GaXD8x). For noise levels of 1%, 5%, 10%, 20%, and 30% of the data standard deviation, **the model with MMD consistently outperforms the version without MMD**. Specifically, for sMAPE@128, the relative improvement ranges from **7.4% to 9.6%**, while for sMAPE@512, the improvement ranges from **4.4% to 6.0%**. It indicates that the regularization is indeed beneficial in noisy settings.
>
>
> [1] Li et al. Higher-order Granger reservoir computing: simultaneously achieving scalable complex structures inference and accurate dynamics prediction. Nature Communications 2024.
>
> [2] Lai et al. Panda: A pretrained forecast model for chaotic dynamics. ICLR 2026.
>
> [3] Hemmer et al. True zero-shot inference of dynamical systems preserving long-term statistics. NeurIPS 2025.
>
> [4] Sun et al.. Towards a foundation model for partial differential equations: Multioperator learning and extrapolation. Physical Review E 2025.
>
> [5] Liu et al. Prose-fd: A multimodal pde foundation model for learning multiple operators for forecasting fluid dynamics. arXiv 2024.

---

> > ### Author Rebuttal · Reviewer_qRtZ · 2026-04-04
> >
> > Thank you for the additional experiments. If you can scale to ~1000 dimensions, I would still like to see experiments/comparisons on spatiotemporal PDEs, which are always discretized to high-dimensional ODEs when solved numerically. The ODEs generated from discretizing PDEs would likely represent a more complex, high-dimensional problem.

---

> > > ### Author Response · Authors · 2026-04-06
> > >
> > > We thank the reviewer for this helpful suggestion. To directly address this concern, we add experiments on the **Kuramoto–Sivashinsky (KS)** equation, a canonical spatiotemporal chaotic PDE whose numerical solution **induces a high-dimensional ODE after discretization**. Specifically, we set the spatial domain length to $L=100$ and solve the system with a pseudo-spectral solver, retaining 1024 nonzero Fourier modes spanning both positive and negative frequencies. Together with the zero-frequency mode involved in the inverse transform, **this yields a 1025-dimensional state space**. We integrate the system from $t=0$ to $t=100$ with relative and absolute tolerances of $10^{-8}$, and save 4096 uniformly spaced snapshots for evaluation.
> > >
> > > We then perform 512-step zero-shot forecasting with 512 steps of contexts on this PDE-derived system using **ChaosNexus, Panda, DynaMix, and Parrot**, and additionally compare against **a PDE-specific FNO baseline** [1] trained on the same contexts to convergence (5000 epochs; detailed hyperparameters are fully-tuned and provided in [this link](https://reurl.cc/531ZaM)). We summarize the results as follows.
> > >
> > >
> > > | Method | sMAPE@128 | sMAPE@512 | $ME_{\text{LRw}}$@128 | $ME_{\text{LRw}}$@512 |
> > > | :-: | :-: | :-: | :-: | :-: |
> > > | **ChaosNexus** | **27.197 ± 2.819** | **97.642 ± 3.475** | **1.143 ± 0.042** | **1.532 ± 0.065** |
> > > | Panda | 34.025 ± 3.251 | 105.902 ± 2.379 | 1.343 ± 0.037 | 1.664 ± 0.071 |
> > > | DynaMix | 57.198 ± 5.492 | 102.869 ± 5.015 | 3.224 ± 0.198 | 4.606 ± 0.245 |
> > > | Parrot | 67.836 ± 5.430 | 108.047 ± 5.173 | 1.793 ± 0.072 | 1.900 ± 0.147 |
> > > | FNO | 97.298 ± 3.464 | 128.303 ± 4.321 | 2.270 ± 0.101 | 2.967 ± 0.232 |
> > >
> > > We find that **ChaosNexus achieves the best performance among all compared methods**: it obtains 27.20 ± 2.82 sMAPE@128 and 97.64 ± 3.48 sMAPE@512, improving over Panda (34.03 ± 3.25, 105.90 ± 2.38) as well as DynaMix and Parrot. It also **outperforms the FNO baseline by a large margin** (97.30 ± 3.46 sMAPE@128, 128.30 ± 4.32 sMAPE@512). The same trend holds for the long-range energy metric $ME_{\text{LRw}}$, where ChaosNexus again yields the lowest error at both 128 and 512 steps.
> > >
> > > To provide a more complete picture beyond the endpoint metrics, we also include the full performance curves of sMAPE and $ME_{\text{LRw}}$ from 64 to 512 forecasting steps for all methods at [this link](https://reurl.cc/R2zQyz). In addition, we provide qualitative rollout visualizations with three representative prediction cases at 256 steps ([this link](https://reurl.cc/grDEaV)) and three representative prediction cases at 512 steps ([this link](https://reurl.cc/4loGjR)). These additional results show that **the advantage of ChaosNexus persists across rollout lengths and is also reflected in the qualitative prediction trajectories**.
> > >
> > > In summary, these results also provide two important observations:
> > >
> > > - First, although ChaosNexus is pretrained **only on lower-dimensional ODE systems**, **it transfers effectively to a much more complex PDE-derived high-dimensional regime**, scaling beyond the original ODE setting.
> > >
> > > - Second, its advantage over a PDE-specific FNO trained on the same contexts suggests that **large-scale pretraining on diverse dynamical systems provides a strong inductive bias** and substantially **improves data efficiency**.
> > >
> > > Overall, these additional experiments directly show that ChaosNexus remains effective in this more challenging PDE-derived high-dimensional setting.
> > >
> > > [1] Li, Zongyi, et al. Fourier neural operator for parametric partial differential equations. ICLR 2021.

---

### Official Review · Reviewer_An5P · 2026-03-13

**Soundness:** 3
**Presentation:** 2
**Significance:** 3
**Originality:** 3
**Overall Recommendation:** 4
**Confidence:** 4

**Summary:**

The authors propose ChaosNexus, a foundation model for chaotic systems predictions. Observing the difference in the spectral entropy between general time series data and the chaotic systems, the authors find that the current architecture, i.e., the Panda, is insufficient to handle the chaotic systems. Therefore, they propose a novel hierarchical architecture, by processing multi-scale patch sizes with various embedding dimension, through a UNet-like design for long-range dependencies. Moreover, they enable the mixture of experts layers within each block to augment the capacity of each layer. Empirically, they conduct experiments on 20k chaotic ODEs, with a test set consisting of 9.3k systems, and show that on the long-term forecasting task on WEATHTER-5K, the trained model is able to maintain a low precision error.

**Compliance With Llm Reviewing Policy:**

Affirmed.

**Final Justification:**

I thank the authors for the rebuttal efforts. The scalability to 2D/3D PDE (though attempted to be addressed by the PCA projection) and the MOE motivation could be better improved. Therefore, I maintain my boardline positive rating.

**Key Questions For Authors:**

1. I would wonder the computational cost for training and inference. Also, is there a table to summarize the gap between training and testing, especially, how does the model perform if the gap (measured by distributional statistics) become larger?

2. Can the model be used for higher-dimensional systems like 2d PDEs?

3.How long does the model maintain low error for long-term predictions?

**Strengths And Weaknesses:**

###Strength:

1. The idea of learning multi-scale features for long-term foreacasting is insightful.

2. THe empirical advantage is clear.

###Weakness:

1. The digram needs to be fine-tuned. Personally, I am just not sure what’s the meaning of each color, and how to connect e.g., wavelet transforms, and the mixture-of-expert with the notations.

2. From the problem formulation with the input of the shape $T \times V$, it seems that the model cannot extend to 2d/3D PDEs.

3.The goal of mixture of expert is a bit unclear. Is this a must-be design of just for augmenting the performance? Are there any theoretical insights for justifications?

---

> ### Author Rebuttal · Authors · 2026-03-31
>
> > The digram needs to be fine-tuned.
>
> We will refine the diagram to more explicitly illustrate the role of each module and their interactions.
>
> > Applicability to higher-dimensional systems like 2d PDEs.
>
> - **Example performance.** We include performances on the 2D Navier–Stokes equation in **Appendix B.6 and Figures 10–11**. We first apply PCA to project the grid-based state into a latent space that matches the input format of our model ($T\times V$), then learn the dynamics in that latent space. The results demonstrate that ChaosNexus also outperforms baseline methods (Panda and DynaMix).
>
> - **Applicability to PDE systems.** That said, we want to clarify that ChaosNexus is designed for **ODE-based chaotic systems**, in line with the setting of our main baselines such as Panda and DynaMix. While the model can be adapted to PDE cases through such processing, we do not claim a guaranteed advantage on general PDE systems or over methods specifically designed for PDE forecasting.
>
> > The goal of MoE and theoretical insights.
>
> - **Design motivation of MoE.** The foundation model is pretrained across many chaotic systems with markedly different dynamical regimes. The key idea is that **we should not use exactly the same internal transformation for all systems**. MoE makes the shared expert captures common dynamics, while specialized experts model regime-specific residual structure.
>
> - **Theoretical insights.** We provide theoretical justification from a probabilistic view. Let $h$ and $y$ denotes the intermediate representation and the output, respectively. Since **the training corpus is drawn from many chaotic systems**, the output distribution is heterogeneous and can be written as a mixture over latent dynamical regimes:
> $$ p(y\mid h)=\sum_{z=1}^{M} p(z\mid h)\,p(y\mid h,z),$$
> where $z$ denotes a latent regime. Under squared loss, the Bayes-optimal predictor is the conditional expectation: $$f^*(h)=\mathbb{E}[y\mid h]
> =\sum_{z=1}^{M} p(z\mid h)\,\mathbb{E}[y\mid h,z]
> =\sum_{z=1}^{M}\pi_z(h)\,F_z(h).$$ When the data distribution itself is a heterogeneous mixture, the optimal predictor is therefore naturally **a weighted combination of regime-specific functions** rather than a single shared function. Our MoE layer in Equ. 5-7 is precisely an approximation to this form.
>
> - **Empirical validity.** We validate the necessity and effectiveness of the MoE design through both quantitative ablation studies and qualitative analyses. Removing the MoE layers leads to degradation in zero-shot forecasting performance, and visualizations of expert activations across encoder and decoder layers demonstrate clear specialization for different dynamical patterns.
>
> > Computational cost for training and inference.
>
> - **Training cost.** In **Appendix E**, we demonstrate that the training takes approximately 10 hours on a single NVIDIA A100 GPU, and this cost can be further reduced with multi-GPU parallelization.
>
> - **Inference cost.** In **Appendix B.5** and **Table 3**, we demonstrate that forecasting 512 future steps from a 512-step context takes 0.119s on average on a single NVIDIA A100 GPU.
>
> > The gap between training and testing and the model performance if the gap become larger.
>
> - First, as shown in **Figure 34**, we compare the equations of the training and test founding systems using the tree edit distance metric. **Over 60% of system pairs exhibit a tree edit distance greater than 6**, indicating a substantial difference between the training and test distributions.
>
>
> - Second, we select a subset of test systems, and compute the nearest-neighbor distance in the power spectral density (PSD) space relative to the training systems and partitioned the test set into “Easy” (spectrally close), “Medium”, and “Hard” (spectrally far) groups. Detailed statistics are at [this link](https://reurl.cc/8e5alR). We demonstrate results on spectral statistics ($ME_{\text{LRw}}$) as follows.
>
> | Method | Easy | Medium | Hard |
> |:-:|:-:|:-:|:-:|
> | Panda | 1.830 ± 0.312 | 3.146 ± 0.285 | 3.218 ± 0.270 |
> | ChaosNexus | 1.247 ± 0.175 | 2.125 ± 0.218 | 2.135 ± 0.210 |
> | Improv. | 31.9% | 32.5% | 33.6% |
>
> We find that the relative improvement of ChaosNexus over Panda grows with the spectral distance between test and training systems, highlighting that **ChaosNexus is progressively more effective as the distributional gap widens**.
>
> > How long does the model maintain low error?
>
> We evaluate ChaosNexus in a long-term zero-shot forecasting setting for up to 3,584 steps. The results in [this link](https://reurl.cc/yOml4O) show that while sMAPE gradually increases, and the error growth tends to plateau beyond ~1,500 steps. Importantly, **ChaosNexus maintains sMAPE below 100 for the first 500 steps, corresponding roughly to 3.90 Lyapunov times**. Across the entire horizon, ChaosNexus consistently outperforms Panda, highlighting its superior robustness and ability to capture the intrinsic dynamics of previously unseen chaotic systems.

---

> > ### Author Rebuttal · Reviewer_An5P · 2026-04-03
> >
> > I am not sure the theoretical insights from the probabilistic view makes sense, especially the ground of the claim "When the data distribution itself is a heterogeneous mixture, the optimal predictor is therefore naturally a weighted combination of regime-specific functions rather than a single shared function" seems a bit concerning. I would maintain my boardline positive score.

---

> > > ### Author Response · Authors · 2026-04-04
> > >
> > > We would like to clarify that the probabilistic view is meant to intuitively demonstrate that when the training data is drawn from many chaotic systems with different dynamical regimes, **the predictor is naturally encouraged to be input-adaptive rather than fully shared**.
> > >
> > > Here, **we will make this point more concrete** from a dynamical-systems perspective. For ODE-based chaotic systems, the evolution is governed by a vector field $\dot{x}=F(x)$. Although different systems have distinct evolution patterns, their local dynamics can share recurring patterns in state space, such as saddle-like or spiral-like behaviors [1-2]. This motivates **viewing the vector field of a new system as a state-dependent combination of learned local dynamical components**: $$F_{\text{new}}(x)\approx \sum_{k=1}^{M} G_k(x)E_k(x),$$ where $E_k(x)$ denotes a learned expert that captures a reusable local dynamical transformation, and $G_k(x)$ is an input-dependent gating coefficient. Under this interpretation, **the experts form a learned dictionary of reusable dynamical primitives**, while **the router performs an input-adaptive selection/combination mechanism analogous to online regime identification**.
> > >
> > > We also support the justification of the MoE layer with experimental evidence.
> > >
> > > - **Routing Specialization.** We use routing specialization as a mechanism-level check of our interpretation that MoE handles heterogeneity through input-adaptive selection. In Appendix B.9.1, we compare the clustering of expert activation patterns with the ground-truth system-type labels using the **Adjusted Rand Index (ARI)**, where ARI = 1 indicates perfect alignment and ARI = 0 indicates near-random agreement. The ARI is above 0.5 in most layers and reaches 0.9933 at its peak, showing that **expert selection changes systematically with the underlying system dynamics rather than collapsing to an almost shared path**. Concretely, [Figure 14](https://reurl.cc/6G5Znk) shows that Lorenz63, Lorenz96, and Rossler induce clearly separated activation patterns, while [Figure 6](https://reurl.cc/4lo1VV) shows that **Weather and three atmosphere-related synthetic systems exhibit highly similar expert-usage profiles**. Together, these results support that the router performs input-adaptive conditional computation.
> > >
> > > - **Ablation Study.** The ablation study shows that **the MoE layer provides additional modeling capacity beyond a fully shared layer**. Replacing the MoE layers with standard feed-forward layers consistently degrades performance: sMAPE@128 increases from 68.901 to 73.076, sMAPE@512 from 100.293 to 105.298, and $D_{\text{stsp}}$ from 1.206 to 1.560. This indicates that **a single shared function is too restrictive for the heterogeneous dynamics in the training corpus, while MoE design yields a better inductive bias**.
> > >
> > > - **Performances on Varying System Heterogenity.** We further evaluate **whether MoE design becomes more important as the test system is more different to the training systems**. For each evaluation system, we compute its nearest-neighbor Dynamic Time Warping (DTW) distance [3] to the training systems and partition the evaluation set into Easy, Medium, and Hard groups from low to high distance. The statistics are shown in [this link](https://reurl.cc/r0513y). We then report sMAPE at both the 128-step and 512-step horizons for models with and without MoE. The detailed numerical results are as follows.
> > >
> > > | Split | sMAPE@128 w/ MoE | sMAPE@128 w/o MoE | **Δ↓**    | sMAPE@512 w/ MoE | sMAPE@512 w/o MoE | **Δ↓**    |
> > > | - | - | - | - | - | - | - |
> > > | Easy (closest) | 41.492 ± 2.169   | 46.029 ± 2.069    | **4.537** | 76.743 ± 2.889   | 81.682 ± 2.723    | **4.939** |
> > > | Medium | 68.325 ± 2.572   | 75.184 ± 2.526    | **6.859** | 100.951 ± 2.713  | 106.988 ± 2.560   | **6.037** |
> > > | Hard (farthest) | 99.377 ± 3.362   | 106.153 ± 3.283   | **6.776** | 122.965 ± 2.966  | 128.899 ± 2.607   | **5.934** |
> > > | **Easy → Hard** | **+57.885**    | **+60.124**       | - | **+46.222**      | **+47.217**       | - |
> > >
> > >
> > > Across all three groups, **the variant with MoE consistently outperforms the one without MoE on all metrics**. Moreover, **from Easy to Hard, the degradation is smaller with MoE than without it**, for example, on sMAPE@128, the increase is 57.885 with MoE versus 60.124 without MoE. This suggests that MoE design is beneficial throughout, and **becomes particularly important when the test system is farther from the training distribution**, where a single shared function becomes more restrictive.
> > >
> > >
> > > [1] Guckenheimer, John, and Philip Holmes. Nonlinear oscillations, dynamical systems, and bifurcations of vector fields. Springer Science & Business Media, 2013.
> > >
> > > [2] Thompson, John Michael Tutill, and H. Bruce Stewart. Nonlinear dynamics and chaos. John Wiley & Sons, 2002.
> > >
> > > [3] Berndt, Donald J., and James Clifford. Using dynamic time warping to find patterns in time series. KDD 1994.

---

### Decision · Program_Chairs · 2026-04-30

**Decision:**

Accept (regular)

**Comment:**

The submission presents ChaosNexus, a foundational model for zero-shot forecasating of ODE-based chaotic systems that addresses the limitations of existing architectures in capturing multi-scale temporal structures and distrinct spectral characteristics. The primary architectural contribution is the ScaleFormer architecture, which employs a hierarchical multi-scale awareness via varying patch sizes and a U-Net-like design to preserve both long-range dependencies and high-frequency fluctuations. To handle heterogeneity across diverse systems, the model integrates Mixture of Experts layers and conditions forecasts on a learned frequency fingerprint that provides a global spectral view. The authors demonstrate that ChaosNexus achieves superior fidelity in long-term statistics and maintains competitive point-wise accuracy across over 9000 synthetic systems and real-world weather data.

Reviewers highlighted the insightful use of multi-scale features and the innovative combination of hierarchical patch processing with MoE and frequency fingerprints as comprising significant architectural novelty. The experimental setup also drew praise, including a massive synthetic corpus and real-world benchmarks. There was some skepticism surrounding the theoretical justification of the MoE and potential underutilization of the frequency fingerprint (which appeared to have limited impact according to the ablation studies), although these points were mostly addressed by the authors. Reviewer qRtZ points out that ChaosNexus requires a large initial context window that can limit its applicability in some modelling scenarios.

Any remaining criticisms of the work appear to be relatively minor. With this in mind, I am happy to recommend acceptance.